# Learning Constrained Markov Decision Processes With Non-stationary Rewards and Constraints

## Abstract

In *constrained Markov decision processes* (CMDPs) with *adversarial* rewards and constraints, a well-known impossibility result prevents any algorithm from attaining both sublinear regret and sublinear constraint violation, when competing against a best-in-hindsight policy that satisfies constraints on average. In this paper, we show that this negative result can be eased in CMDPs with *non-stationary* rewards and constraints, by providing algorithms whose performances smoothly degrade as non-stationarity increases. Specifically, we propose algorithms attaining $\tilde{\mathcal{O}}(\sqrt{T}+C)$ regret and *positive* constraint violation under *bandit* feedback, where $C$ is a corruption value measuring the environment non-stationarity. This can be $\Theta(T)$ in the worst case, coherently with the impossibility result for adversarial CMDPs. First, we design an algorithm with the desired guarantees when $C$ is known. Then, in the case $C$ is *unknown*, we show how to obtain the same results by embedding such an algorithm in a general *meta-procedure*. This is of independent interest, as it can be applied to *any* non-stationary constrained online learning setting.

## 1 Introduction

Reinforcement learning (Sutton & Barto, 2018) is concerned with settings where a learner sequentially interacts with an environment modeled as a *Markov decision process* (MDP) (Puterman, 2014). Most of the works in the field focus on learning policies that maximize learner's rewards. However, in most of the real-world applications of interest, the learner also has to meet some additional requirements. For instance, autonomous vehicles must avoid crashing (Isele et al., 2018; Wen et al., 2020), bidding agents in ad auctions must *not* deplete their budget (Wu et al., 2018; He et al., 2021), and users of recommender systems must *not* be exposed to offending content (Singh et al., 2020). These requirements can be captured by *constrained* MDPs (CMDPs) (Altman, 1999), which generalize MDPs by specifying constraints that the learner has to satisfy while maximizing their rewards.

We study *online learning* in *episodic* CMDPs (see, *e.g.*, (Efroni et al., 2020)), where the goal of the learner is twofold. On the one hand, the learner wants to minimize their *regret*, which measures how much reward they lost over the episodes compared to what they would have obtained by always using a best-in-hindsight constraint-satisfying policy. On the other hand, the learner wants to ensure that the *(cumulative) constraint violation* is minimized during the learning process. Ideally, one seeks for algorithms with both regret and constraint violation growing sublinearly in the number of episodes $T$.

A crucial feature distinguishing online learning problems in CMDPs is whether rewards and constraints are selected *stochastically* or *adversarially*. Most of the works focus on the case in which constraints are stochastic (see, *e.g.*, (Wei et al., 2018; Zheng & Ratliff, 2020; Efroni et al., 2020; Qiu et al., 2020; Liu et al., 2021; Bai et al., 2023)), with only one exception addressing settings with adversarial constraints (Stradi et al., 2024b). This is primarily motivated by a well-known impossibility result by Mannor et al. (2009), which prevents any learning algorithm from attaining both sublinear regret and sublinear constraint violation, when competing against a best-in-hindsight policy that satisfies the constraints *on average*. However, dealing with adversarially-selected constraints is of paramount importance to cope with real-world environments, which are typically non-stationary.

## 1.1 ORIGINAL CONTRIBUTIONS

The main contribution of this paper is to show how to ease the negative result of (Mannor et al., 2009), by considering CMDPs with *non-stationary rewards and constraints*. Specifically, we address CMDPs where rewards and constraints are selected from probability distributions that are allowed to change *adversarially* from episode to episode. One may think of our setting as bridging the gap between fully-stochastic and fully-adversarial ones. We design algorithms whose performances—in terms of regret and constraint violation—smoothly degrade as a suitable measure of non-stationarity increases. This is called *(adversarial) corruption*, as it intuitively quantifies how much the distributions of rewards and constraints vary over the episodes with respect to some "fictitious" non-corrupted counterparts.

We propose algorithms that attain $\tilde{\mathcal{O}}(\sqrt{T} + C)$ regret and constraint violation, where $C$ denotes the corruption of the setting. We remark that $C$ can be $\Theta(T)$ in the worst case, and, thus, our bounds are coherent with the impossibility result by Mannor et al. (2009). Notably, our algorithms work under *bandit* feedback, namely, by only observing rewards and constraint costs of the state-action pairs visiting during episodes. Moreover, they are able to manage *positive* constraint violation. This means that they do *not* allow for a negative violation (*i.e.*, a constraint satisfaction) to cancel out a positive one across different episodes. This is a crucial requirement for most of the practical applications. For instance, in autonomous driving, avoiding a collision does *not* "repair" a previously-occurred crash.

In the first part of the paper, we design an algorithm, called `NS-SOPS`, which works assuming that the value of the corruption $C$ is known. This algorithm achieves $\tilde{\mathcal{O}}(\sqrt{T} + C)$ regret and positive constraint violation by employing a *policy search* method that is *optimistic* in both reward maximization and constraint satisfaction. Specifically, the algorithm incorporates $C$ in the confidence bounds of rewards and constraint costs, so as to "boost" its optimism and achieve the desired guarantees.

In the second part of the paper, we show how to embed the `NS-SOPS` algorithm in a *meta-procedure* that allows to achieve $\tilde{\mathcal{O}}(\sqrt{T} + C)$ regret and positive constraint violation when $C$ is *unknown*. The meta-procedure works by instantiating multiple instances of an algorithm for the case in which $C$ is known, each one taking care of a different "guess" on the value of $C$. Specifically, the meta-procedure acts as a *master* by choosing which instance to follow in order to select a policy at each episode. To do so, it employs an adversarial online learning algorithm, which is fed with losses constructed starting from the Lagrangian of the CMDP problem, suitably modified to account for *positive* constraint violation. Our meta-procedure is of independent interest, as it can be applied in *any* non-stationary constrained online learning setting, so as to relax the knowledge of $C$.

## 1.2 RELATED WORKS

Within the literature on CMDPs, settings with *stochastic* rewards and constraints have been widely investigated. However, their *non-stationary* counterparts, including *adversarial* ones in the worst case, are still largely unexplored. In the following, we discuss the works that are most related to ours, while we refer the reader to Appendix A for a comprehensive survey of related works.

Qiu et al. (2020) provide the first primal-dual approach to deal with episodic CMDPs with adversarial losses and stochastic constraints, achieving, under full feedback, both sublinear regret and sublinear (non-positive) constraint violation (*i.e.*, allowing for cancellations). Stradi et al. (2024a) are the first to tackle CMDPs with adversarial losses and stochastic constraints under bandit feedback, by proposing an algorithm that achieves sublinear regret and sublinear positive constraint violation. These works do *not* consider settings where constraints are non-stationary, *i.e.*, they may change over the episodes.

Ding & Lavaei (2023) and Wei et al. (2023) consider the case in which rewards and constraints are non-stationary, assuming that their variation is bounded. Our work differs from theirs in multiple aspects. First, we consider *positive* constraint violation, while they allow for cancellations. As concerns the definition of regret, ours and that used by Ding & Lavaei (2023) and Wei et al. (2023) are *not* comparable. Indeed, they employ a dynamic regret baseline, which, in general, is harder than the static regret employed in our work. However, they compare learner's performances against a dynamic policy that satisfies the constraints at every round. Instead, we consider a policy that satisfies the constraints *on average*, which can perform arbitrarily better than a policy satisfying the constraints at every round. Furthermore, the dependence on $T$ in their regret bound is much worse than ours, even when the non-stationarity is small, namely, when it is a constant independent of $T$

(and, thus, dynamic regret collapses to static regret). Finally, we do *not* make any assumption on $T$, while both regret and constraint violation bounds in (Wei et al., 2023) only hold for large $T$.

Finally, Stradi et al. (2024b) are the first to study CMDPs with adversarial constraints. Given the impossibility result by Mannor et al. (2009), they propose an algorithm that, under full feedback, attains sublinear (non-positive) constraint violation (*i.e.*, with cancellations allowed) and a fraction of the optimal reward, thus resulting in a regret growing linearly in $T$. We show that sublinear regret and sublinear constraint violation can indeed be attained simultaneously if one takes into account the corruption $C$, which can be seen as a measure of how much adversarial the environment is. Moreover, let us remark that our algorithms deal with *positive* constraint violation under *bandit* feedback, and, thus, they are much more general than those in (Stradi et al., 2024b).

## 2 PRELIMINARIES

### 2.1 CONSTRAINED MARKOV DECISION PROCESSES

We study *episodic constrained* MDPs (Altman, 1999) (CMDPs), in which a learner interacts with an unknown environment over $T$ episodes, with the goal of maximizing long-term rewards subject to some constraints. $X$ is a finite set of states of the environment, $A$ is a finite set of actions available to the learner in each state, while the environment dynamics is governed by a transition function $P : X \times A \times X \to [0, 1]$, with $P(x'|x, a)$ denoting the probability of going from state $x \in X$ to $x' \in X$ by taking action $a \in A$.[1] At each episode $t \in [T]$,[2] a reward vector $r_t \in [0, 1]^{|X \times A|}$ is sampled according to a probability distribution $\mathcal{R}_t$, with $r_t(x, a)$ being the reward of taking action $a \in A$ in state $x \in X$ at episode $t$. Moreover, a constraint cost matrix $G_t \in [0, 1]^{|X \times A| \times m}$ is sampled according to a probability distribution $\mathcal{G}_t$, with $g_{t,i}(x, a)$ being the cost of constraint $i \in [m]$ when taking action $a \in A$ in state $x \in X$ at episode $t$. We also denote by $g_{t,i} \in [0, 1]^{|X \times A|}$ the vector of all the costs $g_{t,i}(x, a)$ associated with constraint $i$ at episode $t$. Each constraint requires that its corresponding expected cost is kept below a given threshold. The thresholds of all the $m$ constraints are encoded in a vector $\alpha \in [0, L]^m$, with $\alpha_i$ denoting the threshold of the $i$-th constraint.

We consider a setting in which the sequences of probability distributions $\{\mathcal{R}_t\}_{t=1}^T$ and $\{\mathcal{G}_t\}_{t=1}^T$ are selected *adversarially*. Thus, reward vectors $r_t$ and constraint cost matrices $G_t$ are random variables whose distributions are allowed to change arbitrarily from episode to episode. To measure how much such probability distributions change over the episodes, we introduce the notion of *(adversarial) corruption*. In particular, we define the adversarial corruption $C_r$ for the rewards as follows:

$$C_r := \min_{r \in [0,1]^{|X \times A|}} \sum_{t \in [T]} \|\mathbb{E}[r_t] - r\|_1. \tag{1}$$

Intuitively, the corruption $C_r$ encodes the sum over all episodes of the distances between the means $\mathbb{E}[r_t]$ of the adversarial distributions $\mathcal{R}_t$ and a "fictitious" non-corrupted reward vector $r$. Notice that a similar notion of corruption has been employed in unconstrained MDPs to measure the non-stationarity of transition probabilities; see (Jin et al., 2024). In the following, we let $r^\circ \in [0, 1]^{|X \times A|}$ be a reward vector that attains the minimum in the definition of $C_r$. Similarly, we introduce the adversarial corruption $C_G$ for constraint costs, which is defined as follows:

$$C_G := \min_{G \in [0,1]^{|X \times A| \times m}} \sum_{t \in [T]} \max_{i \in [m]} \|\mathbb{E}[g_{t,i}] - g_i\|_1, \tag{2}$$

where $g_i$ is the $i$-th component of $G$. We let $G^\circ \in [0, 1]^{|X \times A| \times m}$ be the constraint cost matrix that attains the minimum in the definition of $C_G$. Finally, we introduce the total adversarial corruption $C$, which is defined as $C := \max\{C_G, C_r\}$.

---

[1] In this paper, we consider w.l.o.g. *loop-free* CMDPs. This means that $X$ is partitioned into $L$ layers $X_0, \ldots, X_L$ such that the first and the last layers are singletons, *i.e.*, $X_0 = \{x_0\}$ and $X_L = \{x_L\}$. Moreover, the loop-free property implies that $P(x'|x, a) > 0$ only if $x' \in X_{k+1}$ and $x \in X_k$ for some $k \in [0 \ldots L - 1]$. Notice that any episodic CMDP with horizon $L$ that is *not* loop-free can be cast into a loop-free one by suitably duplicating the state space $L$ times, *i.e.*, a state $x$ is mapped to a set of new states $(x, k)$, where $k \in [0 \ldots L]$.

[2] In this paper, we denote by $[a \ldots b]$ the set of all the natural numbers from $a \in \mathbb{N}$ to $b \in \mathbb{N}$ (both included), while $[b] := [1 \ldots b]$ is the set of the first $b \in \mathbb{N}$ natural numbers.

Algorithm 1 summarizes how the learner interacts with the environment at episode $t \in [T]$. In particular, the learner chooses a *policy* $\pi : X \times A \to [0, 1]$ at each episode, defining a probability distribution over actions to be employed in each state. For ease of notation, we denote by $\pi(\cdot|x)$ the probability distribution for a state $x \in X$, with $\pi(a|x)$ being the probability of selecting action $a \in A$. Let us remark that we assume that the learner knows $X$ and $A$, but they do *not* know anything about $P$. Moreover, the *feedback* received by the learner after each episode is *bandit*, as they observe the realizations of rewards and constraint costs only for the state-action pairs $(x_k, a_k)$ actually visited during that episode.

---

**Algorithm 1** Learner-Environment Interaction

---
1: $\mathcal{R}_t$ and $\mathcal{G}_t$ are chosen *adversarially*
2: Choose a policy $\pi_t : X \times A \to [0, 1]$
3: Observe initial state $x_0$
4: **for** $k = 0, \ldots, L - 1$ **do**
5:     Play $a_k \sim \pi_t(\cdot|x_k)$
6:     Observe $r_t(x_k, a_k)$ and $g_{t,i}(x_k, a_k)$ for $i \in [m]$
7:     Observe new state $x_{k+1} \sim P(\cdot|x_k, a_k)$

---

## 2.2 OCCUPANCY MEASURES

Next, we introduce *occupancy measures*, following the notation by (Rosenberg & Mansour, 2019a). Given a transition function $P$ and a policy $\pi$, the occupancy measure $q^{P,\pi} \in [0, 1]^{|X \times A \times X|}$ induced by $P$ and $\pi$ is such that, for every $x \in X_k$, $a \in A$, and $x' \in X_{k+1}$ with $k \in [0 \ldots L - 1]$ $q^{P,\pi}(x, a, x') := \mathbb{P}[x_k = x, a_k = a, x_{k+1} = x'|P, \pi]$, which represents the probability that, under $P$ and $\pi$, the learner reaches state $x$, plays action $a$, and gets to the next state $x'$. Moreover, we also define the following quantities $q^{P,\pi}(x, a) := \sum_{x' \in X_{k+1}} q^{P,\pi}(x, a, x')$ and $q^{P,\pi}(x) := \sum_{a \in A} q^{P,\pi}(x, a)$.

The following lemma characterizes when a vector $q \in [0, 1]^{|X \times A \times X|}$ is a *valid* occupancy measure.

**Lemma 1** (Rosenberg & Mansour (2019b)). *A vector* $q \in [0, 1]^{|X \times A \times X|}$ *is a valid occupancy measure of an episodic loop-free CMDP if and only if it satisfies the following conditions:*

$$\begin{cases} \displaystyle\sum_{x \in X_k} \sum_{a \in A} \sum_{x' \in X_{k+1}} q(x, a, x') = 1 & \forall k \in [0 \ldots L - 1] \\ \displaystyle\sum_{a \in A} \sum_{x' \in X_{k+1}} q(x, a, x') = \sum_{x' \in X_{k-1}} \sum_{a \in A} q(x', a, x) & \forall k \in [1 \ldots L - 1], \forall x \in X_k \\ P^q = P, \end{cases}$$

*where $P$ is the transition function of the CMDP and $P^q$ is the one induced by $q$ (see Equation (3)).*

Notice that any valid occupancy measure $q$ induces a transition function $P^q$ and a policy $\pi^q$ as:

$$P^q(x'|x, a) = \frac{q(x, a, x')}{q(x, a)} \quad \text{and} \quad \pi^q(a|x) = \frac{q(x, a)}{q(x)}. \tag{3}$$

## 2.3 PERFORMANCE METRICS TO EVALUATE LEARNING ALGORITHMS

In order to define the performance metrics used to evaluate our *online* learning algorithms, we need to introduce an *offline* optimization problem. Given a CMDP with transition function $P$, we define the following parametric *linear program* (Program (4)), which is parametrized by a reward vector $r \in [0, 1]^{|X \times A|}$, a constraint cost matrix $G \in [0, 1]^{|X \times A| \times m}$ and a threshold vector $\alpha \in [0, L]^m$.

$$\text{OPT}_{r,G,\alpha} := \begin{cases} \max_{q \in \Delta(P)} & r^\top q \quad \text{s.t.} \\ & G^\top q \leq \alpha, \end{cases} \tag{4}$$

where $q \in [0, 1]^{|X \times A|}$ is a vector encoding an occupancy measure, and $\Delta(P)$ is the set of all valid occupancy measures given the transition function $P$ (this set can be encoded by linear constraints thanks to Lemma 1).

We say that an instance of Program (4) satisfies *Slater's condition* if the following holds.

**Condition 1** (Slater). *There exists an occupancy measure* $q^\circ \in \Delta(P)$ *such that* $G^\top q^\circ < \alpha$.

Moreover, we also introduce a problem-specific *feasibility parameter* related to Program (4). This is denoted by $\rho \in [0, L]$ and formally defined as $\rho := \sup_{q \in \Delta(P)} \min_{i \in [m]} \left[ \alpha - G^\top q \right]_i$.[3] Intuitively, $\rho$

---

[3] In this paper, given a vector $y$, we denote by $[y]_i$ its $i$-th component.

represents by how much feasible solutions to Program (4) strictly satisfy the constraints. Notice that Condition 1 is equivalent to say that $\rho > 0$, while, whenever $\rho = 0$, there is no occupancy measure that allows to strictly satisfy the constraints $G^\top q \leq \alpha$ in Program (4).

We are now ready to introduce the notion of *(cumulative) regret* and *positive (cumulative) constraint violation*, which are the performance metrics that we use to evaluate our learning algorithm. In particular, we define the cumulative regret over $T$ episodes as

$$R_T := T \cdot \text{OPT}_{\overline{r}, \overline{G}, \alpha} - \sum_{t \in [T]} \mathbb{E}[r_t]^\top q^{P, \pi_t},$$

where $\overline{r} := \frac{1}{T} \sum_{t=1}^T \mathbb{E}[r_t]$ and $\overline{G} := \frac{1}{T} \sum_{t=1}^T \mathbb{E}[G_t]$. In the following, we denote by $q^*$ an occupancy measure solving Program (4) instantiated with $\overline{r}$, $\overline{G}$, and $\alpha$, while its corresponding policy (computed by Equation (3)) is $\pi^*$. Thus, $\text{OPT}_{\overline{r}, \overline{G}, \alpha} = \overline{r}^\top q^*$ and the regret is $R_T := \sum_{t=1}^T \mathbb{E}[r_t]^\top (q^* - q^{P, \pi_t})$. Furthermore, we define the positive cumulative constraint violation over $T$ episodes as

$$V_T := \max_{i \in [m]} \sum_{t \in [T]} \left[ \mathbb{E}[G_t]^\top q^{P, \pi_t} - \alpha \right]_i^+,$$

where we let $[\cdot]^+ := \max\{0, \cdot\}$. In the following, for ease of notation, we compactly refer to $q^{P, \pi_t}$ as $q_t$, thus omitting the dependency on $P$ and $\pi$.

**Remark 1** (Relation with adversarial/stochastic CMDPs). *Our setting is more akin to CMDPs with adversarial rewards and constraints, rather than stochastic ones. This is because our notion of regret is computed with respect to an optimal constraint-satisfying policy in hindsight that takes into account the average over episodes of the mean values $\mathbb{E}[r_t]$ and $\mathbb{E}[G_t]$ of the adversarially-selected probability distributions $\mathcal{R}_t$ and $\mathcal{G}_t$. This makes our setting much harder than one with stochastic rewards and constraints. Indeed, in the special case in which the supports of $\mathcal{R}_t$ and $\mathcal{G}_t$ are singletons (and, thus, mean values are fully revealed after each episode), our setting reduces to a CMDP with adversarial rewards and constraints, given that such supports are selected adversarially.*

**Remark 2** (Impossibility results carrying over from adversarial CMDPs). *Mannor et al. (2009) show that, in online learning problems with constraints selected adversarially, it is impossible to achieve both regret and constraint violation growing sublinearly in $T$. This result holds for a regret definition that corresponds to ours. Thus, it carries over to our setting. This is why we look for algorithms whose regret and positive constraint violation scale as $\tilde{\mathcal{O}}(\sqrt{T} + C)$, with a linear dependency on the adversarial corruption $C$. Notice that the impossibility result by Mannor et al. (2009) does not rule out the possibility of achieving such a guarantee, since regret and positive constraint violation are not sublinear when $C$ grows linearly in $T$, as it could be the case in a classical adversarial setting.*

## 3   Learning When $C$ is Known: More Optimism is All You Need

We start studying the case in which the learner *knows* the adversarial corruption $C$. We propose an algorithm (called `NS-SOPS`, see also Algorithm 2), which adopts a suitably-designed UCB-like approach encompassing the adversarial corruption $C$ in the confidence bounds of rewards and constraint costs. This effectively results in "boosting" the *optimism* of the algorithm, and it allows to achieve regret and positive constraint violation of the order of $\tilde{\mathcal{O}}(\sqrt{T} + C)$. The `NS-SOPS` algorithm is also a crucial building block in the design of our algorithm for the case in which the adversarial corruption $C$ is *not* known, as we show in the following section.

### 3.1   `NS-SOPS`: Non-Stationary Safe Optimistic Policy Search

Algorithm 2 provides the pseudocode of the *non-stationary safe optimistic policy search* (`NS-SOPS` for short) algorithm. The algorithm keeps track of suitably-defined confidence bounds for transition probabilities, rewards, and constraint costs. At each episode $t \in [T]$, the algorithm builds a confidence set $\mathcal{P}_t$ for the transition function $P$ by following the same approach as Jin et al. (2020) (see Appendix G for its definition). Instead, for rewards and constraint costs, the algorithm adopts novel *enlarged* confidence bounds, which are suitably designed to tackle non-stationarity. Given $\delta \in (0, 1)$, by letting $N_t(x, a)$ be the total number of visits to the state-action pair $(x, a) \in X \times A$ up to episode $t$ (excluded), the confidence bound for the reward $r_t(x, a)$ is $\phi_t(x, a) :=$

$\min\left\{1, \sqrt{\frac{\ln(2T|X||A|/\delta)}{2\max\{N_t(x,a),1\}}} + \frac{C}{\max\{N_t(x,a),1\}} + \frac{C}{T}\right\}$, while the confidence bound for the constraint

costs $g_{t,i}(x,a)$ is defined as $\xi_t(x,a) \coloneqq \min\left\{1, \sqrt{\frac{\ln(2mT|X||A|/\delta)}{2\max\{N_t(x,a),1\}}} + \frac{C}{\max\{N_t(x,a),1\}} + \frac{C}{T}\right\}$. Intu-

itively, the first term in the expressions above is derived from Azuma-Hoeffding inequality, the second term allows to deal with the non-stationarity of rewards and constraint costs, while the third term is needed to bound how much the average reward vector $\overline{r}$ and the average constraint costs $[\overline{G}]_i$ differ from their "fictitious" non-corrupted counterparts $r^\circ$ and $[G^\circ]_i$, respectively.

Algorithm 2 also computes empirical rewards and constraint costs. At each episode $t \in [T]$, for any state-action pair $(x,a) \in X \times A$ and constraint $i \in [m]$, these are defined as $\widehat{r}_t(x,a) \coloneqq \frac{\sum_{\tau \in [t]} \mathbb{I}_\tau(x,a) r_\tau(x,a)}{\max\{N_t(x,a),1\}}$ and $\widehat{g}_{t,i}(x,a) \coloneqq \frac{\sum_{\tau \in [t]} \mathbb{I}_\tau(x,a) g_{\tau,i}(x,a)}{\max\{N_t(x,a),1\}}$, where $\mathbb{I}_\tau(x,a) = 1$ if and only if $(x,a)$ is visited during episode $\tau$, while $\mathbb{I}_\tau(x,a) = 0$ otherwise. For ease of notation, we let $\widehat{G}_t \in [0,1]^{|X \times A| \times m}$ be the matrix with components $\widehat{g}_{t,i}(x,a)$. We refer the reader to Appendix C for all the technical results related to confidence bounds.

Algorithm 2 selects policies with an UCB-like approach encompassing *optimism* in both rewards and constraints satisfaction, following an approach similar to that employed by Efroni et al. (2020). Specifically, at each episode $t \in [T]$ and for any state-action pair $(x,a) \in X \times A$, the algorithm employs an *upper* confidence bound for the reward $r_t(x,a)$, defined as $\overline{r}_t(x,a) \coloneqq \widehat{r}_t(x,a) + \phi_t(x,a)$, while it uses *lower* confidence bounds for the constraint costs $g_{t,i}(x,a)$, defined as $\underline{g}_{t,i}(x,a) \coloneqq \widehat{g}_{t,i}(x,a) - \xi_t(x,a)$ for every constraint $i \in [m]$. Then, by letting $\overline{r}_t \in [0,1]^{|X \times A|}$ be the vector with components $\overline{r}_t(x,a)$ and $\underline{G}_t$ be the matrix with entries $\underline{g}_{t,i}(x,a)$, Algorithm 2 chooses the policy

---

**Algorithm 2** NS−SOPS

**Require:** $C, \delta \in (0,1)$
1: $\pi_1 \leftarrow$ select any policy
2: **for** $t \in [T]$ **do**
3:     Choose policy $\pi_t$ in Algorithm 1 and observe feedback from interaction
4:     Compute $\mathcal{P}_t$, $\overline{r}_t$, and $\underline{G}_t$
5:     $q \leftarrow$ solution to $\text{OPT-CB}_{\Delta(\mathcal{P}_t),\overline{r}_t,\underline{G}_t,\alpha}$
6:     **if** problem is *feasible* **then**
7:         $\widehat{q}_{t+1} \leftarrow q$
8:     **else**
9:         $\widehat{q}_{t+1} \leftarrow$ take any $q \in \Delta(\mathcal{P}_t)$
10:     $\pi_{t+1} \leftarrow \pi^{\widehat{q}_{t+1}}$

---

to be employed in the next episode $t+1$ by solving the following linear program:

$$\text{OPT-CB}_{\Delta(\mathcal{P}_t),\overline{r}_t,\underline{G}_t,\alpha} \coloneqq \begin{cases} \arg\max_{q \in \Delta(\mathcal{P}_t)} & \overline{r}_t^\top q \quad \text{s.t.} \\ & \underline{G}_t^\top q \leq \alpha, \end{cases} \tag{5}$$

where $\Delta(\mathcal{P}_t)$ is the set of all the possible valid occupancy measures given the confidence set $\mathcal{P}_t$ (see Appendix G). If $\text{OPT-CB}_{\Delta(\mathcal{P}_t),\overline{r}_t,\underline{G}_t,\alpha}$ is feasible, its solution is used to compute a policy to be employed in the next episode, otherwise the algorithm uses any occupancy measure in the set $\Delta(\mathcal{P}_t)$.

### 3.2 THEORETICAL GUARANTEES OF NS−SOPS

Next, we prove the theoretical guarantees attained by Algorithm 2 (see Appendix D for complete proofs of the theorems and associated lemmas). First, we analyze the positive cumulative violation incurred by the algorithm. Formally, we can state the following result.

**Theorem 2.** *Given any $\delta \in (0,1)$, with probability at least $1 - 8\delta$, Algorithm 2 attains violations* $V_T = \mathcal{O}\left(L|X|\sqrt{|A|T \ln\left(mT|X||A|/\delta\right)} + \ln(T)|X||A|C\right)$.

Intuitively, Theorem 2 is proved by showing that every constraint-satisfying occupancy measure is also feasible for Program (5) with high probability. This holds since Program (5) employs lower confidence bounds for constraint costs. Thus, in order to bound $V_T$, it is sufficient to analyze at which rate the feasible region of Program (5) concentrates to the *true* one (*i.e.*, the one defined by $\overline{G}$ in Program (4)). Since by definition of $\xi_t(x,a)$ the feasibility region of Program (5) concentrates as $1/\sqrt{t} + C/t$, the resulting bound for the positive constraint violation $V_T$ is of the order of $\tilde{\mathcal{O}}(\sqrt{T}+C)$.

The regret guaranteed by Algorithm 2 is formalized by the following theorem.

**Theorem 3.** *Given any $\delta \in (0,1)$, with probability at least $1 - 9\delta$, Algorithm 2 attains regret* $R_T = \mathcal{O}\left(L|X|\sqrt{|A|T \ln\left(T|X||A|/\delta\right)} + \ln(T)|X||A|C\right)$.

Theorem 3 is proved similarly to Theorem 2. Indeed, since every constraint-satisfying occupancy measure is feasible for Program (5) with high probability, this also holds for $q^*$, as it satisfies constraints by definition. Thus, since by definition of $\phi_t(x, a)$ the upper confidence bound for the rewards maximized by Program (5) concentrates as $1/\sqrt{t} + C/t$, the regret bound follows.

**Remark 3** (What if some under/overestimate of $C$ is available). *We also study what happens if the learner runs Algorithm 2 with an under/overestimate on the adversarial corruption as input. We defer to Appendix E all the technical results related to this analysis. In particular, it is possible to show that any underestimate on $C$ does not detriment the bound on $V_T$, which remains the one in Theorem 2. On the other hand, an overestimate on $C$, say $\widehat{C} > C$, results in a bound on $V_T$ of the order of $\mathcal{O}(\sqrt{T} + \widehat{C})$, which is worse than the one in Theorem 2. Intuitively, this is because using an overestimate makes Algorithm 2 too conservative. As a result, one could be tempted to conclude that running Algorithm 2 with an underestimate of $C$ as input is satisfactory when the true value of $C$ is unknown. However, this would lead to a regret $R_T$ growing linearly in $T$, since, intuitively, a regret-minimizing policy could be cut off from the algorithm decision space. This motivates the introduction of additional tools to deal with the case in which $C$ is unknown, as we do in Section 4.*

# 4 LEARNING WHEN $C$ IS *Not* KNOWN: A LAGRANGIFIED META-PROCEDURE

In this section, we go beyond Section 3 by studying the more relevant case in which the learner does *not* know the value of the adversarial corruption $C$. In order to tackle this challenging scenario, we develop a *meta-procedure* (called Lag-FTRL, see Algorithm 3) that instantiates multiple instances of an algorithm working for the case in which $C$ is known, with each instance taking care of a different "guess" on the value of $C$. The Lag-FTRL algorithm is inspired by the work of Agarwal et al. (2017) in the context of classical (unconstrained) multi-armed bandit problems. Let us remark that Lag-FTRL is a general algorithm that is *not* specifically tailored for our non-stationary CMDP setting. Indeed, it could be applied to any non-stationary online learn-

---

**Algorithm 3** Lag-FTRL

**Require:** $\delta \in (0, 1)$
1: $\Lambda \leftarrow \frac{Lm+1}{\rho}$, $M \leftarrow \lceil \log_2 T \rceil$
2: $\gamma \leftarrow \sqrt{\ln(M/\delta)/TM}$, $\eta \leftarrow \frac{1}{2\Lambda m \left(\sqrt{\beta_1 T} + \beta_2 + \beta_5 + \sqrt{\beta_4 T}\right)}$
3: **for** $j \in [M]$ **do**
4: $\quad$ Alg$^j \leftarrow$ stabilized Algorithm 2 with $C = 2^j$
5: $w_{1,j} \leftarrow 1/M$ for all $j \in [M]$
6: **for** $t \in [T]$ **do**
7: $\quad$ Sample index $j_t \sim w_t$
8: $\quad$ $\pi_t^{j_t} \leftarrow$ policy that Alg$^{j_t}$ would choose
9: $\quad$ Choose policy $\pi_t^{j_t}$ in Algorithm 1 and observe feedback from interaction
10: $\quad$ Let Alg$^{j_t}$ observe received feedback
11: $\quad$ **for** $j \in [M]$ **do**
12: $\quad\quad$ Build $\ell_{t,j}$ as in Equation (6)
13: $\quad\quad$ Build $b_{t,j}$ as in Equation (7)
14: $\quad$ $w_{t+1} \leftarrow \underset{\substack{w \in \Delta_M, \\ w_j \geq 1/T}}{\arg\min} \; w^\top \sum_{\tau \in [t]} (\ell_t - b_t) + \frac{1}{\eta} \sum_{j \in [M]} \ln \frac{1}{w_j}$

---

ing problem with constraints when the adversarial corruption $C$ is unknown, provided that an algorithm for known $C$ is available. In this section, to deal with our non-stationary CMDP setting, we let Lag-FTRL instantiate multiple instances of the NS-SOPS algorithm developed in Section 3.

## 4.1 LAG-FTRL: LAGRANGIFIED FTRL

At a high level, the *Lagrangified follow-the-regularized-leader* (Lag-FTRL for short) algorithm works by instantiating several instances of Algorithm 2, suitably stabilized (see section H), with each instance Alg$^j$ being run for a different "guess" of the (unknown) adversarial corruption value $C$. The algorithm plays the role of a *master* by choosing which instance Alg$^j$ to use at each episode. The selection is done by employing an FTRL approach with a suitable log-barrier regularization. In particular, at each episode $t \in [T]$, by letting Alg$^{j_t}$ be the selected instance, the Lag-FTRL algorithm employs the policy $\pi_t^{j_i}$ prescribed by Alg$^{j_t}$ and provides feedback to instance Alg$^{j_t}$ only. The Lag-FTRL algorithm faces two main challenges. First, the feedback available to the FTRL procedure implemented at the master level is *partial*. This is because, at each episode $t \in [T]$, the algorithm only observes the result of using the policy $\pi_t^{j_i}$ prescribed by the chosen instance Alg$^{j_t}$,

and *not* those of the policies suggested by other instances. The algorithm tackles this challenge by employing *optimistic loss estimators* in the FTRL selection procedure, following an approach originally introduced by Neu (2015). The second challenge originates from the fact that the goal of the algorithm is to keep under control both the regret and the positive constraint violation. This is accomplished by feeding the FTRL procedure with losses constructed starting from the Lagrangian of the offline optimization problem in Program (4), and suitably modified to manage *positive* violations.

The pseudocode of the Lag-FTRL algorithm is provided in Algorithm 3. At Line 4, it instantiates $M := \lceil \log_2 T \rceil$ instances of Algorithm 2, with each instance $\text{Alg}^j$, for $j \in [M]$, receiving as input a "guess" on the adversarial corruption $C = 2^j$. Notice that, to every instance of Algorithm 2, a standard doubling trick and a stabilization procedure is applied (see Algorithm 4 for additional details). This modification to Algorithm 2 is necessary to guarantee that each instance $j$ attains a regret and positive cumulative constraints violation which linearly degrade in $\nu_{T,j} = 1/\min_{t\in[T]} w_{t,j}$ and $C$, when employed by the master algorithm. The algorithm assigns weights defining a probability distribution to instances $\text{Alg}^j$, with $w_{t,j} \in [0,1]$ denoting the weight of instance $\text{Alg}^j$ at episode $t \in [T]$. We denote by $w_t \in \Delta_M$ the weight vector at episode $t$, with $\Delta_M$ being the $M$-dimensional simplex. At the first episode, all the weights $w_{1,j}$ are initialized to the value $1/M$ (Line 5). Then, at each episode $t \in [T]$, the algorithm samples an instance index $j_t \in [M]$ according to the probability distribution defined by the weight vector $w_t$ (Line 7), and it employs the policy $\pi_t^{j_t}$ prescribed by $\text{Alg}^{j_t}$ (Line 8). The algorithm observes the feedback from the interaction described in Algorithm 1 and it sends such a feedback to instance $\text{Alg}^{j_t}$ (Line 10). Then, at Line 12, the algorithm builds an *optimistic* loss estimator to be fed into each instance $\text{Alg}^j$. In particular, at episode $t \in [T]$ and for every $j \in [M]$, the optimistic loss estimator is defined as:

$$\ell_{t,j} := \frac{\mathbb{I}(j_t = j)}{w_{t,j} + \gamma} \left( L - \sum_{k \in [0...L-1]} r_t(x_k^t, a_k^t) + \Lambda \sum_{i \in [m]} \left[ \left(\widehat{G}_t^j\right)^\top \widehat{q}_t^j - \alpha \right]_i^+ \right), \qquad (6)$$

where $\gamma$ is a suitably-defined implicit exploration factor, $(x_k^t, a_k^t)$ is the state-action pair visited at layer $k$ during episode $t$, $\Lambda$ is a suitably-defined upper bound on the optimal values of Lagrangian multipliers,[4] $\widehat{G}_t^j$ is the matrix of empirical constraint costs built by the instance $\text{Alg}^j$ of Algorithm 2 at episode $t$, while $\widehat{q}_t^j$ is the occupancy measure computed by instance $\text{Alg}^j$ of Algorithm 2 at $t$. Finally, the algorithm updates the weight vector according to an FTRL update on a cut decision space with a suitable log-barrier regularization and a bonus term $b_t$ defined as:

$$b_{t,j} := \left( (m\Lambda\beta_5 + \beta_2) + \left(\sqrt{\beta_1} + m\Lambda\sqrt{\beta_4}\right)\sqrt{T} \right)(\nu_{t,j} - \nu_{t-1,j}), , \qquad (7)$$

where $\nu_{t,j} = \max_{\tau \le t} \frac{1}{w_{\tau,j}}$ and the parameters $\beta$ are linked to the performance of Algorithm 2 (see Line 13 and Section F.2.1 for additional details). See Line 14 for the complete definition of the update. The bonus term purpose is to balance out the term related to the difference between the performance of Algorithm 2 updated at each episode and the performance of its stabilized version, which works under the condition imposed by the master algorithm.

## 4.2 THEORETICAL GUARANTEES OF LAG-FTRL

Next, we prove the theoretical guarantees attained by Algorithm 3 (see Appendix F for complete proofs of the theorems and associated lemmas). As a first preliminary step, we extend the well-known strong duality result for CMDPs (Altman, 1999) to the case of bounded Lagrangian multipliers.

**Lemma 2.** *Given a CMDP with a transition function $P$, for every reward vector $r \in [0,1]^{|X \times A|}$, constraint cost matrix $G \in [0,1]^{|X \times A| \times m}$, and threshold vector $\alpha \in [0,L]^m$, if Program (4) satisfies Slater's condition (Condition 1), then the following holds:*

$$\min_{\|\lambda\|_1 \in [0, L/\rho]} \max_{q \in \Delta(P)} r^\top q - \sum_{i \in [m]} \lambda_i \left[ G^\top q - \alpha \right]_i = \max_{q \in \Delta(P)} \min_{\|\lambda\|_1 \in [0, L/\rho]} r^\top q - \sum_{i \in [m]} \lambda_i \left[ G^\top q - \alpha \right]_i$$

$$= \text{OPT}_{r,G,\alpha},$$

*where $\lambda \in \mathbb{R}_{\ge 0}^m$ is a vector of Lagrangian multipliers and $\rho$ is the feasibility parameter of Program (4).*

---

[4]Notice that, in the definition of $\Lambda$, $\rho$ is the feasibility parameter of Program (4) for the reward vector $\overline{r}$, the constraint cost matrix $\overline{G}$, and the threshold vector $\alpha$. In order to compute $\Lambda$, Algorithm 3 needs knowledge of $\rho$. Nevertheless, our results continue to hold even if Algorithm 3 is only given access to a lower bound on $\rho$.

Intuitively, Lemma 2 states that, under Slater's condition, strong duality continues to hold even when restricting the set of Lagrangian multipliers to the $\lambda \in \mathbb{R}^m_{\geq 0}$ having $\|\lambda\|_1$ bounded by $L/\rho$. Furthermore, we extend the result in Lemma 2 to the case of a Lagrangian function suitably-modified to encompass *positive* violations. We call it *positive Lagrangian* of Program (4), defined as follows.

**Definition 1** (Positive Lagrangian). *Given a CMDP with a transition function $P$, for every reward vector $r \in [0,1]^{|X \times A|}$, constraint cost matrix $G \in [0,1]^{|X \times A| \times m}$, and threshold vector $\alpha \in [0,L]^m$, the* positive Lagrangian *of Program (4) is defined as a function $\mathcal{L} : \mathbb{R}_+ \times \Delta(P) \to \mathbb{R}$ such that it holds $\mathcal{L}(\beta, q) := r^\top q - \beta \sum_{i \in [m]} [G^\top q - \alpha]^+_i$ for every $\beta \geq 0$ and $q \in \Delta(P)$.*

The positive Lagrangian is related to the Lagrangian of a variation of Program (4) in which the $[\cdot]^+$ operator is applied to the constraints. Notice that such a problem does *not* admit Slater's condition, since, by definition of $[\cdot]^+$, it does *not* exist an occupancy measure $q^\circ$ such that $[G^\top q^\circ - \alpha]^+_i < 0$ for every $i \in [m]$. Nevertheless, we show that a kind of strong duality result still holds for $\mathcal{L}(L/\rho, q)$, when Slater's condition is met by Program (4). This is done in the following result.

**Theorem 4.** *Given a CMDP with a transition function $P$, for every reward vector $r \in [0,1]^{|X \times A|}$, constraint cost matrix $G \in [0,1]^{|X \times A| \times m}$, and threshold vector $\alpha \in [0,L]^m$, if Program (4) satisfies Slater's condition (Condition 1), then the following holds:*

$$\max_{q \in \Delta(P)} \mathcal{L}(L/\rho, q) = \max_{q \in \Delta(P)} r^\top q - \frac{L}{\rho} \sum_{i \in [m]} [G^\top q - \alpha]^+_i = \mathrm{OPT}_{r,G,\alpha},$$

*where $\rho$ is the feasibility parameter of Program (4).*

Theorem 4 intuitively shows that a $L/\rho$ multiplicative factor on the positive constraint violation is enough to compensate the large rewards that non-feasible policies would attain when employed by the learner. This result is crucial since, without properly defining the Lagrangian function optimized by Algorithm 3, the FTRL optimization procedure would choose instances with both large rewards and large constraint violation, thus preventing the violation bound from being sublinear. By means of Theorem 4, it is possible to provide the following result.

**Theorem 5.** *If Program (4) instantiated with $\bar{r}$, $\overline{G}$ and $\alpha$ satisfies Slater's condition (Condition 1), then, given any $\delta \in (0,1)$, with probability at least $1 - 34\delta$, Algorithm 3 attains violation $V_T = \mathcal{O}(m^2 L^2 |X| \sqrt{|A|T \log(mT|X||A|/\delta)} \log(T)^2 + m^2 L |X|^2 |A|^2 \log(T)^3 \log(\log(T)/\delta) + m^2 L \log(T)^2 |X||A|C)$.*

Intuitively, to prove Theorem 5, it is necessary to bound the negative regret attained by the algorithm, *i.e.*, how better Algorithm 3 can perform in terms of rewards with respect to an optimal occupancy in hindsight $q^*$. Notice that this is equivalent to showing that the FTRL procedure cannot gain more than $\mathrm{OPT}_{\bar{r},\overline{G},\alpha}$ by playing policies that are *not* feasible, or, equivalently, by choosing instances $\mathtt{Alg}^j$ with a large corruption guess, which, by definition of the confidence sets employed by Algorithm 2, may play non-feasible policies attaining large rewards. This is done by employing Theorem 4, which shows that the positive Lagrangian does *not* allow the algorithm to achieve too large rewards with respect to $q^*$. Thus, the violations are still upper bounded by $\tilde{\mathcal{O}}(\sqrt{T} + C)$. Finally, we prove the regret bound attained by Algorithm 3.

**Theorem 6.** *If Program (4) instantiated with $\bar{r}$, $\overline{G}$ and $\alpha$ satisfies Slater's condition (Condition 1), then, given any $\delta \in (0,1)$, with probability at least $1 - 30\delta$, Algorithm 3 attains regret $R_T = \mathcal{O}(m^2 L^2 |X| \sqrt{|A|T \log(mT|X||A|/\delta)} \log(T)^2 + m^2 L |X|^2 |A|^2 \log(T)^3 \log(\log(T)/\delta) + m^2 L \log(T)^2 |X||A|C)$.*

Bounding the regret attained by Algorithm 3 requires different techniques with respect to bounding constraint violation. Indeed, strong duality is *not* needed, since, even if $\Lambda$ is set to a too small value and thus the algorithm plays non-feasible policies, then the regret would still be sublinear. The regret bound is strongly related to the optimal value of the problem associated with the positive Lagrangian, which, by definition of $[\cdot]^+$ cannot perform worse than the optimum of Program (4), in terms of rewards gained. Thus, by letting $j^*$ be the index of the instance associated with true corruption value $C$, proving Theorem 6 reduces to bounding the regret and the constraint violation of instance $\mathtt{Alg}^{j^*}$, with the additional challenge of bounding the estimation error of the optimistic loss estimator. Finally, by means of the results for the *known $C$* case derived in Section 3, we are able to show that the regret is at most $\tilde{\mathcal{O}}(\sqrt{T} + C)$, which is the desired bound.

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

APPENDIX

The appendix is structured as follows:

- In Appendix A we provide the complete related works.
- In Appendix B we provide the events dictionary.
- In Appendix C we provide the preliminary results on the confidence sets employed to estimate the unknown parameters of the environment.
- In Appendix D we provide the omitted proofs related to the theoretical guarantees when the corruption value is known by the learner, namely, the results attained by Algorithm 2.
- In Appendix E we provide the omitted proofs of the theoretical guarantees attained by Algorithm 2, when a guess on the corruption is given as input to the algorithm.
- In Appendix F we provide the omitted proofs related to the theoretical guarantees when the corruption value is *not* known by the learner, namely, the results attained by Algorithm 3.
- In Appendix G we restate useful results from existing works.
- In Appendix H we provide the results related to stability a corruption-robustness.

## A  RELATED WORKS

In the following, we discuss some works that are tightly related to ours. In particular, we first describe works dealing with the online learning problem in MDPs, and, then, we discuss some works studying the constrained version of the classical online learning problem.

**Online learning in MDPs**  The literature on online learning problems (Cesa-Bianchi & Lugosi, 2006) in MDPs is wide (see (Auer et al., 2008; Even-Dar et al., 2009; Neu et al., 2010) for some initial results on the topic). In such settings, two types of feedback are usually studied: in the *full-information feedback* model, the entire loss function is observed after the learner's choice, while in the *bandit feedback* model, the learner only observes the loss due to the chosen action. Azar et al. (2017) study the problem of optimal exploration in episodic MDPs with unknown transitions and stochastic losses when the feedback is bandit. The authors present an algorithm whose regret upper bound is $\tilde{\mathcal{O}}(\sqrt{T})$, thus matching the lower bound for this class of MDPs and improving the previous result by Auer et al. (2008).

**Online learning in non-stationary MDPs**  The literature on non-stationary MDPs encompasses both works on non-stationary rewards and non-stationary transitions. As concerns the first research line, Rosenberg & Mansour (2019b) study the online learning problem in episodic MDPs with adversarial losses and unknown transitions when the feedback is full information. The authors present an online algorithm exploiting entropic regularization and providing a regret upper bound of $\tilde{\mathcal{O}}(\sqrt{T})$. The same setting is investigated by Rosenberg & Mansour (2019a) when the feedback is bandit. In such a case, the authors provide a regret upper bound of the order of $\tilde{\mathcal{O}}(T^{3/4})$, which is improved by Jin et al. (2020) by providing an algorithm that achieves in the same setting a regret upper bound of $\tilde{\mathcal{O}}(\sqrt{T})$. Related to the non-stationarity of the transitions , Wei et al. (2022) study MDPs with adversarial corruption on transition functions and rewards, reaching a regret upper bound of order $\widetilde{\mathcal{O}}(\sqrt{T} + C)$ (where $C$ is the amount of adversarial corruption) with respect to the optimal policy of the non-corrupted MDP . Finally, Jin et al. (2024) is the first to study completely adversarial MDPs with changing transition functions, providing a $\tilde{\mathcal{O}}(\sqrt{T} + C)$ regret bounds, where $C$ is a corruption measure of the adversarially changing transition functions.

**Online learning with constraints**  A central result is provided by Mannor et al. (2009), who show that it is impossible to suffer from sublinear regret and sublinear constraint violation when an adversary chooses losses and constraints. Liakopoulos et al. (2019) try to overcome such an impossibility result by defining a new notion of regret. They study a class of online learning problems with long-term budget constraints that can be chosen by an adversary. The learner's regret metric is modified by introducing the notion of a *K-benchmark*, *i.e.*, a comparator that meets the problem's

allotted budget over any window of length $K$. Castiglioni et al. (2022a;b) deal with the problem of online learning with stochastic and adversarial losses, providing the first *best-of-both-worlds* algorithm for online learning problems with long-term constraints.

**Online learning in CMDPs** Online Learning In MDPs with constraints is generally studied when the constraints are selected stochastically. Precisely, Zheng & Ratliff (2020) deal with episodic CMDPs with stochastic losses and constraints, where the transition probabilities are known and the feedback is bandit. The regret upper bound of their algorithm is of the order of $\tilde{\mathcal{O}}(T^{3/4})$, while the cumulative constraint violation is guaranteed to be below a threshold with a given probability. Wei et al. (2018) deal with adversarial losses and stochastic constraints, assuming the transition probabilities are known and the feedback is full information. The authors present an algorithm that guarantees an upper bound of the order of $\tilde{\mathcal{O}}(\sqrt{T})$ on both regret and constraint violation. Bai et al. (2020) provide the first algorithm that achieves sublinear regret when the transition probabilities are unknown, assuming that the rewards are deterministic and the constraints are stochastic with a particular structure. Efroni et al. (2020) propose two approaches to deal with the exploration-exploitation dilemma in episodic CMDPs. These approaches guarantee sublinear regret and constraint violation when transition probabilities, rewards, and constraints are unknown and stochastic, while the feedback is bandit. Qiu et al. (2020) provide a primal-dual approach based on *optimism in the face of uncertainty*. This work shows the effectiveness of such an approach when dealing with episodic CMDPs with adversarial losses and stochastic constraints, achieving both sublinear regret and constraint violation with full-information feedback. Stradi et al. (2024a) is the first work to tackle CMDPs with adversarial losses and bandit feedback. They propose an algorithm which achieves sublinear regret and sublinear positive constraints violations, assuming that the constraints are stochastic. Stradi et al. (2024b) are the first to study CMDPs with adversarial constraints. Given the well-known impossibility result to learn with adversarial constraints, they propose an algorithm that attains sublinear violation (with cancellations allowed) and a fraction of the optimal reward when the feedback is full. Finally, Ding & Lavaei (2023) and Wei et al. (2023) consider the case in which rewards and constraints are non-stationary, assuming that their variation is bounded, as in our work. Nevertheless, our settings differ in multiple aspects. First of all, we consider positive constraints violations, while the aforementioned works allow the cancellations in their definition. We consider a static regret adversarial baseline, while Ding & Lavaei (2023) and Wei et al. (2023) consider the stronger baseline of dynamic regret. Nevertheless, our bounds are not comparable, since we achieve linear regret and violations only in the worst case scenario in which $C = T$, while a sublinear corruption would lead to linear dynamic regret in their work. Finally, we do not make any assumption on the number of episodes, while both the regret and violations bounds presented in Wei et al. (2023) hold only for large $T$.

## B  EVENTS DICTIONARY

In the following, we introduce the main events which are related to estimation of the unknown stochastic parameters of the environment.

- **Event $\mathcal{E}_P$**: for all $t \in [T], P \in \mathcal{P}_t$. $\mathcal{E}_P$ holds with probability at least $1 - 4\delta$ by Lemma 19. The event is related to the estimation of the unknown transition function.

- **Event $\mathcal{E}_G$**: for all $t \in [T], i \in [m], (x, a) \in X \times A$:

$$\left| \widehat{g}_{t,i}(x,a) - \frac{1}{T} \sum_{\tau \in [T]} \mathbb{E}[g_{\tau,i}(x,a)] \right| \leq \xi_t(x,a).$$

  Similarly,

$$\left| \widehat{g}_{t,i}(x,a) - g_i^\circ(x,a) \right| \leq \xi_t(x,a),$$

  where $g_i^\circ \in [0,1]^{|X \times A|} := [G^\circ]_i$.

  $\mathcal{E}_G$ holds with probability at least $1 - \delta$ by Corollary 2. The event is related to the estimation of the unknown constraint functions.

- **Event $\mathcal{E}_r$**: for all $t \in [T], (x,a) \in X \times A$:

$$\left| \widehat{r}_t(x,a) - \frac{1}{T} \sum_{\tau \in [T]} \mathbb{E}[r_\tau(x,a)] \right| \leq \phi_t(x,a).$$

Similarly,

$$\left| \widehat{r}_t(x,a) - r^\circ(x,a) \right| \leq \phi_t(x,a).$$

$\mathcal{E}_r$ holds with probability at least $1 - \delta$ by Corollary 4. The event is related to the estimation of the unknown reward function.

- **Event $\mathcal{E}_{\widehat{q}}$**: for any $P_t^x \in \mathcal{P}_t$:

$$\sum_{t \in [T]} \sum_{x \in X, a \in A} \left| q^{P_t^x, \pi_t}(x,a) - q_t(x,a) \right| \leq \mathcal{O}\left( L|X| \sqrt{|A|T \ln\left( \frac{T|X||A|}{\delta} \right)} \right).$$

$\mathcal{E}_{\widehat{q}}$ holds with probability at least $1 - 6\delta$ by Lemma 20. The event is related to the convergence to the true unknown occupancy measure. Notice that $\mathbb{P}\left[ \mathcal{E}_{\widehat{q}} \cap \mathcal{E}_P \right] \geq 1 - 6\delta$ by construction.

## C  CONFIDENCE INTERVALS

In this section we will provide the preliminary results related to the high probability confidence sets for the estimation of the cost constraints matrices and the reward vectors.

We start bounding the distance between the *non-corrupted* costs and rewards with respect to the mean of the adversarial distributions.

**Lemma 3.** *For all $i \in [m]$, fixing $(x,a) \in X \times A$, it holds:*

$$\left| g_i^\circ(x,a) - \frac{1}{T} \sum_{t \in [T]} \mathbb{E}[g_{t,i}(x,a)] \right| \leq \frac{C_G}{T}.$$

*Similarly, fixing $(x,a) \in X \times A$, it holds:*

$$\left| r^\circ(x,a) - \frac{1}{T} \sum_{t \in [T]} \mathbb{E}[r_t(x,a)] \right| \leq \frac{C_r}{T},$$

*Proof.* By triangle inequality and from the definition of $C_G$, it holds:

$$\left| g_i^\circ(x,a) - \frac{1}{T} \sum_{t \in [T]} \mathbb{E}[g_{t,i}(x,a)] \right| = \left| \frac{1}{T} \sum_{t \in [T]} (g_i^\circ(x,a) - \mathbb{E}[g_{t,i}(x,a)]) \right|$$

$$\leq \frac{1}{T} \sum_{t \in [T]} \left| g_i^\circ(x,a) - \mathbb{E}[g_{t,i}(x,a)] \right|$$

$$\leq \frac{C_G}{T}.$$

Notice that the proof holds for all $i \in [m]$ since $C_G$ is defined employing the maximum over $i \in [m]$. Following the same steps, it holds:

$$\left| r^\circ(x,a) - \frac{1}{T} \sum_{t \in [T]} \mathbb{E}[r_t(x,a)] \right| = \left| \frac{1}{T} \sum_{t \in [T]} (r^\circ(x,a) - \mathbb{E}[r_t(x,a)]) \right|$$

$$\leq \frac{1}{T} \sum_{t \in [T]} \left| r^\circ(x,a) - \mathbb{E}[r_t(x,a)] \right|$$

$$\leq \frac{C_r}{T},$$

which concludes the proof. $\square$

In the following lemma, we bound the distance between the empirical mean of the constraints function and the true *non-corrupted* value.

**Lemma 4.** *Fixing $i \in [m]$, $(x, a) \in X \times A$, $t \in [T]$, for any $\delta \in (0, 1)$, it holds with probability at least $1 - \delta$:*

$$\left| \widehat{g}_{t,i}(x, a) - g_i^\circ(x, a) \right| \leq \sqrt{\frac{1}{2 \max\{N_t(x, a), 1\}} \ln\left(\frac{2}{\delta}\right)} + \frac{C_G}{\max\{N_t(x, a), 1\}}.$$

*Proof.* We start bounding the quantity of interest as follows:

$$\left| \widehat{g}_{t,i}(x, a) - g_i^\circ(x, a) \right| = \left| \left( \frac{\sum_{\tau \in [t]} \mathbb{I}_\tau(x, a) g_{\tau,i}(x, a)}{\max\{N_t(x, a), 1\}} \right) - g_i^\circ(x, a) \right|$$

$$\leq \left| \frac{1}{\max\{N_t(x, a), 1\}} \sum_{\tau \in [t]} \mathbb{I}_\tau(x, a) \left( g_{\tau,i}(x, a) - \mathbb{E}[g_{\tau,i}(x, a)] \right) \right|$$

$$+ \left| \frac{1}{\max\{N_t(x, a), 1\}} \sum_{\tau \in [t]} \mathbb{I}_\tau(x, a) [\mathbb{E}[g_{\tau,i}(x, a)] - g_i^\circ(x, a)] \right|, \quad (8)$$

where we employed the triangle inequality and the definition of $\widehat{g}_{t,i}(x, a)$.

We bound the two terms in Equation (8) separately. For the first term, by Hoeffding's inequality and noticing that constraints values are bounded in $[0, 1]$, it holds that:

$$\mathbb{P}\left[ \mathcal{A} \geq \frac{c}{\max\{N_t(x, a), 1\}} \right] \leq 2 \exp\left( - \frac{2c^2}{\max\{N_t(x, a), 1\}} \right),$$

where,

$$\mathcal{A} = \left| \left( \frac{\sum_{\tau \in [t]} \mathbb{I}_\tau(x, a) g_{\tau,i}(x, a)}{\max\{N_t(x, a), 1\}} \right) - \left( \frac{\sum_{\tau \in [t]} \mathbb{I}_\tau(x, a) \mathbb{E}[g_{\tau,i}(x, a)]}{\max\{N_t(x, a), 1\}} \right) \right|,$$

Setting $\delta = 2 \exp\left( -\frac{2c^2}{\max\{N_t(x,a),1\}} \right)$ and solving to find a proper value of $c$ we get that with probability at least $1 - \delta$:

$$\left| \frac{1}{\max\{N_t(x, a), 1\}} \sum_{\tau \in [t]} \mathbb{I}_\tau(x, a) \left( g_{\tau,i}(x, a) - \mathbb{E}[g_{\tau,i}(x, a)] \right) \right| \leq \sqrt{\frac{1}{2 \max\{N_t(x, a), 1\}} \ln\left(\frac{2}{\delta}\right)}.$$

Finally, we focus on the second term. Thus, employing the triangle inequality and the definition of $C_G$, it holds:

$$\left| \frac{1}{\max\{N_t(x, a), 1\}} \sum_{\tau \in [t]} \mathbb{I}_\tau(x, a) \left[ \mathbb{E}[g_{\tau,i}(x, a)] - g_i^\circ(x, a) \right] \right|$$

$$\leq \frac{1}{\max\{N_t(x, a), 1\}} \sum_{\tau \in [t]} \mathbb{I}_\tau(x, a) \left| \mathbb{E}[g_{\tau,i}(x, a)] - g_i^\circ(x, a) \right|$$

$$\leq \frac{1}{\max\{N_t(x, a), 1\}} \sum_{\tau \in [T]} \left| \mathbb{E}[g_{\tau,i}(x, a)] - g_i^\circ(x, a) \right|$$

$$\leq \frac{C_G}{\max\{N_t(x, a), 1\}},$$

which concludes the proof. □

We now prove a similar result for the rewards function.

**Lemma 5.** *Fixing* $(x, a) \in X \times A$ , $t \in [T]$, *for any* $\delta \in (0, 1)$, *it holds with probability at least* $1 - \delta$:

$$\left| \widehat{r}_t(x, a) - r^\circ(x, a) \right| \leq \sqrt{\frac{1}{2 \max\{N_t(x, a), 1\}} \ln\left(\frac{2}{\delta}\right)} + \frac{C_r}{\max\{N_t(x, a), 1\}}.$$

*Proof.* The proof is analogous to the one of Lemma 4. □

We now generalize the previous results as follows.

**Lemma 6.** *Given any* $\delta \in (0, 1)$, *for any* $(x, a) \in X \times A, t \in [T]$, *and* $i \in [m]$, *it holds with probability at least* $1 - \delta$:

$$\left| \widehat{g}_{t,i}(x, a) - g_i^\circ(x, a) \right| \leq \sqrt{\frac{1}{2 \max\{N_t(x, a), 1\}} \ln\left(\frac{2mT|X||A|}{\delta}\right)} + \frac{C_G}{\max\{N_t(x, a), 1\}}.$$

*Proof.* First let's define $\zeta_t(x, a)$ as:

$$\zeta_t(x, a) := \sqrt{\frac{1}{2 \max\{N_t(x, a), 1\}} \ln\left(\frac{2}{\delta}\right)} + \frac{C_G}{\max\{N_t(x, a), 1\}}.$$

From Lemma 4, given $\delta' \in (0, 1)$, we have, fixed any $i \in [m]$, $t \in [T]$ and $(x, a) \in X \times A$:

$$\mathbb{P}\left[ \left| \widehat{g}_{t,i}(x, a) - g_i^\circ(x, a) \right| \leq \zeta_t(x, a) \right] \geq 1 - \delta'.$$

Now, we are interested in the intersection of all the events, namely,

$$\mathbb{P}\left[ \bigcap_{x,a,i,t} \left\{ \left| \widehat{g}_{t,i}(x, a) - g_i^\circ(x, a) \right| \leq \zeta_t(x, a) \right\} \right].$$

Thus, we have:

$$\mathbb{P}\left[ \bigcap_{x,a,i,t} \left\{ \left| \widehat{g}_{t,i}(x, a) - g_i^\circ(x, a) \right| \leq \zeta_t(x, a) \right\} \right]$$

$$= 1 - \mathbb{P}\left[ \bigcup_{x,a,i,t} \left\{ \left| \widehat{g}_{t,i}(x, a) - g_i^\circ(x, a) \right| \leq \zeta_t(x, a) \right\}^c \right]$$

$$\geq 1 - \sum_{x,a,i,t} \mathbb{P}\left[ \left\{ \left| \widehat{g}_{t,i}(x, a) - g_i^\circ(x, a) \right| \leq \zeta_t(x, a) \right\}^c \right] \quad (9)$$

$$\geq 1 - |X||A|mT\delta',$$

where Inequality (9) holds by Union Bound. Noticing that $g_{t,i}(x, a) \leq 1$, substituting $\delta'$ with $\delta := \delta'/|X||A|mT$ in $\zeta_t(x, a)$ with an additional Union Bound over the possible values of $N_t(x, a)$, we have, with probability at least $1 - \delta$:

$$\left| \widehat{g}_{t,i}(x, a) - g_i^\circ(x, a) \right| \leq \sqrt{\frac{1}{2 \max\{N_t(x, a), 1\}} \ln\left(\frac{2mT|X||A|}{\delta}\right)} + \frac{C_G}{\max\{N_t(x, a), 1\}},$$

which concludes the proof. □

We provide a similar result for the rewards function.

**Lemma 7.** *Given any* $\delta \in (0, 1)$, *for any* $(x, a) \in X \times A, t \in [T]$, *it holds with probability at least* $1 - \delta$:

$$\left| \widehat{r}_t(x, a) - r^\circ(x, a) \right| \leq \sqrt{\frac{1}{2 \max\{N_t(x, a), 1\}} \ln\left(\frac{2T|X||A|}{\delta}\right)} + \frac{C_r}{\max\{N_t(x, a), 1\}}.$$

*Proof.* First let's define $\psi_t(x, a)$ as:

$$\psi_t(x, a) := \sqrt{\frac{1}{2\max\{N_t(x, a), 1\}} \ln\left(\frac{2}{\delta}\right)} + \frac{C_r}{\max\{N_t(x, a), 1\}}.$$

From Lemma 5, given $\delta' \in (0, 1)$, we have fixed any $t \in [T]$ and $(x, a) \in X \times A$:

$$\mathbb{P}\left[\left|\widehat{r}_t(x, a) - r^\circ(x, a)\right| \leq \psi_t(x, a)\right] \geq 1 - \delta'.$$

Now, we are interested in the intersection of all the events, namely,

$$\mathbb{P}\left[\bigcap_{x, a, t}\left\{\left|\widehat{r}_t(x, a) - r^\circ(x, a)\right| \leq \psi_t(x, a)\right\}\right].$$

Thus, we have:

$$\mathbb{P}\left[\bigcap_{x, a, t}\left\{\left|\widehat{r}_t(x, a) - r^\circ(x, a)\right| \leq \psi_t(x, a)\right\}\right]$$

$$= 1 - \mathbb{P}\left[\bigcup_{x, a, t}\left\{\left|\widehat{r}_t(x, a) - r^\circ(x, a)\right| \leq \psi_t(x, a)\right\}^c\right]$$

$$\geq 1 - \sum_{x, a, t}\mathbb{P}\left[\left\{\left|\widehat{r}_t(x, a) - r^\circ(x, a)\right| \leq \psi_t(x, a)\right\}^c\right] \quad (10)$$

$$\geq 1 - |X||A|T\delta',$$

where Inequality (10) holds by Union Bound. Noticing that $r_t(x, a) \leq 1$, substituting $\delta'$ with $\delta := \delta'/|X||A|T$ in $\psi_t(x, a)$ with an additional Union Bound over the possible values of $N_t(x, a)$, we have, with probability at least $1 - \delta$:

$$\left|\widehat{r}_t(x, a) - r^\circ(x, a)\right| \leq \sqrt{\frac{1}{2\max\{N_t(x, a), 1\}} \ln\left(\frac{2T|X||A|}{\delta}\right)} + \frac{C_r}{\max\{N_t(x, a), 1\}},$$

which concludes the proof. $\square$

In the following, we bound the distance between the empirical estimation of the constraints and the empirical mean of the mean values of the constraints distribution during the learning dynamic.

**Lemma 8.** *Given $\delta \in (0, 1)$, for all episodes $t \in [T]$, state-action pairs $(x, a) \in X \times A$ and constraint $i \in [m]$, it holds, with probability at least $1 - \delta$:*

$$\left|\widehat{g}_{t, i}(x, a) - \frac{1}{T}\sum_{\tau \in [T]}\mathbb{E}[g_{\tau, i}(x, a)]\right| \leq \xi_t(x, a),$$

*where,*

$$\xi_t(x, a) := \min\left\{1, \sqrt{\frac{1}{2\max\{N_t(x, a), 1\}} \ln\left(\frac{2mT|X||A|}{\delta}\right)} + \frac{C_G}{\max\{N_t(x, a), 1\}} + \frac{C_G}{T}\right\}.$$

*Proof.* We first notice that if $\xi_t(x, a) = 1$, the results is derived trivially by definition on the cost function. We prove now the non trivial case $\sqrt{\frac{1}{2\max\{N_t(x, a), 1\}} \ln\left(\frac{2mT|X||A|}{\delta}\right)} + \frac{C_G}{\max\{N_t(x, a), 1\}} + \frac{C_G}{T} \leq 1$. Employing Lemma 3 and Lemma 6, with probability $1 - \delta$ for all $(x, a) \in X \times A$, for all $t \in [T]$ and for all $i \in [m]$, it holds that:

$$\left|\widehat{g}_{t, i}(x, a) - \frac{1}{T}\sum_{\tau \in [T]}\mathbb{E}[g_{\tau, i}(x, a)]\right|$$

$$\leq \left| \widehat{g}_{t,i}(x,a) - g_i^{\circ}(x,a) \right| + \left| g_i^{\circ}(x,a) - \frac{1}{T} \sum_{t \in [T]} \mathbb{E}[g_{t,i}(x,a)] \right|$$

$$\leq \sqrt{\frac{1}{2 \max\{N_t(x,a),1\}} \ln \left( \frac{2mT|X||A|}{\delta} \right)} + \frac{C_G}{\max\{N_t(x,a),1\}} + \frac{C_G}{T},$$

where the first inequality follows from the triangle inequality. This concludes the proof. $\qquad \square$

For the sake of simplicity, we analyze our algorithm with respect to the total corruption of the environment, defined as the maximum between the reward and the constraints corruption. In the following, we show that this choice does not prevent the confidence set events from holding.

**Corollary 1.** *Given a corruption guess $\widehat{C} \geq C_G$ and $\delta \in (0,1)$, for all episodes $t \in [T]$, state-action pairs $(x,a) \in X \times A$ and constraint $i \in [m]$, with probability at least $1 - \delta$, it holds:*

$$\left| \widehat{g}_{t,i}(x,a) - \frac{1}{T} \sum_{\tau \in [T]} \mathbb{E}[g_{\tau,i}(x,a)] \right| \leq \xi_t(x,a),$$

*where,*

$$\xi_t(x,a) = \min \left\{ 1, \sqrt{\frac{1}{2 \max\{N_t(x,a),1\}} \ln \left( \frac{2mT|X||A|}{\delta} \right)} + \frac{\widehat{C}}{\max\{N_t(x,a),1\}} + \frac{\widehat{C}}{T} \right\}.$$

*Proof.* Following the same analysis of Lemma 8 for $\widehat{C} \geq C_G$, it holds

$$\left| \widehat{g}_{t,i}(x,a) - \frac{1}{T} \sum_{\tau \in [T]} \mathbb{E}[g_{\tau,i}(x,a)] \right|$$

$$\leq \sqrt{\frac{1}{2 \max\{N_t(x,a),1\}} \ln \left( \frac{2mT|X||A|}{\delta} \right)} + \frac{C_G}{\max\{N_t(x,a),1\}} + \frac{C_G}{T}$$

$$\leq \sqrt{\frac{1}{2 \max\{N_t(x,a),1\}} \ln \left( \frac{2mT|X||A|}{\delta} \right)} + \frac{\widehat{C}}{\max\{N_t(x,a),1\}} + \frac{\widehat{C}}{T},$$

which concludes the proof. $\qquad \square$

**Corollary 2.** *Taking the definition of $\xi_t$ employed in Lemma 8 and defining $\mathcal{E}_G$ as the intersection event:*

$$\mathcal{E}_G := \left\{ \left| \widehat{g}_{t,i}(x,a) - g_i^{\circ}(x,a) \right| \leq \xi_t(x,a), \ \forall (x,a) \in X \times A, \forall t \in [T], \forall i \in [m] \right\} \bigcap$$

$$\left\{ \left| \widehat{g}_{t,i}(x,a) - \frac{1}{T} \sum_{\tau \in [T]} \mathbb{E}[g_{\tau,i}(x,a)] \right| \leq \xi_t(x,a), \ \forall (x,a) \in X \times A, \forall t \in [T], \forall i \in [m] \right\},$$

*it holds that $\mathbb{P}[\mathcal{E}_G] \geq 1 - \delta$.*

Notice that by Corollary 1, $\mathcal{E}_G$ includes all the analogous events where $\xi_t$ is built employing an arbitrary adversarial corruption $\widehat{C}$ such that $\widehat{C} \geq C_G$.

In the following, we provide similar results for the reward function.

**Lemma 9.** *Given $\delta \in (0,1)$, for all episodes $t \in [T]$ and for all state-action pairs $(x,a) \in X \times A$, with probability at least $1 - \delta$, it holds:*

$$\left| \widehat{r}_t(x,a) - \frac{1}{T} \sum_{\tau \in [T]} \mathbb{E}[r_\tau(x,a)] \right| \leq \phi_t(x,a),$$

*where,*

$$\phi_t(x,a) := \min \left\{ 1, \sqrt{\frac{1}{2 \max\{N_t(x,a),1\}} \ln \left( \frac{2T|X||A|}{\delta} \right)} + \frac{C_r}{\max\{N_t(x,a),1\}} + \frac{C_r}{T} \right\}.$$

*Proof.* Employing Lemma 3 and Lemma 7, with probability at least $1 - \delta$, for all $(x, a) \in X \times A$ and for all $t \in [T]$, it holds:

$$\left| \widehat{r}_t(x, a) - \frac{1}{T} \sum_{\tau \in [T]} \mathbb{E}[r_\tau(x, a)] \right|$$

$$\leq \left| \widehat{r}_t(x, a) - r^\circ(x, a) \right| + \left| r^\circ(x, a) - \frac{1}{T} \sum_{t \in [T]} \mathbb{E}[r_t(x, a)] \right|$$

$$\leq \sqrt{\frac{1}{2 \max\{N_t(x, a), 1\}} \ln\left(\frac{2T|X||A|}{\delta}\right)} + \frac{C_r}{\max\{N_t(x, a), 1\}} + \frac{C_r}{T},$$

where the first inequality follows from the triangle inequality. Noticing that, by construction,

$$\left| \widehat{r}_t(x, a) - \frac{1}{T} \sum_{\tau \in [T]} \mathbb{E}[r_\tau(x, a)] \right| \leq 1,$$

for all episodes $t \in [T]$ and $(x, a) \in X \times A$ concludes the proof. $\square$

We conclude the section, showing the overestimating the reward corruption does not invalidate the confidence set estimation.

**Corollary 3.** *Given a corruption guess $\widehat{C} \geq C_r$ and $\delta \in (0, 1)$, for all episodes $t \in [T]$ and for all state-action pairs $(x, a) \in X \times A$, with probability at least $1 - \delta$, it holds:*

$$\left| \widehat{r}_t(x, a) - \frac{1}{T} \sum_{\tau \in [T]} \mathbb{E}[r_\tau(x, a)] \right| \leq \phi_t(x, a),$$

*where,*

$$\phi_t(x, a) := \min\left\{ 1, \sqrt{\frac{1}{2 \max\{N_t(x, a), 1\}} \ln\left(\frac{2T|X||A|}{\delta}\right)} + \frac{\widehat{C}}{\max\{N_t(x, a), 1\}} + \frac{\widehat{C}}{T} \right\}.$$

*Proof.* The proof is analogous to the one of Corollary 1. $\square$

**Corollary 4.** *Taking the definition of $\phi_t$ employed in Lemma 9 and defining $\mathcal{E}_r$ as the intersection event:*

$$\mathcal{E}_r := \left\{ \left| \widehat{r}_t(x, a) - r^\circ(x, a) \right| \leq \phi_t(x, a), \ \forall (x, a) \in X \times A, \forall t \in [T] \right\} \bigcap$$

$$\left\{ \left| \widehat{r}_t(x, a) - \frac{1}{T} \sum_{\tau \in [T]} \mathbb{E}[r_\tau(x, a)] \right| \leq \phi_t(x, a), \ \forall (x, a) \in X \times A, \forall t \in [T] \right\},$$

*it holds that $\mathbb{P}[\mathcal{E}_r] \geq 1 - \delta$.*

Notice that by Corollary 3, $\mathcal{E}_r$ includes all the analogous events where $\phi_t$ is built employing an arbitrary adversarial corruption $\widehat{C}$ such that $\widehat{C} \geq C_r$.

## D  OMITTED PROOFS WHEN THE CORRUPTION IS KNOWN

In the following, we provide the main results attained by Algorithm 2 in term of regret and constraints violations. The following results hold when the corruption of the environment is known to the learner.

We start providing a preliminary result, which shows that the linear program solved by Algorithm 2 at each $t \in [T]$ admits a feasible solution, with high probability.

**Lemma 10.** *For any $\delta \in (0, 1)$, for all episodes $t \in [T]$, with probability at least $1 - 5\delta$, the space defined by linear constraints $\left\{ q \in \Delta(\mathcal{P}_t) : \underline{G}_t^\top q \leq \alpha \right\}$ admits a feasible solution and it holds:*

$$\left\{ q \in \Delta(P) : \overline{G}^\top q \leq \alpha \right\} \subseteq \left\{ q \in \Delta(\mathcal{P}_t) : \underline{G}_t^\top q \leq \alpha \right\}.$$

*Proof.* Under the event $\mathcal{E}_P$, we have that $\Delta(P) \subseteq \Delta(\mathcal{P}_t)$, for all episodes $t \in [T]$. Similarly, under the event $\mathcal{E}_G$, it holds that $\left\{ q : \frac{1}{T} \sum_{t \in [T]} \mathbb{E}[G_t]^\top q \le \alpha \right\} \subseteq \left\{ q : \underline{G}_t^\top q \le \alpha \right\}$. This implies that any feasible solution of the offline problem, is included in the optimistic safe set $\left\{ q \in \Delta(\mathcal{P}_t) : \underline{G}_t^\top q \le \alpha \right\}$. Taking the intersection event $\mathcal{E}_P \cap \mathcal{E}_G$ concludes the proof. $\square$

We are now ready to provide the violation bound attained by Algorithm 2.

**Theorem 2.** *Given any $\delta \in (0, 1)$, with probability at least $1 - 8\delta$, Algorithm 2 attains violations*
$$V_T = \mathcal{O}\left( L|X|\sqrt{|A|T \ln\left(\frac{mT|X||A|}{\delta}\right)} + \ln(T)|X||A|C \right).$$

*Proof.* In the following, we will refer as $\mathcal{E}_{\widehat{q}}$ to the event described in Lemma 20, which holds with probability at least $1 - 6\delta$. Thus, under $\mathcal{E}_G \cap \mathcal{E}_{\widehat{q}}$, the linear program solved by Algorithm 2 has a feasible solution (see Lemma 10) and it holds:

$$V_T = \max_{i \in [m]} \sum_{t \in [T]} \left[ \mathbb{E}[G_t]^\top q_t - \alpha \right]_i^+$$

$$= \max_{i \in [m]} \sum_{t \in [T]} \left[ (\mathbb{E}[g_{t,i}] - g_i^\circ)^\top q_t + g_i^{\circ \top} q_t - \alpha_i \right]^+$$

$$\le \max_{i \in [m]} \sum_{t \in [T]} \left[ (\mathbb{E}[g_{t,i}] - g_i^\circ)^\top q_t + \left( \underline{g}_{t-1,i} + 2\xi_{t-1} \right)^\top q_t - \alpha_i \right]^+ \tag{11a}$$

$$= \max_{i \in [m]} \sum_{t \in [T]} \left[ (\mathbb{E}[g_{t,i}] - g_i^\circ)^\top q_t + \underline{g}_{t-1,i}^\top (q_t - \widehat{q}_t) + \underline{g}_{t-1,i}^\top \widehat{q}_t + 2\xi_{t-1}^\top q_t - \alpha_i \right]^+$$

$$\le \max_{i \in [m]} \sum_{t \in [T]} \left[ (\mathbb{E}[g_{t,i}] - g_i^\circ)^\top q_t + \underline{g}_{t-1,i}^\top (q_t - \widehat{q}_t) + 2\xi_{t-1}^\top q_t \right]^+ \tag{11b}$$

$$\le \max_{i \in [m]} \sum_{t \in [T]} \left| (\mathbb{E}[g_{t,i}] - g_i^\circ)^\top q_t \right| + 2\max_{i \in [m]} \sum_{t \in [T]} \left| \xi_{t-1}^\top q_t \right| + \max_{i \in [m]} \sum_{t \in [T]} \left| \underline{g}_{t-1,i}^\top (q_t - \widehat{q}_t) \right| \tag{11c}$$

$$\le \max_{i \in [m]} \sum_{t \in [T]} \| \mathbb{E}[g_{t,i}] - g_i^\circ \|_1 + 2\max_{i \in [m]} \sum_{t \in [T]} \xi_{t-1}^\top q_t + \max_{i \in [m]} \sum_{t \in [T]} \| q_t - \widehat{q}_t \|_1 \tag{11d}$$

$$\le C_G + 2\max_{i \in [m]} \sum_{t \in [T]} \xi_{t-1}^\top q_t + \sum_{t \in [T]} \| q_t - \widehat{q}_t \|_1, \tag{11e}$$

where Inequality (11a) follows from Corollary 2, Inequality (11b) holds since Algorithm 2 ensures, for all $t \in [T]$ and for all $i \in [m]$, that $\underline{g}_{t,i}^\top \widehat{q}_t \le \alpha_i$, Inequality (11c) holds since $[a + b]^+ \le |a| + |b|$, for all $a, b \in \mathbb{R}$, Inequality (11d) follows from Hölder inequality since $\| \underline{g}_{t,i}(x, a) \|_\infty \le 1$ and $\| q_t(x, a) \|_\infty \le 1$, and finally Equation (11e) holds for the definition of $C_G$.

To bound the last term of Equation (11e), we notice that, under $\mathcal{E}_{\widehat{q}}$, by Lemma 20, it holds:

$$\sum_{t \in [T]} \| q_t - \widehat{q}_t \|_1 = \mathcal{O}\left( L|X|\sqrt{|A|T \ln\left(\frac{T|X||A|}{\delta}\right)} \right).$$

To bound the second term of Equation (11e) we proceed as follows. Under $\mathcal{E}_{\widehat{q}}$, with probability at least $1 - \delta$, it holds:

$$\sum_{t \in [T]} \xi_{t-1}^\top q_t \le \sum_{t \in [T]} \sum_{x,a} \xi_{t-1}(x, a) \mathbb{I}_t(x, a) + L\sqrt{2T \ln \frac{1}{\delta}} \tag{12a}$$

$$\le \sum_{x,a} \sum_{t \in [T]} \mathbb{I}_t(x, a) \left( \sqrt{\frac{1}{2 \max\{N_{t-1}(x, a), 1\}} \ln\left(\frac{2mT|X||A|}{\delta}\right)} + \tag{12b-partial} \right.$$

$$+ \frac{C_G}{\max\{N_{t-1}(x,a),1\}} + \frac{C_G}{T}\Bigg) + L\sqrt{2T\ln\frac{1}{\delta}} \tag{12b}$$

$$\leq \sqrt{\frac{1}{2}\ln\left(\frac{2mT|X||A|}{\delta}\right)}\sum_{x,a}\sum_{t\in[T]}\mathbb{I}_t(x,a)\sqrt{\frac{1}{\max\{N_{t-1}(x,a),1\}}} +$$

$$+ C_G\sum_{x,a}\sum_{t\in[T]}\left(\frac{\mathbb{I}_t(x,a)}{\max\{N_{t-1}(x,a),1\}} + \frac{1}{T}\right) + L\sqrt{2T\ln\frac{1}{\delta}}$$

$$\leq 3\sqrt{\frac{1}{2}|X||A|LT\ln\left(\frac{2mT|X||A|}{\delta}\right)} + |X||A|(2+\ln(T))C_G + |X||A|C_G + L\sqrt{2T\ln\frac{1}{\delta}} \tag{12c}$$

$$\leq 3\sqrt{\frac{1}{2}|X||A|LT\ln\left(\frac{2mT|X||A|}{\delta}\right)} + (3+\ln(T))|X||A|C_G + L\sqrt{2T\ln\frac{1}{\delta}}$$

$$= \mathcal{O}\left(\sqrt{|X||A|LT\ln\left(\frac{mT|X||A|}{\delta}\right)} + \ln(T)|X||A|C_G\right),$$

where Inequality (12a) follows from the Azuma-Hoeffding inequality and noticing that $\sum_{x,a}\xi_{t-1}(x,a)q_t(x,a) \leq L$, Equality (12b) follows from the definition of $\xi_t$ and finally, Inequality (12c) holds since $1 + \sum_{t=1}^{N_T(x,a)}\sqrt{\frac{1}{t}} \leq 1 + 2\sqrt{N_T(x,a)} \leq 3\sqrt{N_T(x,a)}$ , since $1 + \sum_{t=1}^{N_T(x,a)}\frac{1}{t} \leq 2 + \ln(T)$ and by Cauchy-Schwarz inequality. Finally, we notice that the intersection event $\mathcal{E}_G \cap \mathcal{E}_{\widehat{q}} \cap \mathcal{E}_{\text{Azuma}}$ holds with the following probability,

$$\mathbb{P}\left[\mathcal{E}_G \cap \mathcal{E}_{\widehat{q}} \cap \mathcal{E}_{\text{Azuma}}\right] = 1 - \mathbb{P}\left[\mathcal{E}_G^C \cup \mathcal{E}_{\widehat{q}}^C \cup \mathcal{E}_{\text{Azuma}}^C\right]$$

$$\geq 1 - \left(\mathbb{P}\left[\mathcal{E}_G^C\right] + \mathbb{P}\left[\mathcal{E}_{\widehat{q}}^C\right] + \mathbb{P}\left[\mathcal{E}_{\text{Azuma}}^C\right]\right)$$

$$\geq 1 - 8\delta.$$

Noticing that, by Corollary 1, what holds for a $\xi_t$ built with corruption value $C_G$, still holds for a higher corruption (by definition, $C \geq C_G$) concludes the proof. $\qquad\square$

We conclude the section providing the regret bound attained by Algorithm 2.

**Theorem 3.** *Given any $\delta \in (0,1)$, with probability at least $1 - 9\delta$, Algorithm 2 attains regret $R_T = \mathcal{O}\left(L|X|\sqrt{|A|T\ln\left(T|X||A|/\delta\right)} + \ln(T)|X||A|C\right)$.*

*Proof.* First, we notice that under the event $\mathcal{E}_r$ it holds that, for all $(x,a) \in X \times A$ and for all $t \in [T]$:

$$\bar{r}_t(x,a) - 2\phi_t(x,a) \leq \frac{1}{T}\sum_{t\in[T]}\mathbb{E}[r_t(x,a)].$$

Let's observe that, by Lemma 10, under the event $\mathcal{E}_G \cap \mathcal{E}_P$, $\widehat{q}_t$ is optimal solution for $\bar{r}_{t-1}$ in $\left\{q \in \Delta(\mathcal{P}_t) : \underline{G}_t^\top q \leq \alpha\right\}$. Thus, under $\mathcal{E}_G \cap \mathcal{E}_P$ the optimal feasible solution $q^*$ is such that:

$$\bar{r}_{t-1}^\top \widehat{q}_t \geq \bar{r}_{t-1}^\top q^*.$$

Thus under the event $\mathcal{E}_r$, it holds:

$$\frac{1}{T}\sum_{t\in[T]}\mathbb{E}[r_t]^\top q^* \leq \bar{r}_{t-1}^\top q^*$$

$$\leq \bar{r}_{t-1}^\top \widehat{q}_t$$

$$\leq \left(\frac{1}{T}\sum_{t\in[T]}\mathbb{E}[r_t] + 2\phi_{t-1}\right)^\top \widehat{q}_t.$$

Thus, we can rewrite the regret (under the event $\mathcal{E}_G \cap \mathcal{E}_r \cap \mathcal{E}_P$) as,

$$R_T = \sum_{t \in [T]} \mathbb{E}[r_t]^\top (q^* - q_t)$$

$$= \sum_{t \in [T]} \frac{1}{T} \sum_{\tau \in [T]} \mathbb{E}[r_\tau]^\top (q^* - q_t) + \sum_{t \in [T]} (\mathbb{E}[r_t] - \bar{r})^\top (q^* - q_t)$$

$$= \sum_{t \in [T]} \frac{1}{T} \sum_{\tau \in [T]} \mathbb{E}[r_\tau]^\top (q^* - \widehat{q}_t + \widehat{q}_t - q_t) + \sum_{t \in [T]} (\mathbb{E}[r_t] - r^\circ + r^\circ - \bar{r})^\top (q^* - q_t)$$

$$\leq \sum_{t \in [T]} \left[ \frac{1}{T} \sum_{\tau \in [T]} \mathbb{E}[r_\tau]^\top (q^* - \widehat{q}_t) \right] + \sum_{t \in [T]} \|\widehat{q}_t - q_t\|_1 + \sum_{t \in [T]} \|\mathbb{E}[r_t] - r^\circ\|_1 + \sum_{t \in [T]} \|r^\circ - \bar{r}\|_1$$

$$\leq \sum_{t \in [T]} 2\phi_{t-1}^\top q_t + \sum_{t \in [T]} \|\widehat{q}_t - q_t\|_1 + 2C_r.$$

By Lemma 19 with probability at least $1 - 6\delta$ under event $\mathcal{E}_{\widehat{q}}$ we can bound $\sum_{t \in [T]} \|\widehat{q}_t - q_t\|_1$ as:

$$\sum_{t \in [T]} \|\widehat{q}_t - q_t\|_1 = \mathcal{O}\left( L|X| \sqrt{|A|T \ln\left(\frac{T|X||A|}{\delta}\right)} \right).$$

Finally with probability at least $1 - \delta$ it holds:

$$\sum_{t \in [T]} \phi_{t-1}^\top q_t \leq \sum_{t \in [T]} \sum_{x,a} \phi_{t-1}(x,a) \mathbb{I}_t(x,a) + L\sqrt{2T \ln \frac{1}{\delta}} \tag{13a}$$

$$\leq \sum_{x,a} \sum_{t \in [T]} \mathbb{I}_t(x,a) \left( \sqrt{\frac{1}{2 \max\{N_{t-1}(x,a), 1\}} \ln\left(\frac{2T|X||A|}{\delta}\right)} + \right.$$

$$\left. + \frac{C_r}{\max\{N_{t-1}(x,a), 1\}} + \frac{C_r}{T} \right) + L\sqrt{2T \ln \frac{1}{\delta}} \tag{13b}$$

$$\leq \sqrt{\frac{1}{2} \ln\left(\frac{2T|X||A|}{\delta}\right)} \sum_{x,a} \sum_{t \in [T]} \mathbb{I}_t(x,a) \sqrt{\frac{1}{\max\{N_{t-1}(x,a), 1\}}} +$$

$$+ C_r \sum_{x,a} \sum_{t \in [T]} \left( \frac{\mathbb{I}_t(x,a)}{\max\{N_{t-1}(x,a), 1\}} + \frac{1}{T} \right) + L\sqrt{2T \ln \frac{1}{\delta}}$$

$$\leq 3\sqrt{\frac{1}{2} |X||A| L T \ln\left(\frac{2T|X||A|}{\delta}\right)} + |X||A|(2 + \ln(T)) C_r + |X||A| C_r + L\sqrt{2T \ln \frac{1}{\delta}} \tag{13c}$$

$$\leq 3\sqrt{\frac{1}{2} |X||A| L T \ln\left(\frac{2T|X||A|}{\delta}\right)} + (3 + \ln(T))|X||A| C_r + L\sqrt{2T \ln \frac{1}{\delta}}$$

$$= \mathcal{O}\left( \sqrt{|X||A| L T \ln\left(\frac{T|X||A|}{\delta}\right)} + \ln(T)|X||A| C_r \right),$$

where Inequality (13a) follows from Azuma-Hoeffding inequality, Equality (13b) holds for the definition of $\phi_t$, and Inequality (13c) holds since $1 + \sum_{t=1}^{N_T(x,a)} \sqrt{\frac{1}{t}} \leq 1 + 2\sqrt{N_T(x,a)} \leq 3\sqrt{N_T(x,a)}$, $1 + \sum_{t=1}^{N_T(x,a)} \frac{1}{t} \leq 2 + \ln(T)$ and by Cauchy-Schwarz inequality. Thus, we observe that with probability at least $1 - 9\delta$ it holds:

$$R_T = \mathcal{O}\left( L|X| \sqrt{|A|T \ln\left(\frac{T|X||A|}{\delta}\right)} + \ln(T)|X||A| C_r \right).$$

Employing Corollary 3 and the definition of $C$, which is at least equal to $C_r$, concludes the proof. $\square$

# E  OMITTED PROOFS WHEN THE KNOWLEDGE OF $C$ IS NOT PRECISE

In this section, we focus on the performances of Algorithm 2 when a guess on the corruption value is given as input. These preliminary results are "the first step" to relax the assumption on the knowledge about the corruption.

First, we present some preliminary results on the confidence set.

**Lemma 11.** *Given the corruption guess $\widehat{C}_G$, where $C_G = \widehat{C}_G + \epsilon$, with $\epsilon > 0$, and confidence $\xi_t$ as defined in Algorithm 2 using $\widehat{C}_G$ as corruption value, for any $\delta \in (0,1)$, with probability at least $1 - \delta$, for all episodes $t \in [T]$, state-action pair $(x,a) \in X \times A$ and constraint $i \in [m]$, the following result holds:*

$$g_i^\circ(x,a) \le \widehat{g}_{t,i}(x,a) + \xi_t(x,a) + \left( \frac{\epsilon}{\max\{N_t(x,a),1\}} + \frac{\epsilon}{T} \right).$$

*Similarly, recalling the definition of $\underline{G}_t$, for all episodes $t \in [T]$, state-action pairs $(x,a) \in X \times A$ and constraints $i \in [m]$, it holds:*

$$g_i^\circ(x,a) \le \underline{g}_{t,i}(x,a) + 2\xi_t(x,a) + \left( \frac{\epsilon}{\max\{N_t(x,a),1\}} + \frac{\epsilon}{T} \right).$$

*Proof.* To prove the result, we recall that, by Corollary 2, with probability at least $1 - \delta$, the following holds, for all episodes $t \in [T]$, state-action pairs $(x,a) \in X \times A$ and constraints $i \in [m]$:

$$\left| \widehat{g}_{t,i}(x,a) - g_i^\circ(x,a)] \right| \le$$

$$\sqrt{ \frac{1}{2\max\{N_t(x,a),1\}} \ln\left( \frac{2mT|X||A|}{\delta} \right) } + \frac{C_G}{\max\{N_t(x,a),1\}} + \frac{C_G}{T},$$

which can be rewritten as:

$$\left| \widehat{g}_{t,i}(x,a) - g_i^\circ(x,a)] \right| \le \xi_t(x,a) + \frac{\epsilon}{\max\{N_t(x,a),1\}} + \frac{\epsilon}{T},$$

where,

$$\xi_t(x,a) := \min\left\{ 1, \sqrt{ \frac{1}{2\max\{N_t(x,a),1\}} \ln\left( \frac{2mT|X||A|}{\delta} \right) } + \frac{\widehat{C}_G}{\max\{N_t(x,a),1\}} + \frac{\widehat{C}_G}{T} \right\},$$

and $C_G = \widehat{C}_G + \epsilon$, which concludes the proof. $\qquad\square$

We are now ready study the regret bound attained by the algorithm when the guess on the corruption is an overestimate.

**Theorem 7.** *For any $\delta \in (0,1)$, Algorithm 2, when instantiated with corruption value $\widehat{C}$ which is an overestimate of the true value of $C$, i.e. $\widehat{C} > C_G$ and $\widehat{C} > C_r$, attains with probability at least $1 - 8\delta$:*

$$R_T = \mathcal{O}\left( L|X|\sqrt{|A|T\ln\left( \frac{T|X||A|}{\delta} \right)} + \ln(T)|X||A|\widehat{C} \right).$$

*Proof.* By Corollary 1, it holds that the decision space of the linear program performed by Algorithm 2 contains with high probability the optimal solution that respects to the constraints. Furthermore, employing Corollary 3 and following the proof of Theorem 3 concludes the proof. $\qquad\square$

We proceed bounding the violation attained by our algorithm when an underestimate of the corruption is given as input.

**Theorem 8.** *For any $\delta \in (0,1)$, Algorithm 2, when instantiated with corruption value $\widehat{C}$ which is an underestimate of the true value of $C_G$, i.e. $\widehat{C} < C_G$, attains with probability at least $1 - 9\delta$:*

$$V_T = \mathcal{O}\left( L|X|\sqrt{|A|T\ln\left( \frac{mT|X||A|}{\delta} \right)} + \ln(T)|X||A|C_G \right).$$

*Proof.* First, let's define $\epsilon \in \mathbb{R}^+$ such that $\epsilon := C_G - \widehat{C}$. Then, with probability at least $1 - \delta$:

$$V_T = \max_{i \in [m]} \sum_{t \in [T]} \left[ \mathbb{E}[G_t]^\top q_t - \alpha \right]_i^+ \tag{14a}$$

$$= \max_{i \in [m]} \sum_{t \in [T]} \left[ (\mathbb{E}[g_{t,i}] - g_i^\circ)^\top q_t + g_i^{\circ\top} q_t - \alpha_i \right]^+$$

$$\leq \max_{i \in [m]} \sum_{t \in [T]} \left[ (\mathbb{E}[g_{t,i}] - g_i^\circ)^\top q_t + \underline{g}_{t-1,i}^\top (q_t - \widehat{q}_t) + \underline{g}_{t-1,i}^\top \widehat{q}_t + 2\xi_{t-1}^\top q_t + \right.$$

$$\left. + \sum_{x,a} \left( \frac{\epsilon}{\max\{N_{t-1}(x,a), 1\}} + \frac{\epsilon}{T} \right) q_t(x,a) - \alpha_i \right]^+ \tag{14b}$$

$$\leq C_G + 2\max_{i \in [m]} \sum_{t \in [T]} \xi_{t-1}^\top q_t + \sum_{t \in [T]} \|q_t - \widehat{q}_t\|_1 +$$

$$+ \sum_{t \in [T]} \sum_{x,a} \frac{\epsilon}{\max\{N_{t-1}(x,a), 1\}} q_t(x,a) + \epsilon L, \tag{14c}$$

where Inequality (14b) follows from Lemma 11 and Inequality (14c) is derived as in the proof of Theorem 2, and considering that $\|q_t\|_1 = L$, $\forall t \in [T]$. Now, employing the Azuma-Hoeffding inequality, we can bound, with probability at least $1 - \delta$ the term $\sum_{t=1}^T \sum_{x,a} \frac{\epsilon}{\max\{N_{t-1}(x,a),1\}} q_t(x,a)$ as follows:

$$\sum_{t \in [T]} \sum_{x,a} \frac{\epsilon}{\max\{N_{t-1}(x,a), 1\}} q_t(x,a) \leq L\sqrt{2T \ln \frac{1}{\delta}} + \sum_{t \in [T]} \sum_{x,a} \frac{\epsilon}{\max\{N_{t-1}(x,a), 1\}} \mathbb{I}_t(x,a)$$

$$\leq L\sqrt{2T \ln \frac{1}{\delta}} + \epsilon |X||A|(1 + \ln(T)),$$

where we applied Azume Hoeffding inequality and the fact that $\sum_{t \in [N_T(x,a)]} \frac{1}{t} \leq 1 + \ln(T)$. Finally, following the steps of the proof of Theorem 2 to bound the first 3 elements of Inequality (14c) under $\mathcal{E}_{\widehat{q}}$ with probability at least $1 - \delta$, and considering that $\epsilon \leq C_G$ and $\widehat{C} \leq C_G$, it holds, with probability at least $1 - 9\delta$,

$$V_T = \mathcal{O}\left( L|X|\sqrt{|A|T \ln\left(\frac{T|X||A|}{\delta}\right)} + \ln(T)|X||A|C_G \right),$$

which concludes the proof. $\qquad\square$

Finally, we provide the violation bound attained by Algorithm 2 when an overestimate of the corruption value is given as input.

**Theorem 9.** *For any $\delta \in (0,1)$, Algorithm 2, when instantiated with corruption value $\widehat{C}$ which is an overestimate of the true value of $C_G$, i.e. $\widehat{C} > C_G$, attains with probability at least $1 - 8\delta$:*

$$V_T = \mathcal{O}\left( L|X|\sqrt{|A|T \ln\left(\frac{T|X||A|}{\delta}\right)} + \ln(T)|X||A|\widehat{C} \right).$$

*Proof.* The proof follows by employing Corollary 1 to the proof of Theorem 2. $\qquad\square$

# F OMITTED PROOFS WHEN THE CORRUPTION IS *not* KNOWN

In the following section we provide the omitted proofs of the theoretical guarantees attained by Algorithm 3. The algorithm is designed to work when the corruption value is *not* known.

## F.1 LAGRANGIAN FORMULATION OF THE CONSTRAINED OPTIMIZATION PROBLEM

Since Algorithm 3 is based on a Lagrangian formulation of the constrained problem, it is necessary to show that this approach is well characterized. Precisely, we show that a *strong duality-like* result holds even when the Lagrangian function is defined taking the positive violations.

First, we show that strong duality holds with respect to the standard Lagrangian function, even considering a subset of the Lagrangian multiplier space.

**Lemma 2.** *Given a CMDP with a transition function $P$, for every reward vector $r \in [0,1]^{|X \times A|}$, constraint cost matrix $G \in [0,1]^{|X \times A| \times m}$, and threshold vector $\alpha \in [0,L]^m$, if Program (4) satisfies Slater's condition (Condition 1), then the following holds:*

$$\min_{\|\lambda\|_1 \in [0, L/\rho]} \max_{q \in \Delta(P)} r^\top q - \sum_{i \in [m]} \lambda_i \left[G^\top q - \alpha\right]_i = \max_{q \in \Delta(P)} \min_{\|\lambda\|_1 \in [0, L/\rho]} r^\top q - \sum_{i \in [m]} \lambda_i \left[G^\top q - \alpha\right]_i$$

$$= \mathrm{OPT}_{r,G,\alpha},$$

*where $\lambda \in \mathbb{R}^m_{\geq 0}$ is a vector of Lagrangian multipliers and $\rho$ is the feasibility parameter of Program (4).*

*Proof.* The proof follows the one of Theorem 3.3 in (Castiglioni et al., 2022b). First we prove that, given the Lagrangian function $\mathcal{Q}(\lambda, q) := r^\top q - \sum_{i \in [m]} \lambda_i \left(G_i^\top q - \alpha_i\right)$, it holds:

$$\min_{\|\lambda\|_1 \in [0, L/\rho]} \max_{q \in \Delta(P)} \mathcal{Q}(\lambda, q) = \min_{\lambda \in \mathbb{R}^m_{\geq 0}} \max_{q \in \Delta(P)} \mathcal{Q}(\lambda, q),$$

with $\lambda \in \mathbb{R}^m_{\geq 0}$. In fact notice that for all $\lambda \in \mathbb{R}^m_{\geq 0}$ such that $\|\lambda\|_1 > L/\rho$:

$$\max_{q \in \Delta(P)} \mathcal{Q}(\lambda, q) \geq \mathcal{Q}(\lambda, q^\circ) \geq -\sum_{i \in [m]} \lambda_i \left(G_i^\top q^\circ - \alpha_i\right) \geq \|\lambda\|_1 \rho > L,$$

where $q^\circ$ is defined as $q^\circ := \arg\max_{q \in \Delta(P)} \min_{i \in [m]} \left[\alpha_i - G_i^\top q\right]$. Moreover since

$$\min_{\|\lambda\|_1 \in [0, L/\rho]} \max_{q \in \Delta(P)} \mathcal{Q}(\lambda, q) \leq \max_{q \in \Delta(P)} \mathcal{Q}(\underline{0}, q) = \max_{q \in \Delta(P)} r^\top q \leq L,$$

it holds:

$$\min_{\lambda \in \mathbb{R}^m_{\geq 0}} \max_{q \in \Delta(P)} \mathcal{Q}(\lambda, q) = \min \left\{ \min_{\|\lambda\|_1 \in [0, L/\rho]} \max_{q \in \Delta(P)} \mathcal{Q}(\lambda, q), \min_{\|\lambda\|_1 \geq L/\rho} \max_{q \in \Delta(P)} \mathcal{Q}(\lambda, q) \right\}$$

$$= \min_{\|\lambda\|_1 \in [0, L/\rho]} \max_{q \in \Delta(P)} \mathcal{Q}(\lambda, q).$$

Thus,

$$\mathrm{OPT}_{r,G,\alpha} = \max_{q \in \Delta(P)} \min_{\lambda \in \mathbb{R}^m_{\geq 0}} \mathcal{Q}(\lambda, q)$$

$$\leq \max_{q \in \Delta(P)} \min_{\|\lambda\|_1 \geq L/\rho} \mathcal{Q}(\lambda, q)$$

$$\leq \min_{\|\lambda\|_1 \geq L/\rho} \max_{q \in \Delta(P)} \mathcal{Q}(\lambda, q)$$

$$= \min_{\lambda \in \mathbb{R}^m_{\geq 0}} \max_{q \in \Delta(P)} \mathcal{Q}(\lambda, q)$$

$$= \mathrm{OPT}_{r,G,\alpha},$$

where the second inequality holds by the *max-min* inequality and the last step holds by the well-known strong duality result in CMDPs (Altman, 1999). This concludes the proof. □

In the following, we extend the previous result for the Lagrangian function which encompasses the positive violations.

**Theorem 4.** *Given a CMDP with a transition function $P$, for every reward vector $r \in [0,1]^{|X \times A|}$, constraint cost matrix $G \in [0,1]^{|X \times A| \times m}$, and threshold vector $\alpha \in [0,L]^m$, if Program (4) satisfies Slater's condition (Condition 1), then the following holds:*

$$\max_{q \in \Delta(P)} \mathcal{L}(L/\rho, q) = \max_{q \in \Delta(P)} r^\top q - \frac{L}{\rho} \sum_{i \in [m]} \left[G^\top q - \alpha\right]_i^+ = \mathrm{OPT}_{r,G,\alpha},$$

*where $\rho$ is the feasibility parameter of Program (4).*

*Proof.* Following the definition of Lagrangian function, we have:

$$\max_{q \in \Delta(P)} \mathcal{L}(L/\rho, q) = \max_{q \in \Delta(P)} r^\top q - \frac{L}{\rho} \sum_{i \in [m]} \left[ G_i^\top q - \alpha_i \right]^+$$

$$\leq \max_{q \in \Delta(P)} \min_{\|\lambda\|_1 \in [0, L/\rho]} r^\top q - \sum_{i \in [m]} \lambda_i [G_i^\top q - \alpha_i]^+$$

$$\leq \min_{\|\lambda\|_1 \in [0, L/\rho]} \max_{q \in \Delta(P)} r^\top q - \sum_{i \in [m]} \lambda_i [G_i^\top q - \alpha_i]^+$$

$$\leq \min_{\|\lambda\|_1 \in [0, L/\rho]} \max_{q \in \Delta(P)} r^\top q - \sum_{i \in [m]} \lambda_i \left( G_i^\top q - \alpha_i \right)$$

$$= \mathrm{OPT}_{r,G,\alpha}$$

where $\lambda \in \mathbb{R}_{\geq 0}^m$ is the Lagrangian vector, the second inequality holds by the *max-min inequality* and the last step follows from Lemma 2. Noticing that for all $q$ belonging to $\{q \in \Delta(P) : G^\top q \leq \alpha\}$, we have $\mathcal{L}(1/\rho, q) = r^\top q$, which implies that $\max_{q \in \Delta(P)} \mathcal{L}(1/\rho, q) \geq \mathrm{OPT}_{r,G,\alpha}$, concludes the proof. $\square$

## F.2 PRELIMINARY RESULTS

In the following sections we will refer as:

$$\widehat{V}_T := \sum_{t \in [T]} \sum_{j \in [M]} \frac{w_{t,j} \mathbb{I}(j_t = j)}{w_{t,j} + \gamma} \sum_{i \in [m]} \left[ \widehat{g}_{t,i}^{j\top} \widehat{q}_t^j - \alpha_i \right]^+, \tag{15}$$

to the estimated violation attained by the instances of Algorithm 3. Furthermore, we will refer as:

$$\widehat{V}_{T,j^*} := \sum_{t \in [T]} \frac{\mathbb{I}(j_t = j^*)}{w_{t,j^*} + \gamma} \sum_{i \in [m]} \left[ \widehat{g}_{t,i}^{j^*\top} \widehat{q}_t^{j^*} - \alpha_i \right]^+, \tag{16}$$

to the estimated violation attained by the optimal instance $j^*$, namely, the integer in $[M]$ such that the true corruption $C \in [2^{j^*-1}, 2^{j^*}]$.

Furthermore, we will refer as $q_t^j$ to the occupancy measure induced by the policy proposed by $\mathtt{Alg}^j$ at episode $t$, with $j \in [M], t \in [T]$, and we will refer as:

$$\widehat{g}_{t,i}^j(x,a) := \frac{\sum_{\tau \in [t]} \mathbb{I}_\tau(x,a) \mathbb{I}(j_\tau = j) g_{\tau,i}(x,a)}{\max\{N_t^j(x,a), 1\}},$$

to the estimate of the cost computed for $j$-th algorithm, where $N_t^j(x,a)$ is a counter initialize to 0 in $t = 0$, and which increases by one from episode $t$ to episode $t+1$ whenever $\mathbb{I}_t(x,a)\mathbb{I}(j_t = j) = 1$.

### F.2.1 STABILITY PARAMETERS

In the following sections, we will employ the stability parameters $\beta$ defined as follows:

- $\beta_1 = \mathcal{O}\left( L^2 |X|^2 |A| \ln\left( \frac{T|X||A|}{\delta} \right) \right)$

- $\beta_2 = \mathcal{O}\left( |X|^2 |A|^2 \log(T) \log\left( \frac{\log(T)}{\delta} \right) \right)$

- $\beta_3 = \mathcal{O}\left( \ln(T)^2 |X||A| \right)$

- $\beta_4 = \mathcal{O}\left( L^2 |X|^2 |A| \ln\left( \frac{mT|X||A|}{\delta} \right) \right)$

- $\beta_5 = \mathcal{O}\left( |X|^2 |A|^2 \log(T) \log\left( \frac{\log(T)}{\delta} \right) \right)$

- $\beta_6 = \mathcal{O}\left( \ln(T)^2 |X||A| \right)$

F.2.2 OMITTED PROOFS AND LEMMAS

We start providing some preliminary results on the optimistic estimator employed by Algorithm 3.

**Lemma 12.** *For any $\delta \in (0,1)$, given $\gamma \in \mathbb{R}_{\geq 0}$, with probability at least $1 - \delta$, it holds:*

$$\widehat{R}_T \leq \mathcal{O}\left(\gamma TLM + L\sqrt{2T \ln\left(\frac{1}{\delta}\right)}\right),$$

*where $\widehat{R}_T = \sum_{t \in [T]} \sum_{j \in [M]} \left(w_{t,j}\left(L - \mathbb{E}[r_t]^\top q_t^j\right) - \frac{w_{t,j}\mathbb{I}(j_t=j)}{w_{t,j}+\gamma} \sum_{(x_k^t, a_k^t)} \left(1 - r_t\left(x_k^t, a_k^t\right)\right)\right).$*

*Proof.* We first observe that by construction:

$$\mathbb{E}\left[\sum_{t \in [T]} \sum_{j \in [M]} \frac{w_{t,j}\mathbb{I}(j_t = j)}{w_{t,j}+\gamma} \sum_{(x_k^t, a_k^t)} \left(1 - r_t\left(x_k^t, a_k^t\right)\right)\right] = \sum_{t \in [T]} \sum_{j \in [M]} \frac{w_{t,j}^2}{w_{t,j}+\gamma}\left(L - \mathbb{E}[r_t]^\top q_t^j\right).$$

Moreover, still by construction, for all episodes $t \in [T]$, it holds:

$$\sum_{j \in [M]} \frac{w_{t,j}\mathbb{I}(j_t = j)}{w_{t,j}+\gamma} \sum_{(x_k^t, a_k^t)} \left(1 - r_t\left(x_k^t, a_k^t\right)\right) \leq \sum_{j \in [M]} \mathbb{I}(j_t = j) \sum_{(x_k^t, a_k^t)} \left(1 - r_t\left(x_k^t, a_k^t\right)\right) \leq L.$$

Thus, employing Azuma-Hoeffding inequality, with probability at least $1 - \delta$, it holds:

$$\sum_{t \in [T]} \sum_{j \in [M]} \left(\frac{w_{t,j}^2}{w_{t,j}+\gamma}(L - \mathbb{E}[r_t]^\top q_t^j) - \frac{w_{t,j}\mathbb{I}(j_t=j)}{w_{t,j}+\gamma} \sum_{(x_k^t, a_k^t)} \left(1 - r_t(x_k^t, a_k^t)\right)\right) \leq L\sqrt{2T \ln\left(\frac{1}{\delta}\right)}.$$

Finally we notice that:

$$\sum_{t \in [T]} \sum_{j \in [M]} w_{t,j}\left(L - \mathbb{E}[r_t]^\top q_t^j\right) - \sum_{t \in [T]} \sum_{j \in [M]} \frac{w_{t,j}^2}{w_{t,j}+\gamma}\left(L - \mathbb{E}[r_t]^\top q_t^j\right)$$

$$= \sum_{t \in [T]} \sum_{j \in [M]} \left(\frac{w_{t,j}}{w_{t,j}+\gamma}\right)\gamma\left(L - \mathbb{E}[r_t]^\top q_t^j\right)$$

$$\leq \gamma TLM.$$

Adding and subtracting $\mathbb{E}\left[\sum_{t \in [T]} \sum_{j \in [M]} \frac{w_{t,j}\mathbb{I}(j_t=j)}{w_{t,j}+\gamma} \sum_{(x_k^t, a_k^t)} \left(1 - r_t\left(x_k^t, a_k^t\right)\right)\right]$ to the quantity of interest and employing the previous bound concludes the proof. $\square$

We provide an additional result on the optimistic estimator employed by Algorithm 3.

**Lemma 13.** *For any $\delta \in (0,1)$, given $\gamma \in \mathbb{R}_{\geq 0}$, with probability at least $1 - \delta$, it holds:*

$$\sum_{t \in [T]} \frac{\mathbb{I}(j_t = j^*)}{w_{t,j^*}+\gamma} \sum_{(x_k^t, a_k^t)} \left(1 - r_t\left(x_k^t, a_k^t\right)\right) - \sum_{t \in [T]} \left(L - \mathbb{E}[r_t]^\top q_t^{j^*}\right) = \mathcal{O}\left(\frac{L}{\gamma} \ln\left(\frac{1}{\delta}\right)\right)$$

*Proof.* The proof closely follows the idea of Corollary 5. We define the loss $\bar{\ell}_t = \sum_{(x_k^t, a_k^t)}(1 - r_t(x_k^t, a_k^t))$, the optimistic loss estimator $\widehat{\ell}_t := \frac{\mathbb{I}(j_t=j^*)}{w_{t,j^*}+\gamma} \sum_{(x_k^t, a_k^t)}(1 - r_t(x_k^t, a_k^t))$ and the unbiased estimator $\widetilde{\ell}_t := \frac{\mathbb{I}(j_t=j^*)}{w_{t,j^*}} \sum_{(x_k^t, a_k^t)}(1 - r_t(x_k^t, a_k^t))$.

Employing the same argument as Neu (2015) it holds:

$$\widehat{\ell}_t = \frac{\mathbb{I}(j_t = j^*)}{w_{t,j^*}+\gamma}\bar{\ell}_t \leq \frac{\mathbb{I}(j_t = j^*)}{w_{t,j^*}+\gamma\bar{\ell}_t/L}\bar{\ell}_t \leq \frac{L}{2\gamma}\frac{2\gamma\bar{\ell}_t/w_{t,j^*}L}{1+\gamma\bar{\ell}_t/w_{t,j^*}L}\mathbb{I}(j_t = j^*) \leq \frac{L}{2\gamma}\ln\left(1 + \frac{2\gamma}{L}\widetilde{\ell}_t\right),$$

since $\frac{z}{1+z/2} \leq \ln(1 + z), z \in \mathbb{R}^+$. Employing the previous inequality, it holds:

$$\mathbb{E}\left[\exp\left(\frac{2\gamma}{L}\widehat{\ell}_t\right)\bigg|\mathcal{F}_{t-1}\right] \leq \mathbb{E}\left[\exp\left(\frac{2\gamma}{L}\frac{L}{2\gamma}\ln\left(1 + \frac{2\gamma}{L}\widetilde{\ell}_t\right)\right)\bigg|\mathcal{F}_{t-1}\right]$$

$$= \mathbb{E}\left[1 + \frac{2\gamma}{L}\widetilde{\ell}_t \middle| \mathcal{F}_{t-1}\right]$$

$$= 1 + \frac{2\gamma}{L}\mathbb{E}\left[\frac{\mathbb{I}(j_t = j^*)}{w_{t,j^*}}\sum_{(x_k^t, a_k^t)}(1 - r_t(x_k^t, a_k^t))\middle| \mathcal{F}_{t-1}\right]$$

$$\leq 1 + \frac{2\gamma}{L}\left(L - \mathbb{E}[r_t]^\top q_t^{j^*}\right)$$

$$\leq \exp\left(\frac{2\gamma}{L}\left(L - \mathbb{E}[r_t]^\top q_t^{j^*}\right)\right),$$

where $\mathcal{F}_{t-1}$ is the filtration up to episode $t$. We conclude the proof employing the Markov inequality as follows:

$$\mathbb{P}\left(\sum_{t \in [T]}\frac{2\gamma}{L}\left(\widehat{\ell}_t - \left(L - \mathbb{E}[r_t]^\top q_t^{j^*}\right)\right) \geq \epsilon\right)$$

$$\leq \mathbb{E}\left[\exp\left(\sum_{t \in [T]}\frac{2\gamma}{L}\left(\widehat{\ell}_t - \left(L - \mathbb{E}[r_t]^\top q_t^{j^*}\right)\right)\right)\right]\exp(-\epsilon)$$

$$\leq \exp(-\epsilon).$$

Solving $\delta = \exp(-\epsilon)$ for $\epsilon$ we obtain:

$$\mathbb{P}\left(\sum_{t \in [T]}\left(\widehat{\ell}_t - \left(L - \mathbb{E}[r_t]^\top q_t^{j^*}\right)\right) \geq \frac{L}{2\gamma}\ln\left(\frac{1}{\delta}\right)\right) \leq \delta.$$

This concludes the proof. $\qquad\square$

We are now ready to prove the regret bound attained by FTRL with respect to the Lagrangian underlying problem.

**Lemma 14.** *For any $\delta \in (0,1)$ and properly setting the learning rate $\eta$ such that $\eta \leq \frac{1}{2\Lambda m(\sqrt{\beta_1 T} + \beta_2 + \beta_5 + \sqrt{\beta_4 T})}$, Algorithm 3 attains, with probability at least $1 - 2\delta$:*

$$\sum_{t \in [T]}\mathbb{E}[r_t]^\top q_t^{j^*} - \sum_{t \in [T]}\sum_{j \in [M]}w_{t,j}\mathbb{E}[r_t]^\top q_t^j + \frac{Lm+1}{\rho}\widehat{V}_T - \frac{Lm+1}{\rho}\widehat{V}_{T,j^*}$$

$$+ \left(\frac{m(mL+1)}{\rho}\beta_5 + \beta_2\right)\nu_{T,j^*} + \left(\sqrt{\beta_1} + \left(\frac{m(Lm+1)}{\rho}\right)\sqrt{\beta_4}\right)\sqrt{T}\nu_{T,j^*}$$

$$\leq \mathcal{O}\left(\frac{M\ln T}{\eta} + \eta\, m^4 L^4 T M + \eta\, M\ln(T)m^4 L^2\beta_5^2 + \eta\, M\ln(T)\beta_2^2\right.$$

$$\left. + \eta T(\beta_1 + L^2 m^4 \beta_4)M\log(T) + \gamma TLM + L\sqrt{T\ln(1/\delta)} + \frac{L}{\gamma}\ln(1/\delta)\right).$$

*Proof.* First, we define $\ell_{t,j}$, for all $t \in [T]$, for all $j \in [M]$ as:

$$\ell_{t,j} := \frac{\mathbb{I}(j_t = j)}{w_{t,j} + \gamma}\left(\sum_{(x_k^t, a_k^t)}(1 - r_t(x_k^t, a_k^t)) + \frac{Lm+1}{\rho}\sum_{i \in [m]}\left[\widehat{g}_{t,i}^{j\top}\widehat{q}_t^j - \alpha_i\right]^+\right),$$

and $b_{t,j}$ for all $t \in [T]$, for all $j \in [M]$ as:

$$b_{t,j} := \left(\left(\frac{m(mL+1)}{\rho}\beta_5 + \beta_2\right) + \left(\sqrt{\beta_1} + \frac{m(Lm+1)}{\rho}\sqrt{\beta_4}\right)\sqrt{T}\right)(\nu_{t,j} - \nu_{t-1,j}),$$

with $\nu_{t,j} = \max_{\tau \in [t]}\frac{1}{w_{\tau,j}}$.

First we prove that $\eta w_{t,j}|\ell_{t,j} - b_{t,j}| \leq 1/2$ for all $t \in [T], j \in [M]$, to apply Lemma 17. It

holds that $\eta w_{t,j}|\ell_{t,j}| \leq \frac{\eta(L\rho + L^2m^2 + Lm)}{\rho} \leq \frac{1}{2}$ for all $j \in [M]$, for all $t \in [T]$ as long as $\eta \leq \frac{\rho}{2(L\rho + L^2m^2 + Lm)} \leq \frac{\rho}{2(L^2m^2 + Lm)}$, which is true if $\eta \leq \frac{\rho}{2Lm(Lm+1)}$. It also holds that

$$\eta w_{t,j}|b_{t,j}| = \eta w_{t,j}\left(\left(\frac{m(Lm+1)}{\rho}\beta_5 + \beta_2\right) + \left(\frac{m(Lm+1)}{\rho}\sqrt{\beta_4} + \sqrt{\beta_1}\right)\sqrt{T}\right)(\nu_{t,j} - \nu_{t-1,j})$$

$$\leq \eta\left(\left(\frac{m(Lm+1)}{\rho}\beta_5 + \beta_2\right) + \left(\frac{m(Lm+1)}{\rho}\sqrt{\beta_4} + \sqrt{\beta_1}\right)\sqrt{T}\right)\left(1 - \frac{\nu_{t-1,j}}{\nu_{t,j}}\right)$$

$$\leq \eta\left(\left(\frac{m(Lm+1)}{\rho}\beta_5 + \beta_2\right) + \left(\frac{m(Lm+1)}{\rho}\sqrt{\beta_4} + \sqrt{\beta_1}\right)\sqrt{T}\right)$$

$$\leq \frac{1}{2},$$

if $\eta \leq \frac{1}{2\Lambda m\left(\sqrt{\beta_1 T} + \beta_2 + \beta_5 + \sqrt{\beta_4 T}\right)}$, where we used the fact that $\nu_{t,j} \neq \nu_{t-1,j} \iff 1/w_{t,j} = \nu_{t,j}$. Thus, if the previous conditions on $\eta$ hold, and notice that the second condition implies the first, Algorithm 3 attains, by Lemma 17 :

$$\sum_{t\in[T]}\left[\sum_{j\in[M]}\frac{w_{t,j}\mathbb{I}(j_t = j)}{w_{t,j} + \gamma}\sum_{(x_k^t, a_k^t)}(1 - r_t(x_k^t, a_k^t)) - \frac{\mathbb{I}(j_t = j^*)}{w_{t,j^*} + \gamma}\sum_{(x_k^t, a_k^t)}(1 - r_t(x_k^t, a_k^t))\right] + \frac{Lm+1}{\rho}\widehat{V}_T$$

$$\leq \frac{M\ln T}{\eta} + 2\eta\frac{TM(L\rho + L^2m^2 + Lm)^2}{\rho^2}$$

$$+ 2\eta\left(2\left(\frac{m(mL+1)}{\rho}\beta_5 + \beta_2\right)^2 M\ln(T) + 2T\left(\sqrt{\beta_1} + \left(\frac{m(Lm+1)}{\rho}\right)\sqrt{\beta_4}\right)^2 M\ln(T)\right)$$

$$+ \frac{Lm+1}{\rho}\widehat{V}_{T,j^*} + \sum_{t\in[T]}\sum_{j\in[M]}w_{t,j}b_{t,j} - \sum_{t\in[T]}b_{t,j^*}, \qquad (17)$$

where we used the following inequalities:

- First inequality:

$$\sum_{t\in[T]}\sum_{j\in[M]}w_{t,j}^2(\ell_{t,j} - b_{t,j})^2 \leq 2\sum_{t\in[T]}\sum_{j\in[M]}w_{t,j}^2\ell_{t,j}^2 + 2\sum_{t\in[T]}\sum_{j\in[M]}w_{t,j}^2b_{t,j}^2,$$

- Second inequality:

$$\left(\sum_{(x_k^t, a_k^t)}(1 - r_t(x_k^t, a_k^t)) + \frac{Lm+1}{\rho}\sum_{i\in[m]}\left[\widehat{g}_{t,i}^{j\top}\widehat{q}_t^j - \alpha_i\right]^+\right) \leq \frac{(L\rho + L^2m^2 + Lm)}{\rho},$$

- Third inequality:

$$\sum_{t\in[T]}\sum_{j\in[M]}w_{t,j}^2\ell_{t,j}^2 \leq \frac{TM(L\rho + L^2m^2 + Lm)^2}{\rho^2},$$

and that, it holds:

$$\sum_{t\in[T]}\sum_{j\in[M]}w_{t,j}^2b_{t,j}^2$$

$$= \sum_{t\in[T]}\sum_{j\in[M]}(w_{t,j}b_{t,j})^2$$

$$\leq \left(\left(\frac{m(Lm+1)}{\rho}\beta_5 + \beta_2\right) + \left(\frac{m(Lm+1)}{\rho}\sqrt{\beta_4} + \sqrt{\beta_1}\right)\sqrt{T}\right)^2\sum_{j\in[M]}\sum_{t\in[T]}\left(\frac{1}{\nu_{t,j}}(\nu_{t,j} - \nu_{t-1,j})\right)^2$$

$$\tag{18a}$$

$$\leq \left( 2\left( \frac{m(mL+1)}{\rho}\beta_5 + \beta_2 \right)^2 + 2T\left( \frac{m(Lm+1)}{\rho}\sqrt{\beta_4} + \sqrt{\beta_1} \right)^2 \right) \sum_{j\in[M]}\sum_{t\in[T]} \left( 1 - \frac{\nu_{t-1,j}}{\nu_{t,j}} \right)^2$$

$$\leq \left( 2\left( \frac{m(mL+1)}{\rho}\beta_5 + \beta_2 \right)^2 + 2T\left( \frac{m(Lm+1)}{\rho}\sqrt{\beta_4} + \sqrt{\beta_1} \right)^2 \right) \sum_{j\in[M]}\sum_{t\in[T]} \left( 1 - \frac{\nu_{t-1,j}}{\nu_{t,j}} \right)$$

$$\leq \left( 2\left( \frac{m(mL+1)}{\rho}\beta_5 + \beta_2 \right)^2 + 2T\left( \frac{m(Lm+1)}{\rho}\sqrt{\beta_4} + \sqrt{\beta_1} \right)^2 \right) \sum_{j\in[M]}\sum_{t\in[T]} \ln\left( \frac{\nu_{t,j}}{\nu_{t-1,j}} \right)$$

(18b)

$$\leq \left( 2\left( \frac{m(mL+1)}{\rho}\beta_5 + \beta_2 \right)^2 + 2T\left( \frac{m(Lm+1)}{\rho}\sqrt{\beta_4} + \sqrt{\beta_1} \right)^2 \right) \sum_{j\in[M]} \ln\left( \prod_{t\in[T]} \frac{\nu_{t,j}}{\nu_{t-1,j}} \right)$$

$$\leq \left( 2\left( \frac{m(mL+1)}{\rho}\beta_5 + \beta_2 \right)^2 + 2T\left( \frac{m(Lm+1)}{\rho}\sqrt{\beta_4} + \sqrt{\beta_1} \right)^2 \right) \sum_{j\in[M]} \ln\left( \frac{\nu_{T,j}}{\nu_{0,j}} \right)$$

$$\leq \left( 2\left( \frac{m(mL+1)}{\rho}\beta_5 + \beta_2 \right)^2 + 2T\left( \frac{m(Lm+1)}{\rho}\sqrt{\beta_4} + \sqrt{\beta_1} \right)^2 \right) M\ln(T), \qquad (18c)$$

where Inequality (18a) is true since $\nu_{t,j} - \nu_{t-1,j} \neq 0$ only when $w_{t,j} = 1/\nu_{t,j}$ by definition, Inequality (18b) holds since $1 - a \leq -\ln a$, and Inequality (18c) holds since by definition $\nu_{T,j} \leq T$ and $\nu_{0,j} = M$. Notice also that, following a similar reasoning, it holds:

$$\sum_{t\in[T]} w_{t,j}b_{t,j} - \sum_{t\in[T]} b_{t,j^*}$$

$$= \left( \left( \frac{m(Lm+1)}{\rho}\beta_5 + \beta_2 \right) + \left( \frac{m(Lm+1)}{\rho}\sqrt{\beta_4} + \sqrt{\beta_1} \right)\sqrt{T} \right) \sum_{t\in[T]}\sum_{j\in[M]} \left( 1 - \frac{\nu_{t-1,i}}{\nu_{t,i}} \right)$$

$$- \left( \left( \frac{m(Lm+1)}{\rho}\beta_5 + \beta_2 \right) + \left( \frac{m(Lm+1)}{\rho}\sqrt{\beta_4} + \sqrt{\beta_1} \right)\sqrt{T} \right) \sum_{t\in[T]} (\nu_{t,j^*} - \nu_{t-1,j^*})$$

$$\leq \mathcal{O}\left( m^2 L\beta_5 M\log(T) + \beta_2 M\log(T) + (\sqrt{\beta_1} + Lm^2\sqrt{\beta_4})\sqrt{T}M\log(T) \right)$$

$$- \left( \left( \frac{m(Lm+1)}{\rho}\beta_5 + \beta_2 \right) + \left( \frac{m(Lm+1)}{\rho}\sqrt{\beta_4} + \sqrt{\beta_1} \right)\sqrt{T} \right)\nu_{T,j^*}$$

Thus, with probability at least $1 - 2\delta$, it holds:

$$\sum_{t\in[T]} \mathbb{E}[r_t]^\top q_t^{j^*} - \sum_{t\in[T]}\sum_{j\in[M]} w_{t,j}\mathbb{E}[r_t]^\top q_t^j + \frac{Lm+1}{\rho}\widehat{V}_T$$

$$= \sum_{t\in[T]}\sum_{j\in[M]} w_{t,j}\left( L - \mathbb{E}[r_t]^\top q_t^j \right) - \sum_{t\in[T]} \left( L - \mathbb{E}[r_t]^\top q_t^{j^*} \right) + \frac{Lm+1}{\rho}\widehat{V}_T \qquad (19)$$

$$\leq \mathcal{O}\left( \frac{M\ln T}{\eta} + \eta\, m^4 L^4 TM + \eta\, M\ln(T)m^4 L^2\beta_5^2 + \eta\, M\ln(T)\beta_2^2 \right.$$

$$\left. + \eta T(\beta_1 + L^2 m^4\beta_4)M\log(T) + \gamma TLM + L\sqrt{T\ln(1/\delta)} + \frac{L}{\gamma}\ln(1/\delta) \right) + \frac{Lm+1}{\rho}\widehat{V}_{T,j^*}$$

$$- \left( \frac{m(mL+1)}{\rho}\beta_5 + \beta_2 \right)\nu_{T,j^*} - \left( \sqrt{\beta_1} + \left( \frac{m(Lm+1)}{\rho} \right)\sqrt{\beta_4} \right)\sqrt{T}\nu_{T,j*}, \qquad (20)$$

where Equation (19) holds since $\sum_{j\in[M]} w_{t,j} = 1$, $\forall t \in [T]$, and Inequality (20) holds, with probability at least $1 - 2\delta$, by Lemma 12, Lemma 13 and Equation (17). This concludes the proof. $\qquad\square$

In order to provide the desired bound $R_T$ and $V_T$ for Algorithm 3, it is necessary to study the relation between the aforementioned performance measures and the terms appearing from the FTRL analysis in Lemma 14.

Thus, we bound the distance between the incurred violation and the estimated one.

**Lemma 15.** *For any $\gamma \in \mathbb{R}_{\geq 0}$, given $\delta \in (0,1)$, with probability at least $1 - 10\delta$, it holds:*

$$V_T - \widehat{V}_T = \mathcal{O}\left(mL|X|\sqrt{|A|T\ln\left(\frac{mT|X||A|}{\delta}\right)} + m\ln(T)|X||A|C + \gamma TLM\right).$$

*Proof.* We start defining the quantity $\widehat{\xi}_{t,j}(x,a)$ – for all episode $t \in [T]$, for all state-action pairs $(x,a) \in X \times A$, for all instance $j \in [M]$ – as in Theorem 2 but using the true value of adversarial corruption $C$, considering that the counter $N_t^j(x,a)$ increases on one unit from episode $t$ to $t+1$, if and only if $\mathbb{I}(j_t = j)\mathbb{I}_t(x,a) = 1$, and by applying a Union Bound over all instances $j \in [M]$ namely,

$$\widehat{\xi}_{t,j}(x,a) := \min\left\{1, \sqrt{\frac{1}{2\max\{N_t^j(x,a),1\}}\ln\left(\frac{2mMT|X||A|}{\delta}\right)} + \frac{C}{\max\{N_t^j(x,a),1\}} + \frac{C}{T}\right\},$$
(21)

By Corollary 2, and applying a Union Bound on instances $j \in [M]$ simultaneously $\forall t \in [T], \forall i \in [m], \forall (x,a) \in X \times A, \forall j \in [M]$, with probability at least $1 - \delta$, it holds:

$$\widehat{g}_{t,i}^j(x,a) + \widehat{\xi}_{t,j}(x,a) \geq g_i^\circ(x,a).$$
(22)

Resorting to the definition of $\widehat{V}_T$, we obtain that, with probability at least $1 - \delta$, under $\mathcal{E}_{\widehat{q}}$:

$$\widehat{V}_T = \sum_{t \in [T]} \sum_{j \in [M]} \frac{w_{t,j}\mathbb{I}(j_t = j)}{w_{t,j} + \gamma} \sum_{i \in [m]} \left[\widehat{g}_{t,i}^{j\top}\widehat{q}_t^j - \alpha_i\right]^+$$

$$= \sum_{t \in [T]} \sum_{j \in [M]} \frac{w_{t,j}\mathbb{I}(j_t = j)}{w_{t,j} + \gamma} \sum_{i \in [m]} \left[(\widehat{g}_{t,i}^{j\top}q_t^j + \widehat{\xi}_{t,j}^\top q_t^j - \alpha_i) - \widehat{\xi}_{t,j}^\top q_t^j - \widehat{g}_{t,i}^{j\top}(q_t^j - \widehat{q}_t^j)\right]^+$$

$$\geq \sum_{t \in [T]} \sum_{j \in [M]} \frac{w_{t,j}\mathbb{I}(j_t = j)}{w_{t,j} + \gamma} \sum_{i \in [m]} \left(\left[(\widehat{g}_{t,i}^j + \widehat{\xi}_{t,j})^\top q_t^j - \alpha_i\right]^+ - \widehat{\xi}_{t,j}^\top q_t^j - \widehat{g}_{t,i}^{j\top}|q_t^j - \widehat{q}_t^j|\right) \quad (23a)$$

$$\geq \sum_{t \in [T]} \sum_{j \in [M]} \frac{w_{t,j}\mathbb{I}(j_t = j)}{w_{t,j} + \gamma} \sum_{i \in [m]} \left(\left[g_i^{\circ\top}q_t^j - \alpha_i\right]^+ - \widehat{\xi}_{t,j}^\top q_t^j - \|q_t^j - \widehat{q}_t^j\|_1\right) \quad (23b)$$

$$\geq \sum_{t \in [T]} \sum_{j \in [M]} \frac{w_{t,j}\mathbb{I}(j_t = j)}{w_{t,j} + \gamma} \sum_{i \in [m]} \left(\left[\mathbb{E}[g_{t,i}]^\top q_t^j - \alpha_i\right]^+ - \widehat{\xi}_{t,j}^\top q_t^j\right) - \sum_{t \in [T]} \sum_{j \in [M]} \frac{w_{t,j}\mathbb{I}(j_t = j)}{w_{t,j} + \gamma}$$

$$\cdot \sum_{i \in [m]} \left[(g_i^\circ - \mathbb{E}[g_{t,i}])^\top q_t^j\right]^+ - \mathcal{O}\left(mL|X|\sqrt{|A|T\ln\left(\frac{T|X||A|}{\delta}\right)}\right), \quad (23c)$$

where Inequality (23a) holds since $[a - b]^+ \geq [a]^+ - b$, $a \in \mathbb{R}, b \in \mathbb{R}_{\geq 0}$, Inequality (23b) follows from Inequality (22) and since, by definition, $\widehat{g}_{t,i}^j(x,a) \leq 1, \forall (x,a) \in X \times A, \forall i \in [m], \forall t \in [T], \forall j \in [M]$ and, finally, Inequality (23c) holds under event $\mathcal{E}_{\widehat{q}}$ by Lemma 20 after noticing that $\sum_{t \in [T]} \sum_{j \in [M]} \frac{w_{t,j}\mathbb{I}(j_t=j)}{w_{t,j}+\gamma} \sum_{i \in [m]} \|q_t^j - \widehat{q}_t^j\|_1 \leq \sum_{t \in [T]} \sum_{j \in [M]} \mathbb{I}(j_t = j)\left(\frac{w_{t,j}}{w_{t,j}+\gamma}\right) \sum_{i \in [m]} \|q_t^j - \widehat{q}_t^j\|_1 \leq m\sum_{t \in [T]} \|q_t^{j_t} - \widehat{q}_t^{j_t}\|_1$.

We will bound the previous terms separately.

**Lower-bound to** $\sum_{t \in [T]} \sum_{j \in [M]} \frac{w_{t,j}\mathbb{I}(j_t=j)}{w_{t,j}+\gamma} \sum_{i \in [m]} \left[\mathbb{E}[g_{t,i}]^\top q_t^j - \alpha_i\right]^+$.

We bound the term by the Azuma-Hoeffding inequality. Indeed, with probability at least $1 - \delta$, it holds:

$$\sum_{t \in [T]} \sum_{j \in [M]} \frac{w_{t,j}\mathbb{I}(j_t = j)}{w_{t,j} + \gamma} \sum_{i \in [m]} \left[\mathbb{E}[g_{t,i}]^\top q_t^j - \alpha_i\right]^+$$

$$\geq \left( \sum_{t \in [T]} \sum_{j \in [M]} \frac{w_{t,j}^2}{w_{t,j} + \gamma} \sum_{i \in [m]} \left[ \mathbb{E}[g_{t,i}]^\top q_t^j - \alpha_i \right]^+ \right) - mL\sqrt{2T \ln\left(\frac{1}{\delta}\right)},$$

where we used the following upper-bound to the martingale sequence:

$$\sum_{j \in [M]} \frac{w_{t,j} \mathbb{I}(j_t = j)}{w_{t,j} + \gamma} \sum_{i \in [m]} \left[ \mathbb{E}[g_{t,i}]^\top q_t^j - \alpha_i \right]^+ \leq \sum_{j \in [M]} \mathbb{I}(j_t = j) \left( \frac{w_{t,j}}{w_{t,j} + \gamma} \right) \sum_{i \in [m]} \left[ \mathbb{E}[g_{t,i}]^\top q_t^j \right]^+$$

$$\leq \sum_{j \in [M]} \mathbb{I}(j_t = j) \sum_{i \in [m]} \|q_t^j\|_1$$

$$\leq m\|q_t^{j_t}\|_1$$

$$\leq mL.$$

Moreover, we observe the following bounds:

$$\sum_{t \in [T]} \sum_{j \in [M]} w_{t,j} \sum_{i \in [m]} \left[ \mathbb{E}[g_{t,i}]^\top q_t^j - \alpha_i \right]^+ - \sum_{t \in [T]} \sum_{j \in [M]} \frac{w_{t,j}^2}{w_{t,j} + \gamma} \sum_{i \in [m]} \left[ \mathbb{E}[g_{t,i}]^\top q_t^j - \alpha_i \right]^+$$

$$\leq \gamma T L m,$$

and,

$$\sum_{t \in [T]} \sum_{j \in [M]} w_{t,j} \sum_{i \in [m]} \left[ \mathbb{E}[g_{t,i}]^\top q_t^j - \alpha_i \right]^+ \geq \sum_{j \in [M]} \max_{i \in [m]} \sum_{t \in [T]} w_{t,j} \left[ \mathbb{E}[g_{t,i}]^\top q_t^j - \alpha_i \right]^+.$$

Combining the previous results, we obtain, with probability at least $1 - \delta$:

$$\sum_{t \in [T]} \sum_{j \in [M]} \frac{w_{t,j} \mathbb{I}(j_t = j)}{w_{t,j} + \gamma} \sum_{i \in [m]} \left[ \mathbb{E}[g_{t,i}]^\top q_t^j - \alpha_i \right]^+$$

$$\geq \sum_{j \in [M]} \max_{i \in [m]} \sum_{t \in [T]} w_{t,j} \left[ \mathbb{E}[g_{t,i}]^\top q_t^j - \alpha_i \right]^+ - \left( \gamma T L m + L m \sqrt{2T \ln\left(\frac{1}{\delta}\right)} \right).$$

**Upper-bound to** $\sum_{t \in [T]} \sum_{j \in [M]} \frac{w_{t,j} \mathbb{I}(j_t = j)}{w_{t,j} + \gamma} \sum_{i \in [m]} \widehat{\xi}_{t,j}^\top q_t^j$.

We bound the term noticing that, with probability at least $1 - \delta$, it holds:

$$\sum_{t \in [T]} \sum_{j \in [M]} \frac{w_{t,j} \mathbb{I}(j_t = j)}{w_{t,j} + \gamma} \sum_{i \in [m]} \widehat{\xi}_{t,j}^\top q_t^j$$

$$\leq \sum_{j \in [M]} m \max_{i \in [m]} \sum_{t \in [T]} \frac{w_{t,j} \mathbb{I}(j_t = j)}{w_{t,j} + \gamma} \widehat{\xi}_{t,j}^\top q_t^j$$

$$\leq \sum_{j \in [M]} m \max_{i \in [m]} \sum_{t \in [T]} \sum_{x,a} \mathbb{I}(j_t = j) \mathbb{I}_t(x,a) \widehat{\xi}_{t,j}(x,a) + L\sqrt{2T \ln \frac{1}{\delta}}$$

$$= \mathcal{O}\left( m\sqrt{|X||A|LT \ln\left(\frac{mMT|X||A|}{\delta}\right)} + m \ln T|X||A|C + L\sqrt{T \ln \frac{1}{\delta}} \right),$$

where we employed the Azuma-Hoeffding inequality and where the last step holds following the proof of Theorem 2.

**Upper-bound to** $\sum_{t \in [T]} \sum_{j \in [M]} \frac{w_{t,j} \mathbb{I}(j_t = j)}{w_{t,j} + \gamma} \sum_{i \in [m]} \left[ (g_i^\circ - \mathbb{E}[g_{t,i}])^\top q_t^j \right]^+$.

We simply bound the quantity of interest as follows:

$$\sum_{t \in [T]} \sum_{j \in [M]} \frac{w_{t,j} \mathbb{I}(j_t = j)}{w_{t,j} + \gamma} \sum_{i \in [m]} \left[ (g_i^\circ - \mathbb{E}[g_{t,i}])^\top q_t^j \right]^+$$

$$\leq m \max_{i \in [m]} \sum_{t \in [T]} \sum_{j \in [M]} \mathbb{I}(j_t = j) \|g_i^\circ - \mathbb{E}[g_{t,i}]\|_1$$

$$\leq mC.$$

**Final result.** To conclude we employ the Azuma-Hoeffding inequality on the violation definition, obtaining, with probability at least $1 - \delta$:

$$V_T = \sum_{j \in [M]} \max_{i \in [m]} \sum_{t \in [T]} \mathbb{I}(j_t = j) \left[ \mathbb{E}[g_{t,i}]^\top q_t^j - \alpha_i \right]^+$$

$$\leq \sum_{j \in [M]} \max_{i \in [m]} \sum_{t \in [T]} w_{t,j} \left[ \mathbb{E}[g_{t,i}]^\top q_t^j - \alpha_i \right]^+ + L\sqrt{2T \ln\left(\frac{1}{\delta}\right)}.$$

Thus, plugging the previous bounds in Equation (23c), we obtain, with probability at least $1 - 10\delta$:

$$V_T - \widehat{V}_T$$

$$\leq \sum_{j \in [M]} \max_{i \in [m]} \sum_{t \in [T]} \mathbb{I}(j_t = j) \left[ \mathbb{E}[g_{t,i}]^\top q_t^j - \alpha_i \right]^+ - \sum_{t \in [T]} \sum_{j \in [M]} \frac{w_{t,j} \mathbb{I}(j_t = j)}{w_{t,j} + \gamma} \sum_{i \in [m]} \left[ \widehat{g}_{t,i}^{j}{}^\top \widehat{q}_t^j - \alpha_i \right]^+$$

$$\leq m \sum_{t \in [T]} \sum_{j \in [M]} \frac{w_{t,j} \mathbb{I}(j_t = j)}{w_{t,j} + \gamma} \widehat{\xi}_{t,j}^\top q_t^j + \sum_{t \in [T]} \sum_{j \in [M]} \frac{w_{t,j} \mathbb{I}(j_t = j)}{w_{t,j} + \gamma} \sum_{i \in [m]} \left[ \frac{1}{T} \sum_{\tau \in [T]} (\mathbb{E}[g_{\tau,i}] - \mathbb{E}[g_{t,i}])^\top q_t^j \right]^+$$

$$+ \gamma T L m + 2Lm\sqrt{2T\left(\frac{1}{\delta}\right)} + \mathcal{O}\left( mL|X|\sqrt{|A|T \ln\left(\frac{T|X||A|}{\delta}\right)} \right)$$

$$= \mathcal{O}\left( mL|X|\sqrt{|A|T \ln\left(\frac{mMT|X||A|}{\delta}\right)} + m\ln(T)|X||A|C + \gamma T L M \right)$$

This concludes the proof. $\qquad\square$

We proceed bounding the estimated violation attained by the optimal instance $j^*$.

**Lemma 16.** *For any $\delta \in (0,1)$, with probability at least $1 - 16\delta$, it holds:*

$$\widehat{V}_{T,j^*} \leq \mathcal{O}\left( mL|X|\sqrt{|A|T \ln\left(\frac{mMT|X||A|}{\delta}\right)} + m\beta_6 C + m\ln(T)|X||A|C + Lm\frac{\ln\left(\frac{M}{\delta}\right)}{2\gamma} \right)$$

$$+ m\sqrt{\beta_4 T}\nu_{T,j^*} + m\beta_5 \nu_{T,j^*}.$$

*Proof.* We start by observing that with, probability at least $1 - \delta$ under $\mathcal{E}_{\widehat{q}}$, the quantity of interest is bounded as follows:

$$\sum_{t \in [T]} \frac{\mathbb{I}(j_t = j^*)}{w_{t,j^*} + \gamma} \sum_{i \in [m]} \left[ \widehat{g}_{t,i}^{j^*}{}^\top \widehat{q}_t^{j^*} - \alpha_i \right]^+$$

$$\leq \sum_{t \in [T]} \frac{\mathbb{I}(j_t = j^*)}{w_{t,j^*} + \gamma} \sum_{i \in [m]} \left( \left[ \widehat{g}_{t,i}^{j^*}{}^\top (\widehat{q}_t^{j^*} - q_t^{j^*}) + \widehat{g}_{t,i}^{j^*}{}^\top q_t^{j^*} - \widehat{\xi}_{t,j^*}^\top q_t^{j^*} - \alpha_i \right]^+ + \widehat{\xi}_{t,j^*}^\top q_t^{j^*} \right) \quad (24a)$$

$$\leq \sum_{t \in [T]} \frac{\mathbb{I}(j_t = j^*)}{w_{t,j^*} + \gamma} \sum_{i \in [m]} \left( \left[ \mathbb{E}[g_{t,i}]^\top q_t^{j^*} - \alpha_i \right]^+ + \widehat{\xi}_{t,j^*}^\top q_t^{j^*} + \right.$$

$$\left. + \left[ g_i^{\circ\top} q_t^{j^*} - \mathbb{E}[g_{t,i}]^\top q_t^{j^*} \right]^+ + \|\widehat{q}_t^{j^*} - q_t^{j^*}\|_1 \right) \quad (24b)$$

$$\leq \sum_{t \in [T]} \frac{\mathbb{I}(j_t = j^*)}{w_{t,j^*} + \gamma} \sum_{i \in [m]} \left( \left[ \mathbb{E}[g_{t,i}]^\top q_t^{j^*} - \alpha_i \right]^+ + \widehat{\xi}_{t,j^*}^\top q_t^{j^*} + \left[ (g_i^\circ - \mathbb{E}[g_{t,i}])^\top q_t^{j^*} \right]^+ \right)$$

$$+ \mathcal{O}\left( L|X|\sqrt{|A|T\ln\left(\frac{T|X||A|}{\delta}\right)} \right), \quad \text{(24c)}$$

where Inequality (24a) holds since $[a + b]^+ \leq [a]^+ + [b]^+, \ \forall a, b \in \mathbb{R}$ and by the definition of $\widehat{\xi}_{t,j^*}$ (see Equation (21)) which implies that all its elements are positive, Inequality (24b) holds with probability at least $1 - \delta$ by Corollary 2 and by union bound over $M$, and since that $\|\widehat{g}_{t,i}\|_\infty \leq 1$ and Inequality (24c) holds with probability at least $1 - 6\delta$ by Lemma 20.

**Upper-bound to** $\sum_{t\in[T]} \frac{\mathbb{I}(j_t=j^*)}{w_{t,j^*}+\gamma} \sum_{i\in[m]} \left[ (g_i^\circ - \mathbb{E}[g_{t,i}])^\top q_t^{j^*} \right]^+$.

It is immediate to bound the quantity of interest employing the definition of corruption $C$ and by Lemma 18. Indeed, with probability at least $1 - \delta$:

$$\sum_{t\in[T]} \frac{\mathbb{I}(j_t=j^*)}{w_{t,j^*}+\gamma} \sum_{i\in[m]} \left[ (g_i^\circ - \mathbb{E}[g_{t,i}])^\top q_t^{j^*} \right]^+ \leq Lm\sqrt{2T\ln\left(\frac{1}{\delta}\right)} + mC.$$

**Upper-bound to** $\sum_{t\in[T]} \frac{\mathbb{I}(j_t=j^*)}{w_{t,j^*}+\gamma} \sum_{i\in[m]} \left[ \mathbb{E}[g_{t,i}]^\top q_t^{j^*} - \alpha_i \right]^+$.

We bound the quantity of interest as follows. With probability at least $1 - 11\delta$, it holds:

$$\sum_{t\in[T]} \frac{\mathbb{I}(j_t=j^*)}{w_{t,j^*}+\gamma} \sum_{i\in[m]} \left[ \mathbb{E}[g_{t,i}]^\top q_t^{j^*} - \alpha_i \right]^+$$

$$\leq m\sqrt{\beta_4 T}\nu_{T,j^*} + m\beta_5\nu_{T,j^*} + 2m\beta_6 C + Lm\frac{\ln\left(\frac{M}{\delta}\right)}{2\gamma}, \quad \text{(25a)}$$

thank to Corollary 5 and Corollary 6 .

**Upper-bound to** $\sum_{t\in[T]} \frac{\mathbb{I}(j_t=j^*)}{w_{t,j^*}+\gamma} \sum_{i\in[m]} \widehat{\xi}_{t,j^*}^\top q_t^{j^*}$.

First, notice that, with probability at least $1 - \delta$, it holds:

$$\sum_{t\in[T]} \frac{\mathbb{I}(j_t=j^*)}{w_{t,j^*}+\gamma} \sum_{i\in[m]} \widehat{\xi}_{t,j^*}^\top q_t^{j^*} - m\sum_{t\in[T]} \mathbb{I}(j_t=j^*)\widehat{\xi}_{t,j^*}^\top q_t^{j^*} \leq L\sqrt{2T\ln\left(\frac{1}{\delta}\right)},$$

where we employed Lemma 18. Now we observe that, with probability at least $1 - \delta$, it holds:

$$\sum_{t=1}^{T} \widehat{\xi}_{t-1,j^*}^\top q_t \mathbb{I}(j_t=j^*) = \sum_{t=1}^{T}\sum_{x,a} \widehat{\xi}_{t-1,j^*}(x,a)q_t^{j^*}(x,a)\mathbb{I}(j_t=j^*)$$

$$\leq \sum_{t=1}^{T}\sum_{x,a} \widehat{\xi}_{t-1,j^*}(x,a)\mathbb{I}_t(x,a)\mathbb{I}(j_t=j^*) + L\sqrt{2T\ln\frac{1}{\delta}}$$

$$= \mathcal{O}\left( \sqrt{|X||A|LT\ln\left(\frac{mMT|X||A|}{\delta}\right)} + \ln(T)|X||A|C + L\sqrt{T\ln\frac{1}{\delta}} \right),$$

where employed the same steps as in the proof of Theorem 2, considering that the counter increases if and only if $\mathbb{I}_t(x,a)\mathbb{I}(j_t=j^*) = 1$.

Combining the previous bounds concludes the proof. $\qquad\square$

### F.3 MAIN RESULTS

In the following, we provide the main results attained by Algorithm 3 in terms of regret and violations. We start providing the regret bound and the related proof.

**Theorem 6.** *If Program (4) instantiated with $\overline{r}$, $\overline{G}$ and $\alpha$ satisfies Slater's condition (Condition 1), then, given any $\delta \in (0,1)$, with probability at least $1 - 30\delta$, Algorithm 3 attains regret $R_T = \mathcal{O}(m^2 L^2 |X|\sqrt{|A|T\log\left(mT|X||A|/\delta\right)}\log(T)^2 + m^2 L|X|^2|A|^2\log(T)^3\log\left(\log(T)/\delta\right) + m^2 L\log(T)^2|X||A|C)$.*

*Proof.* Employing algorithm 3, with probability at least $1 - 14\delta$, it holds:

$$R_T = \sum_{t \in [T]} \bar{r}^\top q^* - \sum_{t \in [T]} \bar{r}^\top q_t$$

$$= \sum_{t \in [T]} \bar{r}^\top (q^* - q_t^{j^*}) + \sum_{t \in [T]} \bar{r}^\top (q_t^{j^*} - q_t)$$

$$= \sqrt{\beta_1 T} \nu_{T,j^*} + \beta_2 \nu_{T,j^*} + 2\beta_3 C + \sum_{t \in [T]} \bar{r}^\top (q_t^{j^*} - q_t) \tag{26a}$$

$$\leq \sqrt{\beta_1 T} \nu_{T,j^*} + \beta_2 \nu_{T,j^*} + 2\beta_3 C + 2C - \frac{Lm+1}{\rho} \widehat{V}_T + \frac{Lm+1}{\rho} \widehat{V}_{T,j^*}$$

$$- \left(\sqrt{\beta_1} + \frac{m(Lm+1)}{\rho} \sqrt{\beta_4}\right) \sqrt{T} \nu_{T,j^*} - \left(\beta_2 + \frac{m(mL+1)}{\rho} \beta_5\right) \nu_{T,j^*}$$

$$+ \mathcal{O}\left(\frac{M \ln T}{\eta} + \eta\, m^4 L^4 TM + \eta\, M \ln(T) m^4 L^2 \left(\beta_2^2 + \beta_5^2\right)\right.$$

$$\left. + \eta T(\beta_1 + L^2 m^4 \beta_4) M \log(T) + \gamma TLM + L\sqrt{T \ln(1/\delta)} + \frac{Lm}{\gamma} \ln(1/\delta)\right). \tag{26b}$$

where Inequality (26a) hold with probability at least $1 - 11\delta$ by Corollary 7, Inequality (26b) holds with probability at least $1 - 3\delta$ thanks to Lemma 14 and to the following reasoning, which holds with probability at least $1 - \delta$:

$$\sum_{t \in [T]} \bar{r}^\top (q_t^{j^*} - q_t) = \sum_{t \in [T]} (\bar{r} - \mathbb{E}[r_t])^\top (q_t^{j^*} - q_t) + \sum_{t \in [T]} \mathbb{E}[r_t]^\top (q_t^{j^*} - q_t)$$

$$\leq \sum_{t \in [T]} \|\bar{r} - \mathbb{E}[r_t]\|_1 + \sum_{t \in [T]} \mathbb{E}[r_t]^\top \left(q_t^{j^*} - q_t\right) \tag{27a}$$

$$\leq 2C + \sum_{t \in [T]} \mathbb{E}[r_t]^\top \left(q_t^{j^*} - q_t\right) \tag{27b}$$

$$\leq 2C + \sum_{t \in [T]} \mathbb{E}[r_t]^\top q_t^{j^*} - \sum_{t \in [T]} \sum_{j \in [M]} w_{t,j} \mathbb{E}[r_t]^\top q_t^j + L\sqrt{2T \ln(1/\delta)} \tag{27c}$$

where Inequality (27a) holds since $|q_t(x,a) - q_t^{j^*}(x,a)| \leq 1$, $\forall(x,a) \in X \times A$, where Inequality (27b) holds by definition of $C$, and where Inequality (27c) use Azuma-Hoeffding inequality.

We can apply Lemma 16 to bound $\widehat{V}_{T,j^*}$ with high probability. In fact we observe that with probability at least $1 - 16\delta$, it holds:

$$\frac{Lm+1}{\rho} \widehat{V}_{T,j^*}$$

$$\leq \mathcal{O}\left(m^2 L^2 |X| \sqrt{|A|T \ln\left(\frac{mMT|X||A|}{\delta}\right)} + m^2 L \beta_6 C + m^2 L \ln(T)|X||A|C + L^2 m^2 \frac{\ln\left(\frac{M}{\delta}\right)}{2\gamma}\right)$$

$$+ \frac{(Lm+1)m}{\rho} \beta_5 \nu_{T,j^*} + \frac{m(Lm+1)}{\rho} \sqrt{\beta_4 T} \nu_{T,j^*}.$$

Finally, combining the previous results and by Union Bound, with probability at least $1 - 30\delta$, it holds:

$$R_T + \frac{Lm+1}{\rho} \widehat{V}_T$$

$$\leq \mathcal{O}\left(\frac{M \ln T}{\eta} + \eta\, m^4 L^4 TM + \eta\, M \ln(T) m^4 L^2 (\beta_2^2 + \beta_5^2) + \eta T(\beta_1 + L^2 m^4 \beta_4) M \log(T)\right.$$

$$+ \gamma TLM + L\sqrt{T \ln(1/\delta)} + \frac{Lm}{\gamma} \ln(1/\delta)$$

$$\left. + m^2 L^2 |X| \sqrt{|A|T \ln\left(\frac{mMT|X||A|}{\delta}\right)} + mL\beta_6 C + \beta_3 C + m^2 L|X||A| \ln(T)C\right) \tag{28}$$

which concludes the proof after observing that $\widehat{V}_T \geq 0$, by definition, and setting $\gamma = \sqrt{\frac{\ln(M/\delta)}{TM}}$, $\eta \leq \frac{1}{2\Lambda m\left(\sqrt{\beta_1 T} + \beta_2 + \beta_5 + \sqrt{\beta_4 T}\right)}$. $\qquad\square$

We conclude the section providing the violations bound and the related proof.

**Theorem 5.** *If Program (4) instantiated with $\overline{r}$, $\overline{G}$ and $\alpha$ satisfies Slater's condition (Condition 1), then, given any $\delta \in (0,1)$, with probability at least $1 - 34\delta$, Algorithm 3 attains violation $V_T = \mathcal{O}(m^2 L^2 |X| \sqrt{|A|T \log{(mT|X||A|/\delta)}} \log(T)^2 + m^2 L |X|^2 |A|^2 \log(T)^3 \log{(\log(T)/\delta)} + m^2 L \log(T)^2 |X||A|C).$*

*Proof.* Starting from Inequality (28), in order to obtain the final violations bound, it is necessary to find an upper bound for $-R_T$. We proceed as follows,

$$\overline{r}^\top q^* = \mathrm{OPT}_{\overline{r}, \overline{G}, \alpha} \tag{29a}$$

$$= \max_{q \in \Delta(P)} \left(\overline{r}^\top q - \frac{L}{\rho} \sum_{i \in [m]} \left[\overline{G}_i^\top q - \alpha_i\right]^+\right) \tag{29b}$$

$$\geq \overline{r}^\top q_t - \frac{L}{\rho} \sum_{i \in [m]} \left[\overline{G}_i^\top q_t - \alpha_i\right]^+,$$

where Equality (29a) holds since $q^*$ is the feasible occupancy that maximizes the reward vector $\overline{r}$ and Equality (29b) holds by Theorem 4 . This implies $\overline{r}^\top q_t - \overline{r}^\top q^* \leq \frac{L}{\rho} \sum_{i \in [m]} \left[\overline{G}_i^\top q_t - \alpha_i\right]^+$. Moreover, it holds:

$$\sum_{t \in [T]} \sum_{i \in [m]} \left[\overline{G}_i^\top q_t - \alpha_i\right]^+$$

$$\leq \sum_{t \in [T]} \left(\sum_{i \in [m]} \left[\mathbb{E}[g_{t,i}]^\top q_t - \alpha_i\right]^+ + \sum_{i \in [m]} \left[(\overline{G}_i - \mathbb{E}[g_{t,i}])^\top q_t\right]^+\right) \tag{30a}$$

$$\leq \sum_{t \in [T]} \left(\sum_{i \in [m]} \left[\mathbb{E}[g_{t,i}]^\top q_t - \alpha_i\right]^+ + \sum_{i \in [m]} \left\|\overline{G}_i - \mathbb{E}[g_{t,i}]\right\|_1\right) \tag{30b}$$

$$\leq \sum_{t \in [T]} \left(\sum_{i \in [m]} \left[\mathbb{E}[g_{t,i}]^\top q_t - \alpha_i\right]^+ + \sum_{i \in [m]} \left(\left\|\overline{G}_i - g_i^\circ\right\|_1 + \left\|g_i^\circ - \mathbb{E}[g_{t,i}]\right\|_1\right)\right)$$

$$\leq mV_T + 2mC, \tag{30c}$$

where Inequality (30a) holds since $[a + b]^+ \leq [a]^+ + [b]^+, a \in \mathbb{R}, b \in \mathbb{R}$, Inequality (30b) holds since $q_t(x,a) \leq 1 \forall t \in [T], \forall (x,a) \in X \times A$, and finally Inequality (30c) holds by definition of $C$ and $V_T$ and noticing that $m \max_{i \in [m]} a_i \geq \sum_{i \in [m]} a_i, \forall \{a_i\}_{i \in [m]} \subset \mathbb{R}^m$. Thus, combining the previous bounds we lower bound the quantity of interest as follows:

$$R_T + \frac{Lm+1}{\rho} V_T = \sum_{t \in [T]} \mathbb{E}[r_t]^\top (q^* - q_t) + \frac{Lm+1}{\rho} V_T$$

$$= \sum_{t \in [T]} (\mathbb{E}[r_t] - \overline{r})^\top (q^* - q_t) + \sum_{t \in [T]} \overline{r}^\top (q^* - q_t) + \frac{Lm+1}{\rho} V_T$$

$$\geq -\sum_{t \in [T]} \|\mathbb{E}[r_t] - \overline{r}\|_1 + \sum_{t \in [T]} \overline{r}^\top (q^* - q_t) + \frac{Lm+1}{\rho} V_T \tag{31a}$$

$$\geq -2C - \frac{L}{\rho}(mV_T + 2mC) + \frac{Lm+1}{\rho} V_T \tag{31b}$$

$$= -2C - \frac{2LmC}{\rho} + V_T \left(\frac{Lm+1}{\rho} - \frac{Lm}{\rho}\right)$$

$$= \frac{1}{\rho} V_T - \left( 2C + \frac{2LmC}{\rho} \right), \tag{31c}$$

where Inequality (31a) holds since $\underline{v}^\top \underline{w} \geq -\|\underline{v}\|_1 \|\underline{w}\|_\infty, \forall \underline{v}, \underline{w} \in \mathbb{R}^p, p \in \mathbb{N}$, and where Inequality (31b) holds since $\overline{r}^\top (q^* - q_t) \geq -\frac{L}{\rho} \sum_{i \in [m]} \left[ \overline{G}_i^\top q_t - \alpha_i \right]^+ \geq -(mV_T + 2mC)$ and by definition of $C$. Thus, rearranging Inequality (31c), we finally bound the cumulative violation as follows:

$$V_T \leq 2\rho C + 2LmC + \rho R_T + (Lm+1)V_T$$
$$= 2\rho C + 2LmC + (Lm+1)\left( V_T - \widehat{V}_T \right) + \rho \left( R_T + \frac{Lm+1}{\rho} \widehat{V}_T \right)$$
$$\leq \mathcal{O}\left( m^2 L^2 |X| \sqrt{|A|T \ln\left( \frac{mMT|X||A|}{\delta} \right)} + m^2 L \ln(T)|X||A|C + \gamma m T L^2 M \right)$$
$$+ \mathcal{O}\left( R_T + \frac{Lm+1}{\rho} \widehat{V}_T \right),$$

where the last inequality holds by Equation (28) and by Lemma 15, with probability at least $1 - 4\delta$ under $\mathcal{E}_{\widehat{q}}$. Employing a Union Bound, setting $\gamma = \sqrt{\frac{\ln(M/\delta)}{TM}}$ and $\eta \leq \frac{1}{2\Lambda m\left( \sqrt{\beta_1 T} + \beta_2 + \beta_5 + \sqrt{\beta_4 T} \right)}$ concludes the proof. $\square$

# G AUXILIARY LEMMAS FROM EXISTING WORKS

In the following section, we provide useful lemma from existing works.

## G.1 AUXILIARY LEMMAS FOR THE FTRL MASTER ALGORITHM

In the following, we provide the optimization bound attained by the FTRL instance employed by Algorithm 3.

**Lemma 17** (Jin et al. (2024))**.** *The FTRL algorithm over a convex subset $\Omega$ of the $(M-1)$-dimensional simplex $\Delta_M$ :*

$$w_{t+1} = \arg\min_{w \in \Omega} \left\{ \sum_{\tau \in [t]} \ell_\tau^\top w + \frac{1}{\eta} \sum_{j \in [M]} \ln\left( \frac{1}{w_j} \right) \right\},$$

*ensures for all $u \in \Omega$:*

$$\sum_{t \in [T]} \ell_t^\top (w_t - u) \leq \frac{M \ln T}{\eta} + \eta \sum_{t \in [T]} \sum_{j \in [M]} w_{t,j}^2 \ell_{t,j}^2,$$

*as long as $\eta w_{t,j}|\ell_{t,j}| \leq \frac{1}{2}$ for all $t, j$.*

## G.2 AUXILIARY LEMMAS FOR THE OPTIMISTIC LOSS ESTIMATOR

In the following, we provide some results related to the optimistic biased estimator of the loss function. Notice that, given any loss vector $\ell_t \in [0,1]^M$, the following results are provided for $\widehat{\ell}_{t,j} := \frac{\mathbb{I}_t(j)}{w_{t,j}+\gamma_t} \ell_{t,j}$, where $j \in [M]$, $\ell_{t,j}$ is the $j$-th component of the loss vector, $\mathbb{I}_t(j)$ is the indicator functions which is 1 when arm $j$ is played and $\gamma_t$ is defined as in the following lemmas.

**Lemma 18** (Neu (2015))**.** *Let $(\gamma_t)$ be a fixed non-increasing sequence with $\gamma_t \geq 0$ and let $\alpha_{t,j}$ be nonnegative $\mathcal{F}_{t-1}$-measurable random variables satisfying $\alpha_{t,j} \leq 2\gamma_t$ for all $t$ and $j$. Then, with probability at least $1 - \delta$,*

$$\sum_{t \in [T]} \sum_{j \in [M]} \alpha_{t,j} \left( \widehat{\ell}_{t,j} - \ell_{t,j} \right) \leq \ln\left( \frac{1}{\delta} \right).$$

**Corollary 5** (Neu (2015)). *Let $\gamma_t = \gamma \geq 0$ for all t. With probability at least $1 - \delta$,*

$$\sum_{t \in [T]} \left( \widehat{\ell}_{t,j} - \ell_{t,j} \right) \leq \frac{\ln \left( \frac{M}{\delta} \right)}{2\gamma},$$

*simultaneously holds for all $j \in [M]$.*

### G.3 Auxiliary lemmas for the transitions estimation

Next, we introduce *confidence sets* for the transition function of a CMDP, by exploiting suitable concentration bounds for estimated transition probabilities. By letting $M_t(x, a, x')$ be the total number of episodes up to $t \in [T]$ in which $(x, a) \in X \times A$ is visited and the environment transitions to state $x' \in X$, the estimated transition probability at $t$ for $(x, a, x')$ is:

$$\overline{P}_t \left( x' | x, a \right) = \frac{M_t(x, a, x')}{\max \left\{ 1, N_t(x, a) \right\}}.$$

Then, the confidence set for $P$ at episode $t \in [T]$ is defined as:

$$\mathcal{P}_t := \left\{ \widehat{P} : \left| \overline{P}_t(x'|x, a) - \widehat{P}(x'|x, a) \right| \leq \epsilon_t(x'|x, a), \right.$$

$$\left. \forall (x, a, x') \in X_k \times A \times X_{k+1}, k \in [0...L-1] \right\},$$

where $\epsilon_t(x'|x, a)$ is defined as:

$$\epsilon_t(x'|x, a) := 2\sqrt{\frac{\overline{P}_t \left( x' | x, a \right) \ln \left( T |X| |A| / \delta \right)}{\max \left\{ 1, N_t(x, a) - 1 \right\}}} + \frac{14 \ln \left( T |X| |A| / \delta \right)}{3 \max \left\{ 1, N_t(x, a) - 1 \right\}},$$

for some confidence $\delta \in (0, 1)$.

Given the estimated transition function space $\mathcal{P}_t$, the following result can be proved.

**Lemma 19** (Jin et al. (2020)). *With probability at least $1 - 4\delta$, we have $P \in \mathcal{P}_t$ for all $t \in [T]$.*

Notice that we refer to the event $P \in \mathcal{P}_t$ for all $t \in [T]$ as $\mathcal{E}_P$.

We underline that the estimated occupancy measure space by Algorithm 2 is the following:

$$\Delta(\mathcal{P}_t) := \begin{cases} \forall k, & \sum_{x \in X_k, a \in A, x' \in X_{k+1}} q(x, a, x') = 1 \\ \forall k, \forall x, & \sum_{a \in A, x' \in X_{k+1}} q(x, a, x') = \sum_{x' \in X_{k-1}, a \in A} q(x', a, x) \\ \forall k, \forall (x, a, x'), & q(x, a, x') \leq \left[ \overline{P}_t \left( x' | x, a \right) + \epsilon_t \left( x' \mid x, a \right) \right] \sum_{y \in X_{k+1}} q(x, a, y) \\ & q(x, a, x') \geq \left[ \overline{P}_t \left( x' | x, a \right) - \epsilon_t \left( x' \mid x, a \right) \right] \sum_{y \in X_{k+1}} q(x, a, y) \\ & q(x, a, x') \geq 0 \end{cases}$$

To conclude, we restate the result which bounds the cumulative distance between the estimated occupancy measure and the real one.

**Lemma 20** (Jin et al. (2020)). *With probability at least $1 - 6\delta$, for any collection of transition functions $\{P_t^x\}_{x \in X}$ such that $P_t^x \in \mathcal{P}_t$, we have, for all x,*

$$\sum_{t \in [T]} \sum_{x \in X, a \in A} \left| q^{P_t^x, \pi_t}(x, a) - q_t(x, a) \right| \leq \mathcal{O} \left( L |X| \sqrt{|A| T \ln \left( \frac{T |X| |A|}{\delta} \right)} \right).$$

# H AUXILIARY LEMMAS FOR STABILITY

In this section we state the results related to the stability of the arm-algorithms when $C$ is not known. The procedure is inspired by Jin et al. (2024) and Agarwal et al. (2017), but adapted to the case of *Constrained* MDP in high probability. We first give some important definitions. In these definitions we will use $C_t$ as the value of adversarial corruption at episode $t \in [T]$, where $C_t$ is defined as $C_t := \max\{C_t^G, C_t^r\}$, which meets the requirement of upper bounding the adversarial corruption at each considered episode. In addition it holds that $\sum_{t \in [T]} C_t \leq C_r + C_G$ or equivalently $C \leq \sum_{t \in [T]} C_t \leq 2C$, which does not influence the order of the analysis.

**Definition 2.** *A CMDP algorithm is **corruption-robust** if it takes $\theta$ (a guess on the corruption amount) as input, and achieves for any random stopping time $t' \leq T$, whenever $\sum_{t \in [t']} C_t < \theta$:*

$$\sum_{t \in [t']} \overline{r}^\top (q^* - q_t) \leq \sqrt{\beta_1 t'} + (\beta_2 + \beta_3 \theta)\, \mathbb{I}(t' \geq 1),$$

*and*

$$\max_{i \in [m]} \sum_{t \in [t']} \left[ g_{t,i}^\top q_t - \alpha_i \right]^+ \leq \sqrt{\beta_4 t'} + (\beta_5 + \beta_6 \theta)\, \mathbb{I}(t' \geq 1).$$

Notice that Algorithm 2 is corruption-robust after applying a doubling trick to make it work for any stopping time, with probability at least $1 - 9\delta$ thank to Theorem 7 and Theorem 9 Furthermore, we introduce the notion of $\alpha$-stability. An algorithm is considered to be $\alpha$-stable, if its regret under condition imposed by Algorithm 3 is of order $\nu_T^\alpha \cdot \tilde{\mathcal{O}}(R_T)$, where $R_T$ is the upper bound on the regret attained by the algorithm if it receives feedback at each episode. In particular, we are interested in the 1-stability.

**Definition 3.** *An algorithm is 1-**stable** if, under the condition imposed by Algorithm 3, it holds:*

$$\sum_{t \in [T]} \overline{r}^\top (q^* - q_t) \leq \sqrt{\beta_1 T} \nu_{j,T} + \beta_2 \nu_{j,T} + \beta_3 C,$$

*and*

$$\max_{i \in [m]} \sum_{t \in [T]} \left[ g_{t,i}^\top q_t - \alpha_i \right]^+ \leq \sqrt{\beta_4 T} \nu_{j,T} + \beta_5 \nu_{j,T} + \beta_6 C.$$

We can use the procedure defined by Algorithm 4 - and originally proposed by Jin et al. (2024) - to transform a generic corruption robust algorithm to a 1-stable algorithm. Differently from Jin et al. (2024), in our setting, we use the natural symmetry between regret and positive cumulative constraints violation to stabilize both the regret and the positive cumulative constraints violation. We have a different bound for $C_t$ (value of adversarial corruption at episode $t$): indeed, $C_t \leq \max\{\|\mathbb{E}[r_t] - r^\circ\|_1, \max_{i \in [m]} \|\mathbb{E}[g_{t,i}] - g_i^\circ\|_1\}$ is bounded by $|X||A|$. Finally, we are interested in obtaining results that hold in high probability rather than in expectation. To do so, we focus on 1-stability guarantee rather than $1/2$-stability as in Jin et al. (2024) since removing the expectation prevents us from achieving the result above with lower coefficients. We can state the following result.

**Lemma 21.** *Given an algorithm which is corruption robust according to Definition 2 with parameters $(\beta_1, \beta_2, \beta_3, \beta_4, \beta_5, \beta_6)$ and $\beta_1 \geq \mathcal{O}(L^2 \log(T/\delta))$, $\beta_4 \geq \mathcal{O}(L^2 \log(T/\delta))$, with probability at least $1 - p$ with $p \in (0,1)$, then, it is possible convert it to an 1-stable algorithm with probability at least $1 - p - 2\delta$ according to Definition 3 with parameters $(\beta_1', \beta_2', \beta_3', \beta_4', \beta_5', \beta_6')$ as $\beta_1' = \mathcal{O}(\beta_1)$, $\beta_2' = \mathcal{O}(\beta_2 + \beta_3 |X||A| \log(\log(T)/\delta))$, $\beta_3' = \mathcal{O}(\beta_3 \log(T))$, $\beta_4' = \mathcal{O}(\beta_4)$, $\beta_5' = \mathcal{O}(\beta_5 + \beta_6 |X||A| \log(\log(T)/\delta))$, $\beta_6' = \mathcal{O}(\beta_6 \log(T))$, employing Algorithm 4.*

*Proof.* Suppose Algorithm 4 is initialized with the true value of adversarial corruption $C$. We will first prove the result for the regret. We will start by considering a generic instance algorithm $k \in [M]$. Define the quantity $d_{t,k} = \mathbb{I}(w_t \in (2^{-k-1}, 2^{-k}])$ and $h_{t,k} = \mathbb{I}(\text{Instance } k \text{ receives feedback at episode } t)$. We observe that with probability at least $1 - \left( p + \mathbb{P}\left( \bigcup_{k \in [\log(T)]} \{ \sum_{t \in [T]} C_t d_{t,k} h_{t,k} > \theta_k \} \right) \right)$ it holds:

$$\sum_{t \in [T]} \overline{r}^\top (q^* - q_t) d_{t,k} h_{t,k} \leq \sqrt{\beta_1 \sum_{t \in [T]} d_{t,k} h_{t,k}} + (\beta_2 + \beta_3 \theta) \max_{t \in [T]} d_{t,k},$$

by the corruption-robust property of instance $k$. We study now the quantity $\mathbb{P}\left(\bigcup_{k\in[M]}\{\sum_{t\in[T]}C_t d_{t,k}h_{t,k} > \theta_k\}\right)$. Notice that $\mathbb{E}[h_{t,k}|d_{t,k}] = 2^{-k-1}d_{t,k}$, and since $d_{t,k}$ is an indicator function then $\mathbb{E}[h_{t,k}|d_{t,k}]d_{t,k} = \mathbb{E}[h_{t,k}|d_{t,k}]$. In addition, since $\sum_{t\in[T]}C_t \le 2C$, it holds:

$$\sum_{t\in[T]} C_t \mathbb{E}[h_{t,k}|d_{t,k}]d_{t,k} = 2^{-k-1}\sum_{t\in[T]} C_t d_{t,k} \le 2^{-k}C,$$

and with probability at least $1 - \delta/\log(T)$ noticing that $M = \log(T)$:

$$\sum_{t\in[T]} C_t d_{t,k}h_{t,k} - \sum_{t\in[T]} C_t \mathbb{E}[h_{t,k}|d_{t,k}]d_{t,k}$$

$$\le 2\sqrt{\sum_{t\in[T]} C_t^2 d_{t,k}\mathbb{E}[h_{t,k}|d_{t,k}]\log\left(\frac{\log(T)}{\delta}\right)} + |X||A|\log\left(\frac{\log(T)}{\delta}\right) \tag{32a}$$

$$\le 2\sqrt{|X||A|\sum_{t\in[T]} C_t d_{t,k}\mathbb{E}[h_{t,k}|d_{t,k}]\log\left(\frac{\log(T)}{\delta}\right)} + |X||A|\log\left(\frac{\log(T)}{\delta}\right) \tag{32b}$$

$$\le \sum_{t\in[T]} C_t \mathbb{E}[h_{t,k}|d_{t,k}]d_{t,k} + 2|X||A|\log\left(\frac{\log(T)}{\delta}\right), \tag{32c}$$

where Inequality (32a) holds with probability at least $1 - \delta/\log(T)$ by Freedman inequality, Inequality (32b) holds since $C_t \le |X||A|$, and Inequality (32c) holds by AM-GM inequality. Therefore, it holds simultaneously for all $k \in [M]$:

$$\sum_{t\in[T]} C_t d_{t,k}h_{t,k} \le 2\sum_{t\in[T]} C_t \mathbb{E}[h_{t,k}|d_{t,k}]d_{t,k} + 2|X||A|\log\left(\frac{\log(T)}{\delta}\right)$$

$$\le 2^{-k+1}C + 2|X||A|\log\left(\frac{\log(T)}{\delta}\right) = \theta_k,$$

with probability at least $1 - \delta$, so $\mathbb{P}\left(\bigcup_{k\in[M]}\{\sum_{t\in[T]}C_t d_{t,k}h_{t,k} > \theta_k\}\right) \le \delta$. Moreover, notice that with probability at least $1 - p - 2\delta$ thanks to the definition of corruption robust and Azuma-Hoeffding inequality, it holds simultaneously for all $k$:

$$\sum_{t\in[T]} \bar{r}^\top (q^* - q_t)d_{t,k}$$

$$= \frac{1}{2^{-k-1}}\sum_{t\in[T]} \bar{r}^\top (q^* - q_t)2^{-k-1}d_{t,k}$$

$$= \frac{1}{2^{-k-1}}\sum_{t\in[T]} \bar{r}^\top (q^* - q_t)d_{t,k}\mathbb{E}[h_{t,k} \mid d_{t,k}]$$

$$= \frac{1}{2^{-k-1}}\left(\sum_{t\in[T]} \bar{r}^\top (q^* - q_t)d_{t,k}\left(\mathbb{E}[h_{t,k} \mid d_{t,k}] - h_{t,k}\right) + \sum_{t\in[T]} \bar{r}^\top (q^* - q_t)d_{t,k}h_{t,k}\right)$$

$$\le \frac{1}{2^{-k-1}}\left(L\sqrt{2\ln\left(\frac{\log(T)}{\delta}\right)\sum_{t\in[T]} d_{t,k}} + \sqrt{\beta_1 \sum_{t\in[T]} d_{t,k}} + (\beta_2 + \beta_3\theta_k)\max_{t\in[T]} d_{t,k}\right)$$

$$\le \mathcal{O}\left(\frac{1}{2^{-k-1}}\left(\left(\sqrt{\beta_1} + L\sqrt{\log\left(\frac{T}{\delta}\right)}\right)\sqrt{T}\max_{t\in[T]} d_{t,k} + (\beta_2 + \beta_3\theta)\max_{t\in[T]} d_{t,k}\right)\right),$$

noticing that $\mathbb{E}\left[d_{t,k}\left(\mathbb{E}\left[h_{t,k}|d_{t,k}\right] - h_{t,k}\right)\right] = \mathbb{E}\left[h_{t,k}|d_{t,k}\right] - \mathbb{E}[h_{t,k}]d_{t,k} = \mathbb{E}\left[h_{t,k}|d_{t,k}\right] - \mathbb{E}\left[h_{t,k}|d_{t,k}\right] = 0$, since the expectation is taken w.r.t. the randomization of Algorithm 4 and the distribution generated given the external probability of receiving feedback $w_t$.

To conclude with probability at least $1 - p - 2\delta$:

$$\sum_{t\in[T]} \overline{r}^\top (q^* - q_t) \mathbb{I}\left(w_t \geq \frac{1}{T}\right)$$

$$\leq \sum_{k\in[M]} \sum_{t\in[T]} \overline{r}^\top (q^* - q_t) d_{t,k}$$

$$\leq \mathcal{O}\left(\sqrt{\beta_1 T} \max_{t\in[T]} \frac{1}{w_t} + (\beta_2 + \beta_3 |X||A| \log(\log(T)/\delta)) \max_{t\in[T]} \frac{1}{w_t} + \beta_3 \log(T) C\right)$$

$$\leq \mathcal{O}\left(\left(\sqrt{\beta_1' T} + \beta_2'\right) \nu_T + \beta_3' C\right),$$

with $\sqrt{\beta_1} \geq \mathcal{O}(L\sqrt{\log(T/\delta)})$. Notice that the analogous reasoning can be applied to the positive cumulative constraints violation with parameters $\beta_4, \beta_5, \beta_6$. □

---

**Algorithm 4** Adapted `STABILIZE` Jin et al. (2024)

---

**Require:** $C, \delta \in (0,1)$
1: Initialize $M = \log(T)$ instance of Algorithm 2, each instance $k \in [M]$ initialized with corruption parameter:
$$\theta_k := 2^{-k+1} C + 2|X||A| \log\left(\frac{\log(T)}{\delta}\right)$$

2: **for** $t \in [T]$ **do**
3:     Observe $w_t$, probability of receiving feedback.
4:     **if** $w_t > \frac{1}{T}$ **then**
5:         Let $k_t$ be such that $w_t \in (2^{-k_t-1}, 2^{-k_t}]$
6:         Choose $\pi_t$ as policy proposed by instance $k_t$
7:         If the algorithm receives feedback send it to instance $k_t$ with probability $\frac{2^{-k_t-1}}{w_t}$
8:     **if** $w_t \leq \frac{1}{T}$ **then**
9:         Propose random policy $\pi_t$

---

**Corollary 6.** *Being $j^*$ such that $C \in (2^{j^*-1}, 2^{j^*}]$ then with probability at least $1 - 11\delta$ it holds:*

$$\max_{i\in[m]} \sum_{t\in[T]} \left[\mathbb{E}[g_{t,i}]^\top q_t^{j^*} - \alpha_i\right]^+ \leq \sqrt{\beta_4 T} \nu_{T,j^*} + \beta_5 \nu_{T,j^*} + 2\beta_6 C,$$

*with $\sqrt{\beta_4} = \mathcal{O}\left(L|X|\sqrt{|A| \ln(mT|X||A|/\delta)}\right)$, $\beta_5 = \mathcal{O}\left(|X|^2 |A|^2 \log(T) \log(\log(T)/\delta)\right)$ and $\beta_6 = \mathcal{O}\left(\ln(T)^2 |X||A|\right)$.*

**Corollary 7.** *Being $j^*$ such that $C \in (2^{j^*-1}, 2^{j^*}]$ then with probability at least $1 - 11\delta$ it holds:*

$$\sum_{t\in[T]} \overline{r}^\top (q^* - q_t^{j^*}) \leq \sqrt{\beta_1 T} \nu_{T,j^*} + \beta_2 \nu_{T,j^*} + 2\beta_3 C,$$

*where $\sqrt{\beta_1} = \mathcal{O}\left(L|X|\sqrt{|A| \ln(T|X||A|/\delta)}\right)$, $\beta_2 = \mathcal{O}\left(|X|^2 |A|^2 \log(T) \log(\log(T)/\delta)\right)$ and $\beta_3 = \mathcal{O}\left(\ln(T)^2 |X||A|\right)$.*

