# OpenReview forum: "Learning Constrained Markov Decision Processes With Non-stationary Rewards and Constraints"
_ICLR.cc/2025/Conference — Submitted to ICLR 2025_

### Official Review · Reviewer_ba5a · 2024-10-28

**Soundness:** 3
**Presentation:** 3
**Contribution:** 2
**Rating:** 5
**Confidence:** 4

**Summary:**

This paper studies the constrained Markov decision processes (CMDPs) with adversarial rewards and constraints. Given the negative result of Mannor et al., 2009, the authors propose algorithms whose regret bounds depend on the non-stationarity of rewards and constraints. The proposed algorithm works for unknown $C$ by using a Corral-based technique.

**Strengths:**

+ To my knowledge, This is the first work considering the CMDPs problem with adversarial rewards and constraints.

+ Authors provide theoretical guarantees for the proposed algorithms and achieve a linearly additive dependence on C.

+ The paper is generally well-written and easy to follow.

**Weaknesses:**

My primary concern is the limited novelty of the technical contribution, as the approach in this paper appears to be a direct application of techniques from (Jin et al., 2024) for the CMDPs setting. In particular, the core technique and main theoretical analysis to deal with unknown $C$ are largely based on that of (Jin et al., 2024). Given that the corral (Agarwal et al., 2017) with different guesses for $C$ has been used in prior work (e.g., Jin et al., 2024), I believe a more direct comparison to (Jin et al., 2024) is necessary to clarify the technical contributions of this paper.

**Questions:**

If there are unique technical challenges specific to the CMDPs setting that prevent a direct application of the techniques in (Jin et al., 2024), I would be happy to reconsider and adjust my evaluation accordingly.

---

> ### Author Response · Authors · 2024-11-16
>
> >Unique technical challenges specific to the CMDPs setting that prevent a direct application of the techniques  (Jin et al., 2024)
>
> Both [Jin et al., 2024] and [Agarwal et al., 2017] deal with unconstrained problems, so their goal is single-objective: minimizing the regret. In our setting, minimizing the regret is not the only objective, since we want to keep the violations as small as possible. Thus, the master algorithm must be significantly modified. It is necessary to introduce a Lagrangian formulation of the problem which, in turn, requires a careful analysis of the "optimal" Lagrangian multiplier magnitude, namely, the one that guarantees that
> high rewards could not compensate high violations (e.g., see Section F.1). This is of paramount importance for our specific setting, since we study the positive constraints violation, not allowing for cancellation.
>
> Finally, consider that we prove regret and positive constraints violation bounds in high probability, while both the original paper of Corral [Agarwal et al., 2017] and [Jin et al., 2024] give results only in expectation (e.g., see Section H).
>
> Since these are crucial aspect of our work, please let us know if further discussion is necessary.

---

### Official Review · Reviewer_573V · 2024-10-29

**Soundness:** 3
**Presentation:** 3
**Contribution:** 2
**Rating:** 5
**Confidence:** 3

**Summary:**

This paper considers constrained MDP under non-stationary rewards and constraints, where the reward, cost functions and the transition kernel are unknown, the level of non-stationarity is measured by $C$ and the constraints is in terms of the expected number of violations. It solves the problem with the knowledge of $C$, followed by an extension to the unknown $C$ case with the help of Corralling method by Agarwal et al. (2017).

**Strengths:**

- The paper is easy to follow and the intuition of the algorithms is natural.
- This paper adopts $C$ to quantify the level of non-stationarity, bridging the stationary setup and the adversarial setup.
- A meta algorithm is used to deal with the case where the knowledge of $C$ is not provided.

**Weaknesses:**

- Given the previous literature (Efroni et al. (2020), Stradi et al. (2024a), Stradi et al. (2024b), Wei et al. (2023)), the proposed problem formulation lacks novelty and motivations. While considering a positive violation seems to be practical, it is more proper to consider a dynamic baseline. Therefore, it is encouraged to provide a more detailed comparison between this work and the literatures in terms of the setups, motivations, and results.
- The proposed algorithm follows the UCB design with $C$ involved, thus, achieving a sublinear regret with respect to the static policy is kind of expected. It would be great if the authors can highlight the technical novelty.
- Simulations are not provided to illustrate the efficacy of the proposed algorithms, as well as comparisons with other baseline algorithms in the literature.

**Questions:**

- Can the authors provide the motivation for considering a static baseline algorithm? When the time horizon $L$ is large, minimizing the regret in the current episode seems to be a natural choice. Thus, the dynamic regret seems to be more meaningful under this nonstationary setup.
- The paper suggests it can ease the negative result of Mannor et al. (2019) by introducing the nonstationary parameter $C$. Is there any lower bound that involves $C$ for the proposed problem under the proposed regret formulation?
- How efficient it is to solve the optimization problem in Line 5 of Algorithm 2?

**Details Of Ethics Concerns:**

None. A theoretical paper without any ethics concerns.

---

> ### Author Response · Authors · 2024-11-16
>
> >Comparison with literature
>
> * Comparison with [Efroni et al. (2020)]
>
> The algorithm presented by [Efroni et al., 2020] works for stochastic environments: even when $C$ is a constant different from $0$, the confidence interval built by the algorithm might exclude the optimal solution and therefore lead to a linear regret.
>
> * Comparison with [Stradi et al. (2024a)]
>
> [Stradi et al. (2024a)] study the case where rewards are adversarial **but constraints are stochastic**. Therefore, the same reasoning above holds.
>
> * Comparison with [Stradi et al. (2024b)]
>
> [Stradi et al. (2024b)] study the case where both rewards and constraints can be adversarial in principle. However, the algorithm presented attains linear regret whenever both the non-stationarity of the rewards and the non-stationarity on the constraints is different from zero. Moreover, notice that in our work we consider  positive constraints violation while [Stradi et al. (2024b)] allows for cancellation. Finally, [Stradi et al. (2024b)] deals with full feedback while the algorithms presented in this paper work under bandit feedback.
>
>  * Comparison with [Wei et al. (2023)]
>
>  [Wei et al. (2023)] study a similar problem to ours.
>  Nonetheless, please notice the following fundamental differences. First, we consider the harder problem of bounding the positive constraint violation, while they allow for cancellations. As concerns the definition of regret, they employ a dynamic regret baseline, which, in general, is harder than the static regret employed in our work. However, they compare learner’s performances against a dynamic policy that satisfies the constraints at every round. Instead, we consider a policy that satisfies the constraints on average, which can perform arbitrarily better than a policy satisfying the constraints at every round. Furthermore, the dependence on $T$ in their regret bound is much worse than ours, even when the non-stationarity is small, namely, when it is a constant independent of $T$ and thus the dynamic regret definition collapse to the static one. Indeed, when the variation budget is not known the regret is of order $\widetilde{\mathcal{O}}(T^{\frac{7}{8}})$. Finally, notice that the regret guarantees presented by [Wei et al. (2023)] hold only under the assumption of large $T$ while we make no assumption on the number of episodes.
>
> >The proposed algorithm follows the UCB design with $C$ involved, thus, achieving a sublinear regret with respect to the static policy is kind of expected. It would be great if the authors can highlight the technical novelty.
>
>  We agree with the Reviewer that the techniques employed in the known $C$ setting are UCB-like; nonetheless, they are indeed properly adapted to the uncertainty of the environment. While enlarged confidence set are not novel in literature, ours present peculiar characteristic due to our choice of baseline. Moreover, we want to stress that the algorithm developed for the known $C$ setting is mostly presented as a necessary intermediate step to tackle the considerably more challenging problem of dealing with unknown $C$ setting, **which makes standard techniques like UCB inapplicable**.
>
> Please let us know if further discussion is necessary.

---

> ### Author Response · Authors · 2024-11-16
>
> >Can the authors provide the motivation for considering a static baseline algorithm? When the time horizon $L$ is large, minimizing the regret in the current episode seems to be a natural choice. Thus, the dynamic regret seems to be more meaningful under this nonstationary setup.
>
> As concerns the choice of a static baseline, the goal of our paper was to bridge the existent gap in literature between the stochastic world and the adversarial world in CMDP. Both extremes (stochastic and adversarial) are traditionally studied through the employ of a static baseline and therefore the use of a static baseline permits the immediate interpretation of the interpolation between this two cases. In addition, for how we defined the measure of non-stationary, our works fits more closely with the field of corruption robustness (e.g. [1],[2],[3],[4]).
>
> Nonetheless, since the difference between the reward achievable with a static optimal policy and the reward achievable through a dynamic optimal policy is strongly dependent on the corruption, in principle our algorithm should achieve also a dynamic regret of order $\sqrt{T}+C$. To give an intuition of why, we can observe that for all episodes $OPT\_{d,t}=\max\_{q \in \Delta(G^\circ)} \mathbb{E}[r_t]^\top q= \max\_{q \in \Delta(G^\circ)} (r^\circ{}^\top q + (\mathbb{E}[r_t]- r^\circ{})^\top q) \le \max\_{q \in \Delta(G^\circ)} (r^\circ{}^\top q + \|\|\mathbb{E}[r_t]-r^\circ\|\|\_1\|\|q\|\|\_\infty) \le OPT_s + C\_t^r$, where $\Delta(G^\circ)$ is the space of the feasible occupancy measures that respect the uncorrupted constraints.
>
> Moreover, notice that, when $L$ is large enough, any regret bound for episodic MDPs become vacuous due to the existing lower-bound of order $\Omega(\sqrt{LT})$ (see e.g. [5]).
>
> [1] ”A model selection approach for corruption robust reinforcement learning”, Wei et al.
> (2022)
>
> [2]”Noregret online reinforcement learning with adversarial losses and transitions”, Jin et al.
> (2024)
>
> [3], ”Corruption robust exploration in episodic reinforcement learning”, Lykouris et al. (2019)
>
> [4], ”Improved corruption robust algorithms for episodic reinforcement learning”, Chen et al.
> (2021)
>
> [5] "Minimax regret bounds for reinforcement learning", Azar et al. (2017)
>
> >Lower bound involving $C$
>
> To find a lower bound on the minmax regret dependent on $C$, we start from the following known lower bounds: the one in case of completely stochastic rewards and constraints and the one in the worst case of adversarial rewards and constraints. In the stochastic setting (namely, when $C=0$) the instance independent lower bound for both $R_T, V_T$ is of order $\Omega(\sqrt{T})$ making the first element of our upper bound $\mathcal{O}(\sqrt{T}+C)$ tight. On the other end, suppose by absurd that a regret of type $(\sqrt{T}+C^\alpha)$ was possible, with $\alpha \in (\frac{1}{2},1)$. Then, since $C$ is at most linear by definition, we would have that the regret would be $\mathcal{O}(T^\alpha)$ for all possible choice of adversary, violating the impossibility result of [Mannor et al. 2009] in the adversarial setting.
>
> >How efficient it is to solve the optimization problem in Line 5 of Algorithm 2?
>
> The linear optimization problem can be efficiently solved in  $Poly(m,|X|^2|A|)$. This is possible through the use of ellipsoid method (see e.g., ["Introduction to linear optimization", Bertsimas and Tsitsiklis, 1997]).

---

> > ### Comment · Reviewer_573V · 2024-11-30
> >
> > Thanks the authors for the response! Some of my concerns have been properly resolved. The intention that this paper wishes to formulate a problem that can interpolates between the stochastic setup and the adversarial setup is well stated. I revised my evaluation. However,
> >
> > - Regarding the choice of the static baseline, most reviewers posed similar questions. While the intention of the author is perceived, the dynamic regret is often considered in the non-stationary setup which this paper cannot deal with.
> > - Regarding the lower bound, the derived upper bound appears to be compared against two extreme cases: a fully stochastic environment and a fully adversarial one. In the stochastic case, the upper bound is shown to be tight, whereas in the adversarial case, the result adapts to the order of $C$. While this comparison illustrates how the upper bound behaves in these scenarios, the tightness of the result is not justified for environments that lie “between” the stochastic and adversarial extremes. In other words, although the upper bound effectively bridges the two extreme cases, the absence of a corresponding lower bound result limits the paper’s overall contribution.
> > - Another question: Given that the rewards and constraints are stochastically generated, if $C=\Omega(T)$ in this paper, is the problem equivalent to the problem under the adversarial case? It seems that it is still more benign than the adversarial case due to the stochastic nature.

---

> > > ### Author Response · Authors · 2024-12-01
> > >
> > > We thank the Reviewer for the revision! We hope that the following responses to the Reviewer’s remaining concerns will contribute to a further positive evaluation.
> > >
> > > >Another question: Given that the rewards and constraints are stochastically generated, if $C=\Omega(T)$ in this paper, is the problem equivalent to the problem under the adversarial case? It seems that it is still more benign than the adversarial case due to the stochastic nature.
> > >
> > >  Please notice that in our setting, at each episode an adversary chooses a distribution on the rewards and costs with no further assumption on the distribution type (a part from the boundedness). Therefore, the adversarial setting, as traditionally defined, is a particular case of our corrupted setting where the adversary chooses a (Dirac) delta distribution at each episode, with all probability mass on the exact value the adversary wants the rewards and the cost to assume at each episode. A different interpretation can be that the adversarial setting is equivalent to ours when the expected values of the rewards and costs distributions are revealed at the end of the episode, which is a setting with a more powerful feedback than the one studied in our work.
> > >
> > > Thus, even if our setting maintains a stochastic component, the lower bound in the worst case cannot be better than the lower bound of the adversarial setting, i.e., linear, **since our setting is at least as hard as the adversarial one when $C=\Omega(T)$**.
> > >
> > > > However, Regarding the choice of the static baseline, most reviewers posed similar questions. While the intention of the author is perceived, the dynamic regret is often considered in the non-stationary setup which this paper cannot deal with.
> > >
> > > We agree with the Reviewer that the problem of studying the dynamic regret is surely an interesting topic that still presents open questions and room for novelty. However, our work is mainly related to the corruption robust literature that we believe is an interesting topic too (e.g. [1],[2],[3],[4],[5],[6],[7],[8],[9],[10]), in addition to being uncharted territory in the online CMDPs research area. Moreover, our result contributes to bridge a preexistent research gap on the connection between learning in stochastic and adversarial constrained setting, which we believe should be evaluated positively.
> > >
> > > To summarize, while we agree that dynamic regret guarantees are of interest for the community, we believe that, since the corruption robust literature is largely appreciated in the online learning field (see previous references), our work should be of interest, too.
> > >
> > > Finally, if the Reviewer believe it could be of help for future readers, **we would be happy to change the title of the paper to make a direct reference to corruption robustness instead of non-stationary to delimit and frame our work for what it is, and for what it is meant to do, solving the ambiguity with the stochastic non-stationary research line traditionally approached with dynamic regret**.

---

> > > ### Author Response · Authors · 2024-12-01
> > >
> > > >In the stochastic case, the upper bound is shown to be tight,
> > >     whereas in the adversarial case, the result adapts to the order of C. While this comparison
> > >     illustrates how the upper bound behaves in these scenarios, the tightness of the result is not
> > >     justified for environments that lie “between” the stochastic and adversarial extremes. In other
> > >     words, although the upper bound effectively bridges the two extreme cases, the absence of a
> > >     corresponding lower bound result limits the paper’s overall contribution.
> > >
> > > We agree with the Reviewer that, while the lower bound for the extremes are meaningful, there is not a formal lower bound for the in-between cases. However, in the corruption robust literature, an upper bound of order $\sqrt{T} + C$ is a state-of-the-art result.
> > >
> > > While the problem of studying corrupted CMDPs is fairly unexplored, and to the best of our knowledge we are the first to present this type of result for CMDPs, we can draw a parallelism between our work and the literature of corruption-robust unconstrained MDP with corrupted transitions. Indeed, in the easiest scenario of such a setting, that is, MDPs with stochastic rewards and transition, there exists a lower bound of $\sqrt{T}$, while for the adversarial rewards and transitions case there exists linear lower bound similar to the one for adversarial rewards and cost in CMDPs.
> > > Thus, to evaluate the goodness of our results we believe we should confront our paper with the results obtained in the literature of MDPs with corrupted transitions.
> > > Up to recently, the best result in this field showed an upper bound on the regret which is multiplicative in the corruption, i.e. $C\sqrt{T}$ (e.g. [1]).
> > > The first work to obtain a regret with an additive dependence on the corruption (i.e. dependence of type $\sqrt{T}+C^\alpha$ instead of $C\sqrt{T}$ ) is [2] with a regret of order $\sqrt{T}+C^2$.
> > > This result has been improved first by [3] and later by [4] obtaining a regret of order $\sqrt{T}+C$ with a corruption definition and a baseline which is the analogous of our case.
> > >
> > > To the best of our knowledge, the dependence $\sqrt{T}+C$ is considered state of the art for the MDPs with corrupted transitions and for other similar settings (see, [8],[9],[10]) and commonly accepted as an optimal result.
> > > For what concern corrupted CMDPs, to the best of our knowledge, there are no works that achieves even a multiplicative dependence on the corruption, namely, a bound of order $C\sqrt{T}$.
> > >
> > > Therefore, while we agree with the Reviewer that an holistic lower bound on the dependence of $C$ would be of great value, we underline that in the field, a regret bound of order $\sqrt{T}+C$ is commonly accepted as state of the art result (see previous references).
> > >
> > > [1] Lykouris et al. "Corruption robust exploration in episodic reinforcement learning" (2021)
> > >
> > > [2] Chen et al. "Improved Corruption Robust Algorithms for Episodic Reinforcement Learning" (2021)
> > >
> > > [3] Wei et al. "A Model Selection Approach for Corruption Robust Reinforcement Learning" (2022)
> > >
> > > [4] Jin et al. "Noregret online reinforcement learning with adversarial losses and transitions" (2024)
> > >
> > > [5] Lykouris et al.,"Stochastic bandits robust to adversarial corruption" (2018)
> > >
> > > [6] Yang et al., "Adversarial Bandits with Corruptions: Regret Lower Bound and No-regret Algorithm", (2020)
> > >
> > > [7] Zhao et al. "Linear Contextual Bandits with Adversarial Corruptions", (2021)
> > >
> > > [8] Bogunovic at al, "Stochastic linear bandits robust to adversarial attacks" (2021)
> > >
> > > [9] Lee et al., "Achieving near instance optimality and minimax-optimality in stochastic and adversarial linear bandits simultaneously", (2021)
> > >
> > > [10] He et al. "Nearly Optimal Algorithms for Linear Contextual Bandits with Adversarial Corruptions" (2022)
> > >
> > > Please let us know if you have any further question.

---

### Official Review · Reviewer_o4Qz · 2024-10-29

**Soundness:** 3
**Presentation:** 2
**Contribution:** 3
**Rating:** 6
**Confidence:** 4

**Summary:**

This paper considers CMDPs with both non-stationary rewards and constraints. As the fully adversarial CMDP problem is shown to be statistically intractable, the authors propose to consider the case where the per-round reward $r_t$ / constraint $g_t$ are sampled from some time-dependent distributions whose means do not vary from a "reference" reward / constraint vector significantly.

When $C$ is known, to tackle with the extra uncertainty in rewards and constraints, the confidence intervals are enlarged by an additive factor of $C/N_t(s,a)$. Utilizing policy search over the (estimated) occupancy measures, the algorithm guarantees $\mathcal O(\sqrt T + C)$ regret and constraint violations where $C$ is the level of non-stationarity.

Moreover, when $C$ is unknown, an algorithm-selection meta-algorithm that performs log-barrier FTRL on many base-algorithms with doubling $C$'s is developed.

**Strengths:**

1. The constraint-violation metric is pretty strong: Albeit the static hindsight optimal policy is allowed to violate the constraints in some rounds and make it up in the remaining, the algorithm is not because of the $[\cdot]^+$ notion.
2. The known $C$ algorithm is pretty intuitive and well-motivated, especially regarding the enlargement of confidence radii.
3. The algorithm also works well when $C$ is unknown, with almost no performance degeneration -- though its quite hard to get the idea and applicability of the framework from the current writing; see the Weaknesses section.

**Weaknesses:**

1. The algorithmic definitions, especially the $\ell_{t,j}$ and $b_{t,j}$ are written in a highly technical way. For example, without any description on the $\beta$'s in the main text, it is almost impossible to understand the construction of $b_{t,j}$ -- I suggest the authors to exemplify what each $\beta$ would be like if the revised Algorithm 2 is executed on its own. The effect of $\nu_{t,j}$ is also not explained.
2. The authors mentioned that the meta-algorithm Lag-FTRL can be of independent interest. Would it be possible to isolate the framework from the base algorithm -- say, what conditions of $\ell_t$ and $b_t$ are required for the framework to work, or it has to be constructed so meticulously as in Eqs. (6) and (7)?

Minor: Line 155 -- the definition of $C$ should be $\max(C_{G^\circ},C_{r^\circ})$?

**Questions:**

See Weaknesses. These two questions concern the technical hardness and applicability of the Lag-FTRL framework. I am happy to adjust my rating if the answers turn out to be pretty positive.

And also:

3. Is there any intuition why the non-stationarity metric is like this? Would other common metrics like path lengths be harder to tackle?

---

> ### Author Response · Authors · 2024-11-16
>
> >The algorithmic definitions, especially the $\ell_{t,j}$ and $b_{t,j}$ are written in a highly technical way. For example, without any description on the $\beta$'s in the main text, it is almost impossible to understand the construction of $b_{t,j}$ -- I suggest the authors to exemplify what each $\beta$ would be like if the revised Algorithm 2 is executed on its own. The effect of $\nu_{t,j}$ is also not explained.
>
> The role of the bonus quantity is to incentivize the exploration. In this sense, $\nu_{t,j}=\max_{\tau \le t}\frac{1}{w_{\tau,j}}$ is higher for the less explored arms, leading to a bonus quantity that push the algorithm to choose them more. The bonus quantity is also proportional to the upper bound of the regret and positive constraints violation of the arm algorithm, which is represented by the use of the $\beta$'s. Due to space constraints we had to be very concise in explaining this bonus quantity and in giving an intuitive explanation, but we will be glad to expand on these topics in the camera-ready with the additional space.
>
> >Independent interest of Lag-FTRL
>
> As we specified in the previous answer, the problem-specific quantities employed in lag-FTRL can be easily derived observing the regret and constraints guarantees of the employed sub-routines (the ones associated with the arms), leading to a versatile meta-procedure that deals with constrained problem.
>
> >Is there any intuition why the non-stationarity metric is like this? Would other common metrics like path lengths be harder to tackle?
>
>  Our choice of measure of non-stationary is directly inspired by the corruption robustness field (e.g. [1],[2],[3],[4]). In principle, our $C$ and a path length metric are very different, as our metric is considered with respect to a possible fixed distribution equal for all episodes and the cumulative deviation from it. Our choice of $C$ is also a result of our goal of interpolating the result between the completely stochastic case and the completely adversarial one, both traditionally studied through a static prospective, more affine to corruption than to path length. Using the metric of path length would be more meaningful in relation with a dynamic regret problem, which is out of scope for this work .
>
> Since these are crucial aspect of our work, please let us know if further discussion is necessary.
>
> [1] "A model selection approach for corruption
> robust reinforcement learning", Wei et al. (2022)
>
> [2] "Noregret online reinforcement learning with adversarial losses and transitions", Jin et al. (2024)
>
> [3] "Corruption robust exploration in episodic reinforcement learning", Lykouris et al. (2019)
>
> [4] "Improved corruption robust algorithms for episodic reinforcement learning", Chen et al. (2021)

---

### Official Review · Reviewer_iXAY · 2024-11-02

**Soundness:** 3
**Presentation:** 3
**Contribution:** 2
**Rating:** 5
**Confidence:** 3

**Summary:**

The paper studied CMDP with non stationary rewards and transitions. A \sqrt{T} + C style regret bound is shown with LAG-FTRL, which is a combination of FTRL and UCB. The algorithm will maintain several instances of the UCB algorithm and pick one according to FTRL.

**Strengths:**

The paper is well-written and the proof is sound. To the best of my knowledge, this is the first result on CMDP with nonstationary rewards, transitions, and bandit feedback.

**Weaknesses:**

My main concern is about the static baseline employed. To my knowledge, dynamic regret is usually used as a benchmark in learning in nonstationary environments. The paper has only highlighted the importance of considering static regret when the nonstationary is small (as a constant), which is unlikely to be the case in most environments. When we consider static regret, it is not so surprising that adding a layer of FTRL that "guesses" the corruption level can work, as FTRL already guarantees the static regret.

**Questions:**

1. Can you elaborate on the comparison of dynamic and static regret when the nonstationary is large?
2. Can the results be extended to dynamic regret?
3. Can you comment on the technical challenges of deriving the results for static regret?

---

> ### Author Response · Authors · 2024-11-16
>
> >On the reason behind the static baseline
>
> We designed this setting to bridge the existent gap in literature between the stochastic world and the adversarial world in CMDP. Both extremes (stochastic and adversarial) are traditionally studied through the employ of a static baseline and therefore the use of a static baseline permits  the immediate interpretation of the interpolation between this two cases. In addition, for how we defined the measure of non-stationary, our works fits more closely with the field of corruption robustness (e.g. [1],[2],[3],[4]).
>
> [1] "A model selection approach for corruption
> robust reinforcement learning", Wei et al. (2022)
>
> [2]"Noregret online reinforcement learning with adversarial losses and transitions", Jin et al. (2024)
>
> [3] "Corruption robust exploration in episodic reinforcement learning", Lykouris et al. (2019)
>
> [4] "Improved corruption robust algorithms for episodic reinforcement learning", Chen et al. (2021)
>
> >Can the results be extended to dynamic regret?
>
> The difference between the reward achievable with a static optimal policy and the reward achievable through a dynamic optimal policy is strongly dependent on the corruption, therefore in principle our algorithm should achieve also a dynamic regret of order $\sqrt{T}+C$. To give an intuition of why, we can observe that for all episodes $OPT_{d,t}=\max_{q \in \Delta(G^\circ)} \mathbb{E}[r_t]^\top q= \max_{q \in \Delta(G^\circ)} (r^\circ{}^\top q + (\mathbb{E}[r_t]- r^\circ{})^\top q) \le \max\_{q \in \Delta(G^\circ)} (r^\circ{}^\top q + \|\|\mathbb{E}[r_t]-r^\circ\|\|\_1\|\|q\|\|\_\infty) \le OPT_s + C_t^r$, where $\Delta(G^\circ)$ is the space of the feasible occupancy measures that respect the uncorrupted constraints.
>
> Nevertheless we want to stress the fact that our goal was to weaken the linear lower bound for the adversarial rewards and constraints setting (see answer above).
> The problem of learning dynamic regret is in principle very different and for the goal of this paper out of scope.
>
> >Can you comment on the technical challenges of deriving the results for static regret?
>
> We thank the Reviewer for the opportunity to clarify this aspect. In the following, we highlight why **standard FTRL techniques cannot work in our setting** when only bandit feedback is available. Specifically, notice that a FTRL meta-algorithm guarantees that the performances of the master algorithm are "near" the one of the optimal arm, namely, the optimal subroutine. Nevertheless, the optimal subroutine cannot trivially attain the same theoretical guarantees than when applied in the known C scenario, since, when employed as an arm, the subroutine receives the feedback with a certain probability. Indeed, each arm (including the optimal one) receives feedback only in the subsets of episodes in which that specific arm is chosen, thus, its guarantees are in principle arbitrarily worse than the one attained when playing at each round.
>
> Finally, we remark that it exists an extensive literature devoted to the sole scope of designing master algorithms to choose between no-regret minimizer subroutines, as it is not a trivial task under bandit feedback (see e.g., the following references). All these works study static regret.
>
> Since it is a crucial aspect of our work, please let us know if further discussion is necessary.
>
> [1] "Regret Balancing for Bandit and RL Model Selection", Abbasi-Yadkori et al. (2020)
>
> [2] "Corralling a band of bandit algorithms", Agarwal et al. (2016)

---

### Author Response · Authors · 2024-11-27

Dear Reviewers,

thank you again for the effort in reviewing our paper. We take the liberty to ask if our rebuttal has adequately addressed your concerns. We are happy to provide further clarification and engage in any discussion.

Thank you!

The Authors

---

### Author Response · Authors · 2024-12-04
**Part 1**

We want to thank again the Reviewers for their time and for the effort in evaluating our work. Based on the comments and questions posed by the Reviewers (for more specific discussion we answered each comment singularly), we kindly ask the Reviewers to consider the following recap considerations that we believe should help in solving their possible concerns.

>On the static baseline

Some Reviewers expressed concerns about the choice of a static baseline in the regret definition. We would like to readdress this point here, since it is a crucial point.

First, please consider the objective of this paper. Indeed, this paper's goal is to weaken the impossibility result for CMDPs with adversarial constraints and rewards to achieve both sublinear regret and sublinear constraints violation. Thus, we propose an algorithm with theoretical guarantees that are in between the purely stochastic case ($\mathcal{O}(\sqrt{T})$) and the worst case adversarial one ($\Omega(T)$). **To address this challenge we believe the static baseline is the more meaningful, as it is the one used by both extremes and the only one that allows the immediate interpretation of the result**.

In addition, please consider that our work's most natural related literature is the corruption robust online learning literature, which is still mostly unexplored for CMDPs but it is long established in other settings, as the unconstrained MDPs with corrupted transition probabilities and other online learning problems (e.g., [1],[2],[3],[4],[5],[6],[7],[8],[9],[10]). **Corruption-robust literature traditionally uses a static regret baseline, which is perfectly coherent with the definition of corruption itself**.

In that sense, **if the Reviewer think it might be of help to future readers, we would be happy to change the title of the paper to make a direct reference to corruption robustness instead of non-stationary to delimit and frame our work for what it is, and for what it is meant to do, solving the ambiguity with the stochastic non-stationary research line traditionally approached with dynamic regret**.

Finally, we want to remark that the task of attaining the desired guarantees with static regret when corruption is not known in advance is highly non-trivial, and a classic FTRL approach when only bandit feedback is available fails in such a task. Specifically, notice that a FTRL meta-algorithm guarantees that the performances of the master algorithm are ”near” the one of the optimal arm, namely, the optimal subroutine.
Nevertheless, the optimal subroutine cannot trivially attain the same theoretical guarantees
than when applied in the known $C$ scenario, since, when employed as an arm, the subroutine
receives the feedback with a certain probability, namely, the probability that arm is selected. To tackle the aforementioned problem, it exists an extensive literature devoted to the sole scope of designing master algorithms to choose between no-regret minimizer subroutines, as it is not a trivial task under bandit feedback (see e.g. [11] ). Consider in addition that even this literature use a static regret.

[1] Lykouris et al. "Corruption robust exploration in episodic reinforcement learning" (2021)

[2] Chen et al. "Improved Corruption Robust Algorithms for Episodic Reinforcement Learning" (2021)

[3] Wei et al. "A Model Selection Approach for Corruption Robust Reinforcement Learning" (2022)

[4] Jin et al. "Noregret online reinforcement learning with adversarial losses and transitions" (2024)

[5] Lykouris et al.,"Stochastic bandits robust to adversarial corruption" (2018)

[6] Yang et al., "Adversarial Bandits with Corruptions:
        Regret Lower Bound and No-regret Algorithm", (2020)

[7] Zhao et al. "Linear Contextual Bandits with Adversarial Corruptions", (2021)

[8] Bogunovic at al, "Stochastic linear bandits robust to adversarial attacks", 2021

[9] Lee et al., "Achieving near instance optimality and minimax-optimality in stochastic and adversarial linear bandits simultaneously", (2021)

[10] He et al. "Nearly Optimal Algorithms for Linear Contextual Bandits with Adversarial Corruptions", (2022)

[11] Agarwal et al. "Corralling a Band of Bandit Algorithms", (2017)

---

> ### Author Response · Authors · 2024-12-04
> **Part 2**
>
> >Comparison with Corral
>
> The online model selection research line, originally proposed by the seminal work of Agarwal et al. (2017) (which first proposed the Corral algorithm) and recently employed for unconstrained MDPs with adversarial transitions (see, [Jin et al., 2024]) focuses on the design of meta-procedures to choose between different instances of algorithms. However, these meta-procedures are specifically designed to deal with unconstrained problems, namely, their goal is single-objective: minimizing the regret. In our setting, minimizing the regret is not the only objective, since we want to additionally keep the violations as small as possible. Thus, it is not possible to blindly apply Corrall-like techniques, as it would lead to arbitrarily big violations.
>
> In our work, **we propose a Lagrangian approach to the online model selection problem, which we believe is of independent interest** since it can be easily generalized to the employment of different kind of sub-routines compared to the ones employed in our work, adapting the learning rate and the bonus function. Please notice that **using a Langrangified loss is not sufficient to guarantee bounded positive constraints violation.** First, we build the meta-algorithm's loss employing a **positive Lagrangian** function, which to the best of our knowledge is a novel tool in the online CMDPs literature. Furthermore, properly balancing the regret and the positive constraints violation requires a careful calibration of the Lagrangian multiplier. On the one hand the Lagrangian multipliers must be large enough to guarantee that it is not possible to compensate large constraints violation with a regret sufficiently negative; on the other hand, since the FTRL regret bound depends on the Lagrangian loss range of values, the Lagrangian multiplier must be small enough **not** to prevent FTRL from being no-regret (see Section F.1 for additional details on these aspects). Moreover, the rest of the analysis presents various challenges derived from the constrained nature of the problem and the definition of **positive** constraints violation (e.g., Section F.3). Thus, we believe that, while inspired by Corral for what concerns the stability results, **our analysis is novel and independent from existing works in online model selection.**
>
> Finally notice that our work presents results which hold in high probability, while both [Agarwal et al., 2017] and [Jin et al., 2024] attains results which hold in expectation only. To do so, we rely on an high probability definition of stability and we prove that from a corruption-robust algorithm in high probability it is possible to build a stable algorithm given our definition of stability (see Section H).

---

### Meta-Review · Area_Chair_aZyn · 2024-12-21

**Metareview:**

This paper studies CMDPs with non-stationary rewards and constraints, and propose an algorithm whose performance smoothly degrades as non-stationarity increases. More specifically, the algorithm achieves $O(\sqrt{T} + c)$ regret and positive constraint violation under bandit feedback, where $C$ is a corruption value measuring the environment non-stationarity.

The main weakness of the paper is the lack of technical novelty. For instance, the authors failed to demonstrate why lag-FTRL could be a versatile meta-procedure that deals with constrained problems.

Given the high standards of ICLR and the weakness mentioned above, I would recommend rejecting this paper.

**Additional Comments On Reviewer Discussion:**

The reviewers raised concerns regarding the choice of a static baseline in the regret definition, as well as the lack of technical novelty. Although the authors provided responses which addressed concerns regarding the regret definition, concerns regarding the lack of technical novelty remain.

---

### Decision · Program_Chairs · 2025-01-22

Reject